# Linear Superposition as a Core Theorem of Quantum Empiricism

**Yurii V. Brezhnev** 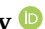

Department of Quantum Field Theory, Tomsk State University, 634050 Tomsk, Russia; brezhnev@phys.tsu.ru

**Abstract:** Clarifying the nature of the quantum state $|\Psi\rangle$ is at the root of the problems with insight into counter-intuitive quantum postulates. We provide a direct—and math-axiom free—empirical derivation of this object as an element of a vector space. Establishing the linearity of this structure—quantum superposition—is based on a set-theoretic creation of ensemble formations and invokes the following three principia: (**I**) quantum statics, (**II**) doctrine of the number in the physical theory, and (**III**) mathematization of matching the two observations with each other (quantum covariance). All of the constructs rest upon a formalization of the minimal experimental entity—the registered micro-event, detector click. This is sufficient for producing the $\mathbb{C}$-numbers, axioms of linear vector space (superposition principle), statistical mixtures of states, eigenstates and their spectra, and non-commutativity of observables. No use is required of the spatio-temporal concepts. As a result, the foundations of theory are liberated to a significant extent from the issues associated with physical interpretations, philosophical exegeses, and mathematical reconstruction of the entire quantum edifice.

**Keywords:** quantum foundations; non-axiomaticity; detector clicks; ensembles; superposition principle; arithmetic; numbers; vector space; abstracting; interpretations; self-referentiality

## 1. Introduction and Summary

> ... somewhat curious that, even after nearly a full century, physicists still do not quite agree on what the theory tells us ...—G. 't Hooft ([1], p. 5)

> It is almost a crying shame that we are nowhere close to that with quantum mechanics, given that it is over 70 years old now—C. Fuchs ([2], p. 32)

The contradiction between the fundamental nature of quantum theory (QT) and the phenomenological feature of its mathematics [3] is likely to never cease instigating the attempts to overcome it. As H. Putnam had said, "Human curiosity will not rest until ... questions [of the nature of the QT-formalism] are answered".

The subject-matter and leitmotiv of what follows is that the linear superposition and theory's axioms have an origin—they are derivable, and it is entirely empirical. The theory is thereby demystified, and the interpretative challenges that accompany the exegeses of QT are a *nonexistent* problem coming from "a confusion of categories" ([4], p. 89), i.e., from the "semantic confusion" ([5], p. 10). A direct outgrowth of this ideology is not only a derivation of the superposition principle (page 35) but also the axiom-free production of the "chief" quantum formula—the Born rule $p = |\mathfrak{a}|^2$ [6].

### 1.1. On the Foundations of Quantum Theory

The debates concerning the foundations of quantum mechanics (QM) hitherto "show no sign of abating" ([7], p. 222, [8,9]), and despite widespread scepticism [10–15], it is generally acknowledged that the problem is a real one [9,16–19]; it is not something made up or "just a dispute over words" ([20], p. 5) and sometimes "has been regarded as a very serious one" ([15], p. 418, [21,22]). Say, R. Penrose has expressed (2004) an even more radical "conviction that present-day quantum mechanics has no credible ontology, so that it *must* be seriously modified".

In recent decades, the discussions have even worsened [2,23], and current research has intensified due to the tremendously increased and formerly inconceivable technological possibilities of operating with individual micro-objects and the urge to implement the idea of quantum computing [20,24].

The reason for this state of affairs remains the same as it was before. Unlike the classical theories, e.g., thermodynamics or relativity theory, "Ma di assiomatizzazioni della teoria quantistica ce ne sono moltissime" ([17], p. 30) and the QM-axiomatics itself seems wholly divorced from human language [5,8,9,15,17,25–32]. Quantum postulates are not merely formal. They are phrased in terms of linear operators on a complex Hilbert space $\mathbb{H}$ [4,10,13,25,33–37] and, with that, literally not a single word here can be brought into conjunction with reality by means that have *at least some kind* of relationship with the classical description. What is more, it is very well known that the abstract character of these terms is required by the essence of the point (covariance) and, at the same time, that the attempt to link them with physical images is imposed by a decree and results in the famous paradoxes associated with concepts, such as causality, (non)locality, and realism [9,27,28,38–45]. All of that causes a problem with interpretations of QM.

It is well known that the theory has steadfastly resisted any unique ontological reading and, in particular, reconciliation between interpretations. This is reflected not only in the voluminousness of the literature. The differences in viewpoint are often based on points of principle [3,8,14,15,46–51], and even highly qualified publications face criticism [52–55]. Among other things, we encounter appeals [3,12,16,17,43,56–58] (there is even a manifesto(s) [50], (p. 990), [59], ), striking titles such as "scandal of quantum mechanics" [60,61], "QUANTUM OUTCOME: ALLAH WILLED IT?" ([62], p. 188; Wheeler), "the Oxford Questions ... to two clouds" ([63], p. 6), "The Canon for Most of the Quantum Churches" ([50], p. 988), "Quantum mechanics for the Soviet naval officers" ([64], p. 161), "the patron saint of heretics in the One True Church of Copenhagen" (about D. Bohm), "A Feminist Approach to Teaching Quantum Physics" ([2], p. 182), "Church of the Larger Hilbert Space" (J. Smolin) [2,12], and also April Fools' [65] and the medical jokes about "the "state of health of the quantum patient'" ([66], p. vii), political parallels with "Marxism ... [and] the Cold War" [67], and many more [3,9,27,68–71].

An interesting fact is that Cambridge University Press has published a 500-page-long book [2] containing an arresting electronic correspondence—D. Mermin called it "samizdat" (self-published)—between C. Fuchs and modern researchers and philosophers in the field of quantum foundations. This correspondence has continued ([23], over 2300 pages) and now covers 1995–2011. It characterizes the state of affairs in the field, and does not merely add to one's impression of the unending discussions about quantum matters (see introductory sections in [50] (!) and in [64]), it also represents, due to the lack of formality, a plentiful source of ideas and of valuable thoughts. Schlosshauer's very informative "quantum interviews" [16] pursue the same goals.

It is worth mentioning that the quantum challenges had led, quite a while ago, to attempts to revise, even formalizing, the logic of our thinking [72,73]—a very nice mathematical theory dating back to von Neuman in the 1930s ([25], Section III.5) termed quantum logic [74]. There are handbooks on that subject [75], and this topic is still under intensive investigation now. See also the last paragraph in Section 6.3.1.

The lack of transparent motivations for mathematics—a pressing requirement of physics—means that QM-formalism is hard to distinguish from a "cook book of procedures and rituals" (J. Nash), a "user-manual" ([32], p. 1690), [76], ([27], p. xiii), or from a "library of ... tricks and intuitions" [21]. Therefore, the "dissatisfaction regarding comprehension" and the "need for interpretation that is alien to an exact science" ([77], pp. 7–8) lead to the fact that "we admit, be it willingly or not, that quantum mechanics is not a physical theory but a mathematical model" ([32], p. 1701) or that "nature imitates a mathematical scheme" ([78], p. 347; Heisenberg). De facto, QT "is in a sense like a traditional herbal medicine used by "witch doctors". We do not REALLY understand what is happening" (J. Nash) and "we have essentially *no* grasp on why the theory takes the precise structure that it does" ([2],

p. 32), which raises the suspicion that "something is wrong with the theory" (H. Putnam) and that "this quantum skyscraper is built on very shaky ground" ([64], p. 8). (Throughout the text, the *italic* and *slanted* type in "quotations" is original, unless otherwise indicated.)

At the same time, well-founded opinions have long been known to the effect that "quantum theory needs no "interpretation'" [43], in refs. [3,12,60] or that "only consequences of the basic tenets of quantum mechanics can be verified by experiment, and not its basic laws" ([11], p. 16). In other words, the discrepancies between opinions are significant and often radical: from epithets such as "schizoid, … situation is desperate" ([15], p. 420), ([79], p. 424) to supporting the rationale for quantum computations [24] and whole books written on the subject [8]. Concerning the "schizoid", the case in point is the many-world conception by Everett–DeWitt. See also pages 158, 161, 176–179 in [80] regarding the "state of schizophrenia" and "explanations" as to why "schizophrenia cannot be blamed on quantum mechanics" ([80], p. 182).

In any case, the controversy between "the warring factions, …, many [quantum] faiths, … and instrumentalist camps" ([16], pp. 60–61), ([81], Section 5.5), [30,33]—"[t]hey all declare to see the light, the ultimate light" ([50], p. 987)—cannot be considered as an acceptable state of affairs (see also Section 11.1) for the simple reason that the quantum philosophy issues turn into an "industry"of interpretations—an unhealthy state of affairs—while, at the same time, the very same philosophers call for its denial: "interpretation of QM emerged as a growth industry" ([82], p. 92).

*1.2. Formula of Superposition*

Conversely, the "dominant role of mathematics in constructing quantum mechanics" has led to the conclusion that mathematical "assumptions are usually considered to be physical" ([32], p. 1691). That is to say, "there has been a substitution of concepts" ([76], p. 295) and "one of the consequences of quantum revolution was the replacement of explanations of physical phenomena by their mathematical description" ([76], p. 296). These characteristics convey, in the best possible way, the dissatisfaction with the fact that quantum physics "was actually reduced to a physical interpretation of the Hilbert space theory" ([32], p. 1690). The $\mathbb{H}$-space in itself is a fairly cumbersome mathematical structure and even determines a crucial principle: the superposition of states [26]. It is thus not surprising that this principle becomes "one of the vague points … the [Dirac] argument is difficult to consider as rational … the physical principle simply fits underneath it" (excerpt from the preface to the Russian edition of [83]).

The mathematics of the $\mathbb{H}$-space contains three constituents: a vector space, the inner-product add-on over it, and topology. The two latter ones invoke the first one, which is completely independent (algebra) and begins with the formula

$$|\psi\rangle = \mathfrak{a} \cdot |\varphi\rangle + \mathfrak{b} \cdot |\chi\rangle \,. \tag{1}$$

This is the pivotal expression of quantum theory. Comprehending its genesis is tantamount to comprehending the nature of the *linearity* of QM.

In Formula (1), there occur the complex numbers $\mathfrak{a}, \mathfrak{b} \in \mathbb{C}$, symbols of operations $\cdot$ and $+$, and also vectors $|\psi\rangle, |\varphi\rangle, |\chi\rangle \in \mathbb{H}$. It is clear that until an empirical basis for all these devices is found, the interpretation of Abstraction (1) and questions of the kind "Quantum States: What the Hell Are They?" (55 times in [23]) will remain a problem. To all appearances, the problem is considered so difficult—"quantum states … cannot be "found out'" ([8], p. 428)—that the non-axiomatic meaning of these symbols was not even discussed in the literature. In the meantime, not only is the situation far from hopeless, but it also admits a solution. The present work is devoted to gradual progress towards an understanding of Formula (1). Stated differently, Equality (1) becomes an empirical "theorem" (p. 56).

- - The main part of the challenge is not only to ascertain what is being added/multiplied in Formula (1), but also to realize primarily *what "to add/multiply" is*, and "Where Mathematics Comes from" [84] at all.

"What does it mean, physically, to "add" things?" [2], (p. 178; D. Darling). More than that, aside from the symbols $\{\mathfrak{a}, \mathfrak{b}, |\psi\rangle, |\varphi\rangle, |\chi\rangle, \cdot, +\}$, Expression (1) contains the sign of equality = (see also [85] (pp. 29, 30 (!), [86]), and, surprising as it may seem, it conceals one of the key points—the third principium of quantum theory (**III**, p. 29).

The guiding observation is based on the fact that the only thing that we have access to are the microscopic events, and therefore, "we have little to begin with other than what an experimental physicist would call experiments with a single microsystem" ([87], p. 5).

> "[W]e must recognize that the focusing on individual elements whatever these may be is absolutely indispensable for all our thinking. … What may be regarded as an individual event?"
>
> R. Haag ([88], p. 302)

Consequently, we must begin from individual events and from collecting them into ensemble formations. It is precisely in this context that we will use the word empiricism—quantum empiricism of micro-acts of perception—and it is in this respect that QT has a statistical nature. As Einstein put it, "It may be a correct theory of statistical laws, but an inadequate conception of individual elementary processes" ([30], p. 156; Einstein); see also ([25], ([30], Chs. 7–8), [70], [89], pp. 38–40), ([89], p. 40). Such a viewpoint has been long championed by L. Ballentine [90] and H. Groenewold ([91], p. 468) and justified in detail by G. Ludwig [87,92–94]. A. Leggett proposes accordingly "extreme statistical interpretation"[16] (p. 79) , [95], in the sense that "to seek any further "meaning" in the formalism is pointless and can only generate pseudoquestions". With that, he overtly applies such characteristics as "complete gibberish" ([95], p. 70) and "verbal window dressing" ([16], p. 79).

The difficulty is, of course, in creating the object $|\psi\rangle$ itself. A step-by-step characterization of this procedure (Sections 3–8) and key words to what follows have been reflected in the (sub)section titles listed in the Contents.

*1.3. Physics $\rightleftarrows$ Mathematics; Doctrine of Numbers*

Thus, the situation appears to be one whereby the physics itself faces inconsistencies in its foundations and the mathematical superstructures are difficult to reconcile with its motivations (physical principles) [96]. However, on the other hand, attempts to axiomatize an interface between them [97] only conceal a deeper insight [22]. M. Born had called attention to the fact that "probable refinements of mathematical methods will not suffice to produce a satisfactory theory, but that somewhere in our doctrine is hidden a concept" and T. Maudlin was more definite: "physicists have been misled by the mathematical language they use to represent the physical world".

In other words, we observe an overemphasis on the role of the ready-made math-structures—algebras, spaces, and the like—and an under-evaluation of "seemingly naïve" empirical aspects voiced in the ordinary language [98]. The situation is no different from that which H. Weil had characterized in the introductory section to ([99], p. 10) as follows.

> "All beginnings are obscure. Inasmuch as the mathematician operates with their conceptions along strict and formal lines, he, above all, must be reminded from time to time that the origins of things lie in greater depths than those to which their methods enable them to descend".

The "origins" are expressible of course only in the natural language; Section 2 is devoted to this.

What we propose below is an implementation of the idea that the postulational view must be abandoned and replaced by a negation of the prior existence of both the physical "preconceived notions" [92] (p. 328) and the mathematical structures. Physics and mathematics should be created "from scratch". Paul Benioff calls this idea "a coherent

theory of physics and mathematics" [2] (p. 33; P. Benioff), [96] (p. 639), Then, due to the initial absence of mathematics, introducing mathematical structures is almost ruled out, proofs must be replaced by an empirical inference, and semantics of physics—the language of physical reasoning—is initially under a linguistic ban. It cannot exist a priori. That is to say, even the natural-language conjunction of mathematical terms with physical adjectives (and verbs [100] (p. 3102; "to happen, to be, to exist")) becomes far from being free, as with the classical description's language (Sections 2.1, 5.4, 6.4 and 6.5). R. Haag, on the first page of the work [101], emphasizes:

- "we should not consider ["vocabulary of Quantum Theory"] as sacrosanct. ... every word in the vocabulary is subject to criticism".

Returning to the ensemble formations, it is only they that have to come to the fore, and argumentation should be subordinated only to the low-level microscopic empiricism. The predominance of the empirical over the theoretical will then immediately touch on the closest creature of the latter—the notion of a number—since numbers do not come "from the sky", and the theory will have to be a quantitative one.

Despite the overflow of abstracta in QT, the *doctrine of number*—⌈number × unit⌉, to be precise (Sections 7 and 9)—has, it seems, not yet entered foundational discussions [102]. Consequently, the numbers turn into a kind of "problem of numbers" (principium **II**), and we are thus led to the necessity of revising the take on the foundations themselves:

$$\lceil \text{quantum fundamentals} \rceil \quad \dashrightarrow \quad \lceil \text{the problem/doctrine of the number} \rceil .$$

This paradigm shift is a unique trait of the quantal (not the classical) view of things and a substantial part of the following is devoted specifically to that.

In the outline of the present work, the workflow will constitute re-creating the structure of a linear vector space. More precisely, producing an *a priori unknown* mathematics, which *will be* an algebra of such a space with a complex conjugation. As a matter of fact, we provide an answer to Haag's question "How do we translate the description of an experimental arrangement into mathematical symbols?" in the context of their own "idea of basing the interpretation of quantum theory on the concept of "events" which may be considered as facts independent of the consciousness of an observer" ([88], p. 295).

The main point to be immediately emphasized is that the mathematization of the discrete micro-acts of observations is quite a nontrivial procedure (105), and further, the strategy, along with the structure of this article, can be schematized as follows.

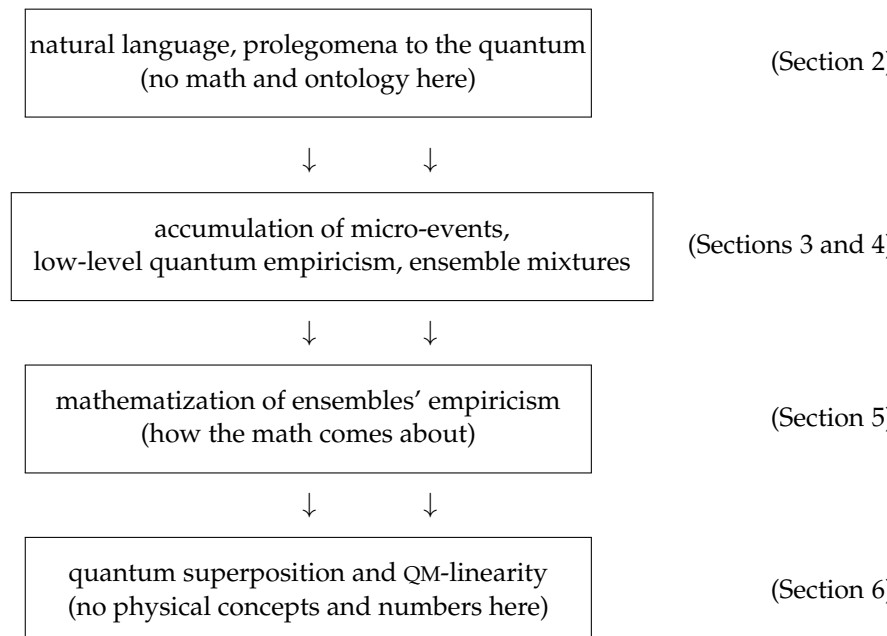

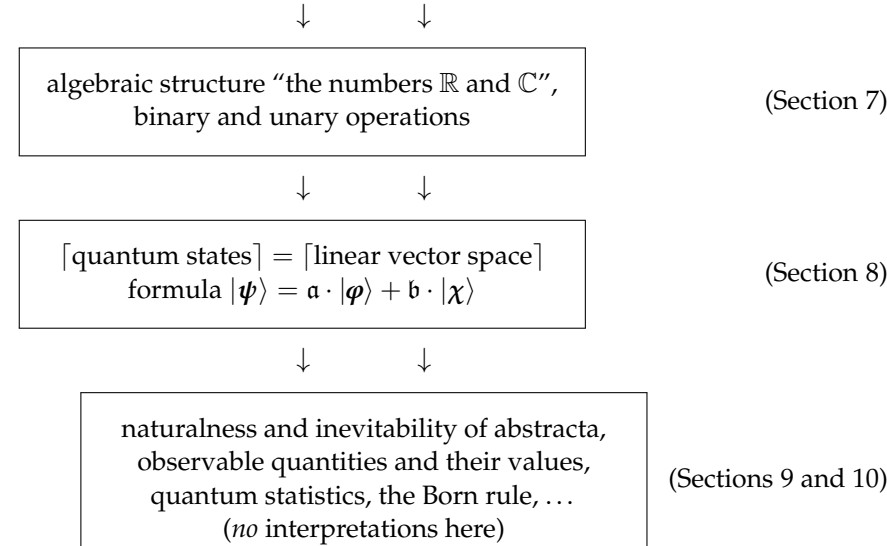

This box-diagram cannot be reduced or restructured. For example,

- *Superposition foregoes numbers*, and measurement and physical properties follow *strictly after* the $|\texttt{ket}\rangle$-vectors have been created.

By and large, the aforesaid ideology is supported by the common belief—often certainty even [12]—that QM is *not* perturbative, its linearity is *not* associated with linear approximation of something else, and, in general, it is *not* extensible (ultimate [103] and non-deformable) and must be free of interpretations [12,43]. All of these concerns, in one way or another, are directly related to the derivation of Formula (1).

## 2. Points of Departure

In the Beginning was the Word—A. Zeilinger ([39], 01:05′47″)

Most of the time the apparatus is empty and sometimes you have a photon coming through—A. Zeilinger ([39], 12′39″)

Since empiricism is in essence supra-mathematical [104], i.e., it is concerned with *meta*mathematics [105,106], its mathematization, i.e., theory construction, should begin not with postulates and definitions, but rather with the semantic formation of an object language (of "the Quanta") of vocabulary that "may only be described by "words" and not by a theory" [58], [87] ( p. 106), [93]. As A. Peterson and K. Popper had observed, "Math can never be used in phys until have words" ([107], p. 209), i.e., "we cannot construct theories without using words" ([108], p. 12). Therefore, relying on the established understanding of the underlying causes for the quantum eye on physics [13,25,27,33], up until the end of this section, we will adopt the natural-language meaning of the words observation, system, state, numbers (!), plus and to divide, physical influence, transition, large/small, micro/macro, etc. Their contents will later be defined more precisely or entirely changed. For instance, the sense of the word "state" will be drastically transformed, to which we are drawing attention in advance. Accordingly, a degree of informality—it has been clarified in Remark 10—is inevitable here, but "the lack of precision … is a necessity" ([109], p. 48) at the moment.

### 2.1. Variations as Micro-Level Transitions

We will (and "must" [105] (Ch. 3)) first view the concept of a system at an intuitive level ([110], Section 1.1)—there is what is referred to as an isolated system $\mathcal{S}$.

**S**    Let us tentatively (a priori) relate the concept of a *state* to the associated context describable by the words "the system $\mathcal{S}$ can *vary*, *be different*, or *in different states*". That is to say, system $\mathcal{S}$ is always in a certain state $\underline{\Psi}$ belonging to the set $\mathfrak{T} = \{\underline{\Psi}, \underline{\Phi}, \ldots\}$,

each element of which is admissible for $\mathcal{S}$, and all of them are different from each other: $\underline{\Psi} \neq \underline{\Phi}$.

In other words, the concept of a quantum system may not have a precise/axiomatical definition at the moment. Otherwise, if it comes to that, the system is what is being constantly varied when observed, and "varied" is the key word here.

The statement "states are different" does not require a consideration when $\underline{\Psi}$ and $\underline{\Phi}$, referred to as state, are the abstract elements of an abstract set $\{\underline{\Psi}, \underline{\Phi}, \ldots\}$. However, in order to tie its elements to reality, we have to introduce the criteria of coincidence/distinguishability of one from the other. Criteria may not come from observational procedures, without which it is impossible to either detect states or claim that they differ, coincide, or that they are, if any.

On the other hand, the nature of micro-phenomena shows that observations are always associated with an irreducible intervention in the system, manifesting in what is known as *transition* $\underline{\Psi} \rightsquigarrow \underline{\Psi}'$ (or destruction). As an example, observations at accelerators are literally the destructions, and bulk at that. Due to a lack of criteria, there is no sense in attributing to this concept the adjectives small/large, (in)significant/partial, or collocations such as "comparison of destructions at instants $t_1, t_2$". Let us proceed from the idea that initially there is nothing but the transition. Transitions may actually occur without destruction $\underline{\Psi} \rightsquigarrow \underline{\Psi}$, however.

Two different $\underline{\Psi}, \underline{\Phi}$ may be destroyed into new $\underline{\Psi}', \underline{\Phi}'$, as well as into the combinations of the old/new. Thus, strictly speaking, the sense of words "different, new, ..." eludes us in this case, which is why even the identification of $\underline{\Psi}$-elements and the $\mathfrak{T}$ itself, as a set, becomes questionable. Therefore, besides the formal writings $\underline{\Psi} = \underline{\Phi}$ and $\underline{\Psi} \neq \underline{\Phi}$ for $\underline{\Psi}, \underline{\Phi} \in \mathfrak{T}$, the *physical* distinguishability/equivalence (recognizability $\not\approx / \approx$) needs to be established. As to the identification (and to the identity) in this regard, see von Neumann's reasoning: "`One might object against ∥ ...`" on page 302 of their book [25]. The sole thing that distinguishability may rely on is the transition acts. In turn, variation is a key element in transitions, which is why we will begin constructing with distinguishability.

Let us take the still virtually unlimited way $\mathscr{A}$ of intervening $\overset{\mathscr{A}}{\rightsquigarrow}$ in $\mathcal{S}$ and attempt to introduce distinguishability $\underline{\Psi} \not\approx \underline{\Phi}$ as $\mathscr{A}$-distinguishability. Due to the fact that micro-transition $\underline{\Psi} \overset{\mathscr{A}}{\rightsquigarrow} \underline{\Psi}'$ is not pre-determined, initial states $\underline{\Psi}$ undergo arbitrarily free changes. Next time, the results will be different and absolutely arbitrary (the term "different" is understood to mean $\neq$), and each act is indiscernible from a case in which it contains ones similar to itself within itself. It would be natural to associate such a case to the absurd, which is unrelated to the meaning of the words "physical observation", and to discard the given $\mathscr{A}$.

Non-meaninglessness arises only if we impose the negation of random combinations of $\neq$ and $=$ in transitions, at least for a part of $\mathfrak{T}$, i.e., introduce the preservation acts $\underline{\Psi} \overset{\mathscr{A}}{\rightsquigarrow} \underline{\Psi}$. The "preservation" should be read here as indestructibility of state, i.e., as a $(=)$-coincidence under the secondary act $\underline{\Psi} \rightsquigarrow \underline{\Psi} \rightsquigarrow \underline{\Psi}$. Otherwise, the vanishing difference between "preservation" and "variation" leads to linguistic chaos ([111], p. 232). This means that the destruction $\underline{\Psi} \rightsquigarrow \underline{\Psi}'$ may not be considered as a one-fold one. State $\underline{\Psi}'$ on the right should be examined for changeability and transform into the left part of the subsequent transition: $\underline{\Psi} \rightsquigarrow \underline{\Psi}' \rightsquigarrow \underline{\Psi}''$. Thereby, the structure $\underline{\Psi}' \rightsquigarrow \underline{\Psi}''$ with the *binate* entity "before/after" or "on the left/right" becomes the key one, and we consider it an initial object in subsequent constructs. The preserved states are, by definition, those that pass the reproducibility test.

Thus, logic requires beginning with the transition compositions

$$\underline{\Psi} \overset{\mathscr{A}}{\relbar\rightsquigarrow} \underline{\Psi}' \overset{\mathscr{A}}{\relbar\rightsquigarrow} \underline{\Psi}'' \overset{\mathscr{A}}{\relbar\rightsquigarrow} \cdots ,$$

wherein the cases such as

$$\cdots \quad \underline{\Psi}' \overset{\mathscr{A}}{\relbar\rightsquigarrow} \underline{\Psi}' \overset{\mathscr{A}}{\relbar\rightsquigarrow} \underline{\Psi}'' \quad \cdots \tag{2}$$

are ruled out (a ban on changing of what has been unchanged), and the never-ending sequence

$$\underline{\Psi} \xrightarrow{\mathscr{A}} \underline{\Psi}' \xrightarrow{\mathscr{A}} \cdots \xrightarrow{\mathscr{A}} \underline{\Psi}'' \xrightarrow{\mathscr{A}} \cdots \qquad (3)$$

(non-recognisability of states) must be terminated

$$\underline{\Psi} \xrightarrow{\mathscr{A}} \underline{\Psi}' \xrightarrow{\mathscr{A}} \cdots \xrightarrow{\mathscr{A}} \underline{\Psi}'' \xrightarrow{\mathscr{A}} \underline{\alpha} \xrightarrow{\mathscr{A}} \underline{\alpha} \,, \qquad (4)$$

yielding a "finiteness" (= realisticness) and the concept of conserved/distinctive $\underline{\alpha}$-states. The terminology $\underline{\alpha}$-event [12] could be used instead.

Freedom of elements in Sequence (4), including the choice of $\underline{\alpha}$-states, is not limited by anything besides the ban on Equation (2). Therefore, this arbitrariness, which is physically never recognizable, curtails the generic chain from Sequence (4) into the shortened one

$$\underline{\Psi} \xrightarrow{\mathscr{A}} \boxed{\cdots\cdots} \xrightarrow{\mathscr{A}} \underline{\alpha} \xrightarrow{\mathscr{A}} \underline{\alpha} \,, \qquad (5)$$

which is identical to the scheme

$$\boxed{\cdots \underline{\Psi} \cdots} \xrightarrow{\mathscr{A}} \underline{\alpha} \qquad (6)$$

with certain $\underline{\alpha} \in \mathfrak{T}$.

Discussions on "what happens … [and] "how" ([25], p. 217) at the very microscopic level are extremely widespread in the literature [18,41,44,45,53,56,77] (see [19,33,40,112] for the exhaustive references), although it is not difficult to predict the fact that the attempts to understand the inner structure of Box (6) will only lead back to an identical box; so, the "turtles all the way down" (ascribed to W. James), followed by the great Wheeler's slogan "No tower of turtles" (1989).

Indeed, the uncontrollability of micro-changes is universally known, yet describing them as a process in time $t \mapsto t + \varepsilon$ will start employing language terminology—functions, arithmetic operations, the physical words, etc.—that has not yet been created even for the fixed instants $t_1, t_2$. However, what may be associated with fixed time are only non-temporal entities, for which we have nothing but transitions (Equation (5)). The attempt to manage them, i.e., to control intervention in $\mathcal{S}$, results in looping or "measuring the measurement", in addition to the ambiguity of this term itself.

> "[I]t is not meaningful to speak of a measurement "at time $\tilde{t}$." … the real physical meaning of the time parameter … has nothing to do with the notion "time of measurement"". "[T]he description of the measurement process in quantum mechanics in terms of "pre-theories" is not possible"
>
> G. Ludwig ([87], p. 288), ([92], p. 340)

See also [58] (p. 100), [92] (p. 365), [94] (p. 150), [113] (pp. 644–646), and [114]. Just as before, the physical assessments such as "abrupt", "(ir)reversible", "(non)simultaneous", "immediately following …" [25] (pp. 231, 410), or the "weak/nondemolishing" (measurements [53]), etc.are unacceptable here. No temporal process may be present in the foundations of the theory (([87], Sections VII.4, 6), ([92], Chs. III, XVII), [93]) since it is immediately not clear: "Furthermore, what exactly are we having at instants $t_1$ or $t_2$?". In the reverse direction—⌈time ⇢ measurement⌉—the situation is also rather indefinite since the ""Time" is not an entity to which the operations of measurement, direct or indirect, apply" ([114], p. 5).

**Remark 1.** *All the information stated above means that attempts to deduce* QM *dynamically ([16], 10 · Reconstructions) are beforehand doomed to vicious circles "round the boxes" and time t, such as attempts to dynamically "vindicate" Lorenz's contraction instead of kinematic postulates of the relativity theory [69]. A consistent theory must rest either on "irreducible" elements (6) or upon "boxes" of a different kind. In the latter case, the theory becomes a particular* model with interpretation; *e.g., the Lindblad equations [115,116], decoherence [112,117–119], stochastic dynamics,*

*and other statistic-dynamical models [40,77]. Anyway, an ability to model and understanding are not the same thing, and this point was repeatedly emphasized in the literature ( [16], ([17], Section I.2), [32]) with regard to* QT.

That said, if theory is built *as a fundamental one rather than as a model* ([16], p. 144), with a primary entity *changeability* $\overset{\mathscr{A}}{\leadsto}$, Box (6) may only be involved in it as the initial starting point and as an *indescribable* object with the absolute rather than with a relative sense. Elements of reality, in whatever understanding—say, Bell's "*be*ables" [28]—may not exist before/after/inside/outside of the box. It can only be the *structureless abstractio*. Accordingly, the notions of preparation, of measurement, of "interaction with", and of a physical process are meaningless without the construction of Box (6).

These statements are clearly in agreement with the fact that any reasoning must not contradict the formal logical rules [105]; hence, there must exist [96,106,120] the empirically undefinable logical atoms. A. Peres writes ([121], p. 173): "While quantum theory can in principle describe *anything*, a quantum description cannot include *everything*. In every physical situation *something* must remain unanalyzed". Moreover, as Pauli put it, "Like the ultimate fact without any cause, the individual outcome of a measurement is … not comprehended by laws". Specifically, the set $\mathfrak{T}$ and transitions arrows $\overset{\mathscr{A}}{\leadsto}$ are also the atoms. It is a "… preexisting concept … We cannot formulate the theory without this concept", concludes B. Englert ([12], p. 2). From the aforesaid, we may formulate the following tenet.

Quantum statics should forego quantum dynamics.

(*The first principium of quantum theory*)

The rationales do not end here and will be later amplified once we begin to exploit the terminology that is usually taken for granted from the outset, viz., the quantitative descriptions ([2], p. 178). If they arise not as numerical interpretations of something but out of an experiment, then observation should be the beginning, and the "manufacture of numbers'—the end. In other words, the model "theory with boxes" other than Boxes (5) and (6) implicitly implies the logical sequence $\lceil$model of process$\rceil \rightarrowtail \lceil$numerical interpretation$\rceil$, in which empiricism holds a role other than primary. It is clear that, regardless of the model, such a situation will always remain unsatisfactory in the physical respect.

*2.2. Observation*

The sequences addressed above lead to the following outcome.

- Any meaningful micro-act $\overset{\mathscr{A}}{\leadsto}$ either saves a state ($\underline{\alpha} \overset{\mathscr{A}}{\leadsto} \underline{\alpha}$) or turns it into a conserved one ($\underline{\Psi} \overset{\mathscr{A}}{\leadsto} \underline{\alpha}$).

The two extremes do not contradict this fact. The first—maximally rough observations—is when all states are destroyed into a certain one: $\underline{\Psi} \leadsto \underline{\Psi}_0$ ("whatever and however we watch, all we see is one and the same"). In this, the state $\underline{\Psi}_0$ is not destroyed: $\underline{\Psi}_0 \leadsto \underline{\Psi}_0$. Another extreme is when none of the states are destroyed: $\underline{\Psi} \leadsto \underline{\Psi}$. This is the case of ideal (quantum) observation, but, due to the absence of any changes, it is indistinguishable from the case where observations are entirely absent.

Situated in between these extremes lies the simplest case with two distinctive states

$$\underline{\alpha}_1 \overset{\mathscr{A}}{\dashrightarrow} \underline{\alpha}_1, \qquad \underline{\alpha}_2 \overset{\mathscr{A}}{\dashrightarrow} \underline{\alpha}_2 . \tag{7}$$

Of course, these are prohibited from transitioning into each other. Because there is still the free admissibility of transitions $\underline{\Psi} \overset{\mathscr{A}}{\leadsto} \underline{\alpha}_1, \underline{\Psi} \overset{\mathscr{A}}{\leadsto} \underline{\alpha}_2$, we can turn the semantic sequence

$$\lceil \text{arbitrariness} \quad \rightarrowtail \quad \text{preservation} \quad \rightarrowtail \quad \text{distinctive } \underline{\alpha}\text{'s} \rceil$$

into the more rigorous scheme

$$\lceil \mathfrak{T} = \{\underline{\Psi}, \underline{\Phi}, \ldots\} \rceil + \lceil \mathscr{A}\text{-observations} \rceil \quad \rightarrowtail \quad \{\underline{\alpha}_1, \underline{\alpha}_2, \ldots\} =: \mathfrak{T}_{\mathscr{A}} \subset \mathfrak{T} , \tag{8}$$

which gives, even though partially, rise to the concept of a physical distinguishability ("distinguo"). It is formally defined only on the subset $\mathfrak{T}_{\mathscr{A}}$: the statement $\underline{\alpha}_1 \not\approx \underline{\alpha}_2$ is equivalent to (7). To avoid overloading the further notation, we do not use symbols such as $\approx_{\mathscr{A}}$ and $\not\approx_{\mathscr{A}}$; the context is always obvious.

**O** By a *physical observation $\mathscr{A}$* or, in short, *observation* we will mean such interventions $-\!\!\overset{\mathscr{A}}{-}\!\!\rightarrow$, in which the "never-ending" chaos (3) is replaced by chaos with the notion of preservation, i.e., "chaos with rule (6)":

$$\underline{\Psi} -\!\!\overset{\mathscr{A}}{-}\!\!\rightarrow \underline{\alpha}, \quad \text{where} \quad \underline{\alpha} -\!\!\overset{\mathscr{A}}{-}\!\!\rightarrow \underline{\alpha}. \tag{9}$$

The set of $\underline{\alpha}$-objects $\mathfrak{T}_{\mathscr{A}}$ with the property

$$\underline{\alpha}_1 -\!\!\overset{\mathscr{A}}{-}\!\!\rightarrow \underline{\alpha}_1, \qquad \underline{\alpha}_2 -\!\!\overset{\mathscr{A}}{-}\!\!\rightarrow \underline{\alpha}_2, \qquad \ldots \tag{10}$$

is discrete, and the $\underline{\alpha}_s$ themselves are termed *the eigen* (proper) *for observation $\mathscr{A}$*. They define $\mathscr{A}$ and do not depend on $\mathcal{S}$. No logical connection between $\underline{\Psi}$ (the left of (9)), family $\mathfrak{T}_{\mathscr{A}}$, and system $\mathcal{S}$ exists.

(The comprehensive terminology here is this: a micro-act of observation by instrument $\mathscr{A}$. The zig-zag arrow $\rightsquigarrow$ is replaced with the straight one $-\!\!-\!\!\rightarrow$.) Expressed another way, the introduction of the concept "the eigen" is equivalent to the following informal, yet minimal, motivation: *at least some* certainty instead of *total* arbitrariness.

Two instruments $\mathscr{A}$ and $\mathscr{B}$ may have arbitrarily different eigen-states $\{\underline{\alpha}_1, \ldots, \underline{\alpha}_n\} \neq \{\underline{\beta}_1, \ldots, \underline{\beta}_m\}$. Accordingly, as regards observation $\mathscr{B}$, the (distinctive) states $\{\underline{\alpha}_s\}$ do not differ, in general, from the "regular" $\underline{\Psi}$'s, i.e., from those chaotically destroyable into the $\mathscr{B}$-eigen states: $\underline{\alpha}_j -\!\!\overset{\mathscr{B}}{-}\!\!\rightarrow \underline{\beta}_k$. All kinds of instruments $\{\mathscr{A}, \mathscr{B}, \ldots\}$ are thus defined by aggregates $\{\mathfrak{T}_{\mathscr{A}}, \mathfrak{T}_{\mathscr{B}}, \ldots\}$. The number $|\mathfrak{T}_{\mathscr{A}}|$ of corresponding $\underline{\alpha}$-objects therein may be an arbitrary integer. There are also no (logical) grounds for restricting/prescribing the composition of $\mathfrak{T}_{\mathscr{A}}$. Any element of $\mathfrak{T}$ may be the conserved one for a certain instrument. Parenthetically, the notion of an eigen-state—in different forms—is sometimes present in axiomatics of QM [18,72,122].

In a generic case, the chaos present in Rule (9) leaves open the problem of correlating the recognizability $\underline{\Psi} \not\approx \underline{\Phi}$ (or $\underline{\Psi} \approx \underline{\Phi}$) with physics. Clearly, the issue is linked to the ambiguity of the term $\underline{\Psi}$-state itself, which is used in pt. **S**—an important point—because we need to start with something since building the mathematical description without some sort of a set is impossible.

**Remark 2.** *Informally, metalinguistic semantics—the association of meanings with texts [58]—is in general as follows. Inasmuch as we are receiving* different $\underline{\alpha}$-responses to each micro-act $-\!\!\overset{\mathscr{A}}{-}\!\!\rightarrow$, *let us say that "on the other side from us there is something that can also be different", and all of that is to be described. This reflects our intuitive perception of reality, which, both at the micro-level and the macro-level, boils down to pt. **S** and to an ineradicable pair:* $\lceil$something outside$\rceil$ + $\lceil$that which can be different for us$\rceil$. *If we give up either of these semantic premises—"something outside" ($\underline{\Psi}$) or "can be different" ($\underline{\alpha}_j$)—then, as above, we face a linguistic dead end, as the possibility for reasoning disappears. There must be two sides present. Because of this, the arrow $-\!\!\overset{\mathscr{A}}{-}\!\!\rightarrow$ must be accompanied by "some things" to the left and right of it. The low-level set $\mathfrak{T} = \{\underline{\Psi}, \underline{\Phi}, \ldots, \underline{\alpha}_1, \underline{\alpha}_2, \ldots\}$ does arise. Then, the arbitrary elements $\underline{\Psi} \in \mathfrak{T}$ (unrestricted chaos) are assigned to the left of this act instead of "some thing" and the micro $\not\approx$-distinguishable $\underline{\alpha}$-objects ($\underline{\alpha}_j \not\approx \underline{\alpha}_s$) to the right. Put another way,*

- *What is being abstracted is not "concrete things" ([13], p. 27) or behavior of things" ([123], p. 414) but a primitive element of perception—a micro-event—the $\underline{\alpha}$-click. Other than "the click", no entities, such as very small objects/particles, fields, or, much less the knowledge, human psychology, "personal judgments", "memory configuration" [52,124], "mysterious interaction . . . brain of the observer" [108] (p. 11, thesis 3), [113] (p. 645), agents, their belief/consciousness [55,71,125], etc., may exist in empiricism. This is a kind of "Radical Empiricism . . . [by] William James" ([23], pp. 289, . . . ). The "click . . . and nothing more" ([16],*

*p. 42; Č. Brukner) is a kind of experimental zero-principium of* QT. *Therefore, the initial math premises of* QT *should contain nothing but the $\not\approx_{\mathscr{A}}$-distinguishability and formalization from* (9) *and* (10).

*Ideas of "a click (signal) in a counter" ([126], p. 758) have, time and again, already been expressed in the literature [2] (A. Peres), [123], and we draw attention to answers of Č. Brukner on pp. 41–43 in [16], their work [127] (p. 98), and page 635 in [22]. "Having grown up collecting clicks … I would start with "clicks" as the only point of contact between observer and observed", wrote J. Summhammer in [23] (p. 261). It may be added that the micro-observation, as such, is terminated at the eigen elements; one and the same $\underline{\alpha}_j$ has always remained on the right.*

As a result, the minimal entity $\underline{\Psi} \;\text{-}^{\mathscr{A}}\text{→}\; \underline{\alpha}_j$ constitutes, mathematically, an ordered pair $(\underline{\Psi}, \underline{\alpha}_j)$ of elements of the set $\mathfrak{T}$, which are labeled by the symbol $\mathscr{A}$, which is equivalent to the $\mathfrak{T}_{\mathscr{A}}$-family (8). Accordingly, the customary physical notion of the observation is substituted for a micro-event, an act. "Physics should forget" about processes or time of interaction when observing about the interaction itself and about anything but $\underline{\Psi} \;\text{-}^{\mathscr{A}}\text{→}\; \underline{\alpha}$. This object represents a completed formalization of the empirical/laboratory notion of a quantum micro-event—a detector click. The click is sometimes considered from an information viewpoint as an information bit [22]. However, it cannot be such a (classical) bit with reified content because it is completely unpredictable. The next (different) click does entirely negate the previous one, and the information bit is in turn a concrete thing— the bit. For the same reason, there cannot be any information behind the single event. It is "too small and too momentary" to possess or to carry information about something inasmuch as even the "something" is composed from elementary clicks—see below.

### 2.3. Numerical Realizations

Is there a possibility of relying exclusively on the inflexibility of the eigen-type elements (10) or of defining the sought-for ultimate distinguishability $\not\approx$ through the $\mathscr{A}$-(micro)distinguishabilities $\underline{\alpha}_j \not\approx_{\mathscr{A}} \underline{\alpha}_s$? Let us formulate a thesis.

**T**    There is no (linguistic) means of recognizing the system $\mathcal{S}$ to be different (pt. **S**) other than through the results of its destructions into the $\{\underline{\alpha}_1, \underline{\alpha}_2, \ldots\}$-objects of observational instruments $\mathscr{A}$.

Granted, the stringency of this linguistic taboo (**T**) must be accompanied by something constructive, and we will adopt the following program, which reflects the fact that the unequivocal description may only take the form of a quantitative mathematical theory.

**R** •   Out of the primary ("proto")elements $\{\underline{\Psi}, \underline{\alpha}, \ldots\} \in \mathfrak{T}$, one constructs a new set $\mathbb{H}$, of which the elements

$$|\Xi\rangle := \oplus(\mathfrak{a}_1, |\boldsymbol{\alpha}_1\rangle; \mathfrak{a}_2, |\boldsymbol{\alpha}_2\rangle; \ldots) \in \mathbb{H} \tag{11}$$

are said to be (number) *representations* in the "reference frame for instrument $\mathscr{A}$", and $\mathfrak{a}_s$ are the numerical objects. The distinguishability relation $\not\approx_{\mathscr{A}}$ is carried over to $\mathbb{H}$ and admits an $\mathfrak{a}$-coordinate realization there—symbol $\not\approx$.

••   No preferential or preordained observational reference frame $\mathscr{A}\{\underline{\alpha}_1, \underline{\alpha}_2, \ldots\}$—an absolute instrument—exists.

Identification (11) is always tied to a certain family $\mathfrak{T}_{\mathscr{A}}$. Accordingly, images of $\underline{\alpha}_s$—symbols $|\boldsymbol{\alpha}_s\rangle$—are present in Equation (11), and character $\oplus$ is also no more than a symbol here. Even though coordinates $\mathfrak{a}_s$ are declared to be numbers or aggregates of numbers, there is no arithmetic stipulated for them yet. The number is a name for $\mathfrak{a}_s$. The distinguishability $|\Psi\rangle \not\approx |\tilde{\Psi}\rangle$ of two representatives

$$\oplus(\mathfrak{a}_1, |\boldsymbol{\alpha}_1\rangle; \mathfrak{a}_2, |\boldsymbol{\alpha}_2\rangle; \ldots) =: |\Psi\rangle, \qquad \oplus(\tilde{\mathfrak{a}}_1, |\boldsymbol{\alpha}_1\rangle; \tilde{\mathfrak{a}}_2, |\boldsymbol{\alpha}_2\rangle; \ldots) =: |\tilde{\Psi}\rangle$$

by means of numbers $\mathfrak{a}_k \neq \tilde{\mathfrak{a}}_k$ and mathematical implementation of (11) and of the $\mathbb{H}$-space, i.e., a "coordinatization" scheme have yet to be established. This will comprise the meaning of the word "constructs" (Sections 7–8), which may not be even linked to the mathematical

term mapping yet, since *no math of QM exists at the moment*. It immediately follows that the question about number entities—specifically, about (11)—is nontrivial in physics.

**II** To speak of an exact correspondence between experiment and mathematics (⌈observation + measurement⌉) makes no sense until there is a detailed *mechanism for the emergence* of what is understood by number.

(***The second principium of quantum theory***)

In other words, we wonder what an empiricist/observer understands (semantics) by the word (syntax) "number". The underlying message here implies that the reliance upon the all-too-familiar arithmetic elucidates nothing. *There is no arithmetic in interferometers/colliders*—there are only clicks there—and the empirical nature arising from this construction (along with the measurement) must be scrutinized.

From pts. **T**, **R**, and **II**, it also follows that the search for a description through hidden variables, over which something is averaged, is indistinguishable from the utopian attempts to find out intrinsic content of boxes (5).

*2.4. Macro and Micro*

The task becomes more precise at this point. Instead of nonphysical identity/noncoincidence ($\underline{\Psi} = \underline{\Phi}$ or $\underline{\Psi} \neq \underline{\Phi}$) of two abstract elements $\underline{\Psi}$, $\underline{\Phi}$ of the abstract set $\mathfrak{T}$, we need the concept of a physical $\approx$-equivalence ($\napprox$-distinguishability) of $\mathbb{H}$-representatives $\{|\Psi\rangle, |\Phi\rangle, \ldots\}$. That is, there must hold either relation $|\Psi\rangle \approx |\Phi\rangle$ or its negation $|\Psi\rangle \napprox |\Phi\rangle$ for all $|\Psi\rangle, |\Phi\rangle \in \mathbb{H}$. The primitive set $\mathfrak{T}$, initially required by point **S**, must disappear from the ultimate mathematics of symbols $|\Psi\rangle \in \mathbb{H}$. Therefore, elements $\underline{\Psi} \in \mathfrak{T}$ are henceforth named *primitives*.

Let us sum up the fallaciousness of the metaphysical belief in the meaningfulness of the wording "there is a quantum state", i.e., the belief that the existence of a state has some math-numerical form.

*   There is no a priori way to endow the term (quantum) state of system $\mathcal{S}$ with any meaning ([15], p. 419). It may not have a definition and any predefined semantics. This term should be created. Meanwhile, one cannot get around the concept of the (micro)observation $\mathscr{A}$ [127] (pp. 98–100), [113] (p. 646), [34,96]. Essentially, no one thing, including $\underline{\Psi}$, $\underline{\alpha}$, or the $\mathfrak{T}$-set itself, can be the primary bearer of data about $\mathcal{S}$. "There is an entirely new idea involved, … in terms of which one must proceed to build up an exact mathematical theory" (P. Dirac [26] (p. 12)).

There is no escape from quoting K. Popper: "… language for the theory; … it remains (like every language) to some extent vague and ambiguous. It cannot be made "precise": the meaning of concepts cannot, essentially, be laid down by any definition, whether formal, operational, or ostensive. Any attempt to make the meaning of the conceptual system "precise" by way of definitions must lead to an infinite regress, and to merely *apparent* precision, which is the worst form of imprecision because it is the most deceptive form. (This holds even for pure mathematics.)" ([108], p. 13).

The notions of a physical observable and of its observable values are also ambiguous at this point ([87], p. 5). Their ambiguity is even greater than that of state due to questions such as "what is being measured?" and even 'what is a measurement?'. Nonetheless, up until the end of this section, we will not discard the term state within the context of pt. **S**.

The irreproducibility of outcomes, i.e., the "turnability of $\underline{\Psi}$-primitives into the various", leaves only one option: "to take a look at $\mathcal{S}$ again, once again, …"—in other words, to seek the source of description in repeatability. It is necessary, then, to move to the subject of macro- rather than micro-observation. This intention fits perfectly with the undefined verb "constructs" in pt. **R**, and the following paradigm should be understood as the macro.

**M**　The only way of handling the uncontrollable micro-level changes is the treatment of the results of repeated destructions, accompanied by what we shall call the common physical macro-setting (experimental context):

$$
\begin{array}{c}
\underline{\Psi} \ \cdots \ \underline{\Psi} \quad \underline{\Psi} \ \cdots \ \underline{\Psi} \quad \cdots\cdots \\
\mathscr{A} \ \Big\downarrow \ \downarrow \ \downarrow \quad \downarrow \ \downarrow \ \downarrow \quad \Big\downarrow \mathscr{A} \\
\underline{\alpha}_1 \ \cdots \ \underline{\alpha}_1 \quad \underline{\alpha}_2 \ \cdots \ \underline{\alpha}_2 \quad \cdots\cdots
\end{array}
\quad + \quad \lceil \text{common macro-environment } \mathbf{M} \rceil \ . \tag{12}
$$

To be precise, we should have to (and we shall do) indicate the different $\{\underline{\Psi}, \underline{\Psi}', \dots\}$ here because the same ingoing $\underline{\Psi}$'s in (12) is a preassumption, which we eschew throughout the work. This point will be very fully addressed further below (Sections 2.5, 2.6, and 3.1).

　　The importance of repetitions and distinguishability had long been noted (Bohr, von Neumann et al. [78]), and recently, it was particularly emphasized in the work [128]. The words "copy/repeat…/distin…" occur 90 times therein.

　　Thus, the empiricism of quantum statics forces us to operate exclusively with such formations of copies $\underline{\alpha}, \dots, \underline{\Psi}$, and this is the maximum amount of data provided by the supra-mathematical problem setup. *All* further mathematical structures may come only from constructions such as (12) and from nothing else. Getting ahead of ourselves, let us once again turn our attention to the fact that the implementation of this idea *is not short-length*—"the mathematization process (cor) is not simple" ([58], p. 24), and Sections 3–9 are devoted specifically to this—see, e.g., the chain (105).

　　One can once more repeat (Section 1.3) that much of what follows does not and cannot contain the mathematical definienda and proofs as they are usually present in the literature on quantum foundations. Instead, there appears a step-by-step *inference* of objects as they result themselves: numbers, operations, groups, algebras, etc. The only instrument that may be applicable here is the empirical inference.

　　The common macro-environment **M** in (12) is also viewed as a supra-mathematical notion [106], the mathematical implementation of which is yet to be created. The same considerations regarding qualitative adjectives are applicable as to the physical convention **M** as well as the transition acts in Section 2.1. Representations (11) will be the formalization of the meaning $\lceil \text{observation} \rceil + \lceil \text{data on system } \mathcal{S} \rceil$, but now with no references to the elementary acts in (12). The physical distinguishability criteria $|\boldsymbol{\Psi}\rangle \napprox |\boldsymbol{\Phi}\rangle$ may not be formulated yet because the physical attributes are not yet available, but $|\boldsymbol{\alpha}_s\rangle$-elements have already appeared in (11) as prototypes of explicitly distinguishable $\underline{\alpha}_s$.

**Remark 3.** *The dual form of the typical quantum statements such as "$\mathcal{S}$ is a* micro-*system and $\mathscr{A}$-instrument is a* macro-*object" (N. Bohr) is identical to the initial premise "observation does always destroy a system". It follows that there is actually little need for that terminology. Indeed,* QM-*micro has no internal structure and, hence, an oft-discussed issue about boundary (and limit (According to A. Zeilinger, "… no limit. The limit is only a question of … money and of experience" ([39], 13′09″).)) between micro and macro [8,30,33,90] is devoid of sense; "The notions of 'microscopic' and 'macroscopic' defy precise definition" ([28], p. 215). Therefore, this may be a matter only of "different macro", either "smaller/bigger", i.e., when they describe certain models.*

　　*As a (partially philosophical) note, what is understood by observational randomness does, in fact, boil down to distinguishability, and more specifically, to postulating the micro-chaos (9). In considering the denial of (9) as an impossible proposition, we arrive at the **M**-paradigm and conclude that the only way to deal with that which is contemplated for the subject-matter of a physical description must be the treatment of micro-acts as assemblages ([129], Lect. 6). In other words, and in accordance with the outline of the clicks' analysis set out below, the determinism of micro-processes (micro-ontology)—much less the microscopic time-arrow—is meaningless as a concept since they are not processes but rather structureless acts that have not even any relationship to each other. Since there are no physical phenomena as of yet, the claim that "phenomenon-1 appears to be the cause that precedes phenomenon-2 as the effect" is no more than a collection of words. To attribute physical content and mathematical formulation at the micro-level to them is*

*impossible in principle—the "problem of boxes" noted above. Accordingly, the cause of (classical) macro-indeterminism is the absurdity of the notion of its twin concept—micro-determinism—and the unavoidable repetition of the arrows --→ (**M**). N. Bohr puts the point very definitely: "there can be no question of causality in the ordinary sense of the word" ([78], p. 351), and Heisenberg adds that "l'indeterminismo, . . . ë necessario, e non solo consistentemente possibile" ([17], Section IX.4). See also ([129], p. 223).*

### 2.5. Quantum Ensembles and Statistics

Let us call the upper row in Scheme (12), as a collection of the $\underline{\Psi}$-copies, a (quantum) homogeneous *ensemble* (Kollektiv, by von Mises [129]). We will designate it, simplifying when needed, by

$$\{\underbrace{\underline{\Psi}\,\underline{\Psi}\cdots\underline{\Psi}}_{N \text{ times}}\} \equiv \{\underline{\Psi}\cdots\underline{\Psi}\}_N \equiv \{\underline{\Psi}\}_N \,,$$

where $N$ is understood to be an arbitrary large number. Scheme (12) also dictates considering the generic ensembles

$$\left\{\{\underline{\alpha}_1\cdots\underline{\alpha}_1\}_{n_1}\,\{\underline{\alpha}_2\cdots\underline{\alpha}_2\}_{n_2}\,\cdots\cdots\right\}, \qquad \{\cdots\underline{\Psi}\cdots\underline{\Psi}\,\underline{\Phi}\cdots\underline{\Phi}\,\underline{\Theta}\cdots\underline{\Theta}\cdots\} \tag{13}$$

as collections of homogeneous sub-ensembles. Ensembles are symbolized in the same manner as sets but, for typographical convenience, without the numerous commas and internal parentheses $\{\}$ in Ensemble (13); for example,

$$\{ab\cdots b\{bca\}\cdots\} = \{a,b,\ldots,b,b,c,a,\ldots\} = \quad\cdots\quad =: \{ab\cdots bbca\cdots\}\,.$$

Scheme (12) is the first point in which numbers emerge in theory, and conversion

$$\lceil\underline{\alpha}\text{-ensemble (13)}\rceil \quad\longmapsto\quad (n_1,n_2,\ldots)$$

into the integer collection anticipates a *numerical $\mathscr{A}$-measurement of $\mathcal{S}$*. Quantities $n_s \in \mathbb{Z}^+$, however, should not be associated with such, as they are potentially infinite. The minimal way of creating the knowingly finite numbers out of independent and potential infinities $n_s$ (without loss of their independence) is to divide each of them by a greater infinity, which is a "constant" $\Sigma$ for Ensemble (13). It is clear that one should put

$$\Sigma := n_1 + n_2 + \cdots \quad\text{and}\quad \left\{\mathfrak{f}_1 := \frac{n_1}{\Sigma},\quad \mathfrak{f}_2 := \frac{n_2}{\Sigma},\quad \ldots\right\}\quad(\Sigma \rightsquigarrow \infty)\,, \tag{14}$$

and that Ensemble (13) does not provide any numerical data besides the relative frequencies (14). All the other data are functions of $\mathfrak{f}_s$. An independence of the theory from the ensemble's $\Sigma$-constant, i.e., the scheme $\Sigma \rightsquigarrow \infty$, is also implied to be a principle, and it can only be the semantic one. Without it—the $\Sigma$-postulate of infinity—there can be no question of a rational theory, i.e., empiricism will not turn into a mathematics (Sections 5.1 and 5.2). In turn, the concepts "closely, limit, the limiting frequencies", and the like will arise later when we obtain the state of space as a Hilbert one $\mathbb{H}$ and topology on it [130].

Thus, the **M**-paradigm in Scheme (12) does not only give birth to a concept of numerical data in the theory per se but also converts their $\mathbb{Z}^+$-discreteness into the $\mathbb{R}$-continuum of real measurements. Namely, numbers $\mathfrak{f}_s \in \mathbb{R}$ are the statistics $(\mathfrak{f}_1,\mathfrak{f}_2,\ldots)$ of destructions $\xrightarrow{\mathscr{A}}$ into the ensemble of primitives $\{\{\underline{\alpha}_1\}_{n_1}, \{\underline{\alpha}_2\}_{n_2}, \ldots\}$.

### 2.6. Distinguishability and Numbers

The distinguishability of the two ensembles now turns out to be the $\mathbb{R}$-numerical, i.e., it is determined by the difference between $\mathfrak{f}$-numbers. As a result, and according to pt. **R**, the two elements $|\mathbf{\Psi}\rangle \not\simeq_{\mathscr{A}} |\tilde{\mathbf{\Psi}}\rangle$ of $\mathbb{H}$ will differ in the numbers $\mathfrak{a}_s$ and $\tilde{\mathfrak{a}}_s$ if the latter turn out to be the bearers of different statistics

$$\mathfrak{f}_j(\mathfrak{a}_1,\mathfrak{a}_2,\ldots) \neq \tilde{\mathfrak{f}}_j(\tilde{\mathfrak{a}}_1,\tilde{\mathfrak{a}}_2,\ldots)\,. \tag{15}$$

As a consequence, distinguishability $\not\approx$ is carried over to $\mathbb{H}$ with an extension to the non-eigen objects, but it is inherently incomplete since it does not take into account the most significant fact—arbitrariness of transitions (6).

The collection $(f_1, \ldots)$, as a final result of transitions $\{\underline{\Psi} \dashrightarrow \underline{\alpha}_s\}$, actually "knows nothing" about their left-hand side, much less about its uniqueness $\underline{\Psi}$. For instance, if under the equal $\underline{\alpha}$-statistics $\{f_s\}$ for the two families $\{\underline{?} \dashrightarrow \alpha_s\}_N$ and $\{\underline{\Psi} \dashrightarrow \alpha_s\}_N$ (collectivity of $\underline{?}$'s), we would claim $\underline{?} = \underline{\Psi}$, which would mean a mass control over transitions (9). Instead of a "black box" above, we find that prior to acts $\xrightarrow{\mathscr{A}}$, all the undefined $\underline{?}$'s were equal to $\underline{\Psi}$. This, however, is the declaration of a property: "prior to observation the system $\mathcal{S}$ was/dwelled in ...". With any continuation of this sentence, it is pointless and prohibited if one theoretically accepts that, prior to observation, nothing exists, and there are no properties (Section 2.1). The words "initial state of $\mathcal{S}$" thus make no sense. The indeterminacy of the ongoing $\underline{?}$'s is therefore mandatory, and numbers $(f_1, f_2, \ldots)$ required for recognition are manifestly insufficient. Considering that the micro-changeability of single primitives $\underline{\Psi}$ also means nothing [15] (p. 419 (!), left column), [33] (p. 493), [41], only a generic ensemble

$$\{\underline{?} \cdots \underline{?}\} \quad \rightarrowtail \quad \{\cdots \underline{\Psi} \cdots \underline{\Psi}\, \underline{\Phi} \cdots \underline{\Phi}\, \underline{\Theta} \cdots \underline{\Theta} \cdots\} =: \mathfrak{A} \tag{16}$$

can be an intermediary in the sought-for translation of $\underline{\Psi}$'s onto representations $|\Xi\rangle \in \mathbb{H}$ under Construction (11).

In the accustomed physical terminology, the above is expressed in the sequence

$$\lceil \text{state} \rceil \quad \xdashrightarrow{\mathscr{A}_{\text{quant}}} \quad \lceil \text{state}' \rceil \quad \longmapsto \quad \lceil \text{measurement} \rceil . \tag{17}$$

The removal of the intermediate component here, i.e., switch to the sequence

$$\lceil \text{state} \rceil \quad \xdashrightarrow{\mathscr{A}_{\text{class}}} \quad \cdots\cdots \quad \longmapsto \quad \lceil \text{measurement} \rceil \tag{18}$$

amounts to the rejection of micro-destructibility and of unpredictability. Even with the classical framework, this supposition is questionable since the notion of a "change when observed" disappears. The relationships between the dual concepts—(micro/macro)-scopicity, big/middle/small, etc.—do also get lost. That is the reason why, developing Heisenberg's question "... is it ... I can only find in nature situations which can be described by quantum mechanics?"' ([78], p. 325), we conclude that, strictly speaking,

- *All observations*, regardless of (the envisioned physical) macro/meso/micro characteristics, do have the structure (17), i.e., are quantum. *No non-quantal observations exist.*

With their idealized "roughening", the classical description appends numerical $f$-statistics to (18), which is when the left/right sides of (18) become indistinguishable with respect to the arrow symbols. The arrows may then be replaced with the equivalence

$$\lceil \text{state} \rceil \quad \xleftrightarrow{\mathscr{A}_{\text{class}}} \quad \lceil f\text{-statistics numbers} \rceil . \tag{19}$$

Supplementing the right-hand side here with the concept of numerical values $\{\alpha_s\}$ for all of the observables $\mathscr{A} = A(q, p)$ (or for phase variables $\{q_1, q_2, \ldots; p_1, p_2, \ldots\}$), this side will turn into an exhaustive numerical realization of the left-hand side. Criterion $\approx$, then, turns into the $\mathbb{R}$-number equality $=$ of all the $\mathscr{A}$-statistics or into an equality of phase distributions $\varrho(q_1, \ldots; p_1, \ldots)$. This is a situation of the classical (statistical) physics (ClassPhys), i.e., when "the physics is initially identified" with quantities being numerical in character: the particle coordinates/numbers, the number values of field functions, etc. The ill-posedness of such a paradigm—the core motive of QT—is discussed further below at greater length in Sections 6.4, 6.5, and 7. Consequently, "classicality" is not and cannot be regarded as a primitive in the logical construct. In both these cases, distinguishability $\not\approx$ depends on the concept of $\underline{\alpha}$-states.

**Remark 4.** *From this point onward, by state we will strictly mean representations* (11). *Thus, it makes* no sense to speak of transitions between states, *much less of "transition from possible to actual" ([107], p. 189; Everett), [117–119]. The writing* $|\Xi\rangle \dashrightarrow |\alpha\rangle$ *and its typical wave-function collapse interpretation are not correct. Indeed, in treating transition* $|\Xi\rangle \dashrightarrow |\alpha_1\rangle$ *as a state-to-state destruction, its left-hand side cannot carry any information about* $\mathbf{f}_{(\mathscr{A})}$*-frequencies for other events* $|\Xi\rangle \dashrightarrow |\alpha_s\rangle$, *much less about the amount of destruction from envisioned* $\mathscr{B}$*-observations* $|\Xi\rangle \overset{\mathscr{B}}{\dashrightarrow} |\beta\rangle$. *Such "*$\mathbf{f}_{(\mathscr{B})}$*-amounts" are always present at the experimental interpretation of the* $|\Xi\rangle$*-symbol. For this reason, the concept of a state should not be used as a correct term at all [58]; the terminology, however, has been settled.*

The motivation given above—**S** (system, primitives), **O** (observation), **R** (representations), **T** (taboo), the semantic principia **I** (QM-statics), and **II** (numbers) complemented below with the principium **III**—is sufficient for further creating the basis of the mathematical formalism of QM. These tenets should hardly be regarded as postulates, at least in the common meaning of the phrase "postulates of a physical theory", since they are a natural language and are, as we believe, the points of departure for reasoning whatever the approach to the micro-world. It is clear that they are directly concerned with the familiar dialogs, which reflect, in the words by Bohr, "[Einstein's] feeling of disquietude as regards the apparent lack of firmly laid down principles . . . , in which all could agree" ([131], p. 228).

The underpinning of QT must thus begin, at least to a large extent, with a simplification/reducing the terminology in use and putting the language and the semantics of observations/numbers in order, rather than giving the "improved" postulates or definitions.

"The task is not to make sense of the quantum axioms by heaping more structure, more definitions, . . . , but to throw them away wholesale"

C. Fuchs ([50], p. 989)

"Simplicity is implicit in the basic goals of scientific inquiry. . . . only simple theories can attain a rich explanatory depth. . . . the Basic Propert[ies] should indeed be very simple"

N. Chomsky ([132], pp. 4–5)

As was underscored above, these (organizing) principles do not stipulate for predetermined mathematics and physics, with the exception of a linguistic/metamathematical understanding [105,106] of how to look at the mathematical axioms, structures, rational theories, and their interpretations altogether. See also Remarks 7 and 10 and Sections 5 and 10.

**3. Ensemble Formations**

Your acquaintance with reality grows literally by buds or drops of perceptions. . . . they come totally or not at all—W. James (1911)

Are billions upon billions of acts of observer- participancy the foundation of everything?—J. Wheeler ([62], p. 199)

The key corollary of Macro-paradigm (12) is not merely the appearance of numerical data in the theory but also the fact that the further construct cannot rely on isolated primitives but rather on their aggregates being considered as an integrated whole, i.e., as a set. This causes a choice for the ensemble notation.

*3.1. Mixtures of Ensembles*

Returning to the analysis of transitions $\underline{?} \dashrightarrow \alpha_s$, one obtains that the lower row in (12) actually comes from indeterminacy

$$\{\underline{\alpha_1}\}_{n_1}\{\underline{\alpha_2}\}_{n_2}\cdots\cdots$$
$$\uparrow \quad \uparrow \quad \uparrow \quad \uparrow \quad \uparrow$$
$$\underline{?}\cdots\underline{?}\cdots\underline{?}\cdots\cdots$$

and thus (12), by virtue of (16), should be replaced with the scheme

$$
\begin{array}{c}
\{\cdots \underline{\Psi} \cdots \underline{\Phi} \cdots \underline{\Theta} \cdots\} \\
\mathscr{A} \downarrow \; \downarrow \; \downarrow \; \downarrow \; \downarrow \; \mathscr{A} \\
\{\cdots \underline{\alpha}_1 \cdots \underline{\alpha}_2 \cdots\}
\end{array} \quad , \tag{20}
$$

wherein the composition of the upper ingoing row may not be predetermined. Fundamentally, according to (17), it may not be withdrawn from (20), yet at the same time, the meaning of the row can in no way be aligned with the adjective "observable" via typical empirical/physical words: properties, readings, quantities/amounts, and other "observable" characteristics. Such non-detectability is the equivalent of a box that may be prepended to Scheme (6):

$$
\boxed{\cdots\cdots} \; \text{-}\rightsquigarrow \; \underline{\Psi} \; \text{-}\xrightarrow{\mathscr{A}} \; \underline{\alpha} \; . \tag{21}
$$

If $\underline{\beta}$'s serve as $\underline{\Psi}$ in (21), then we have the schemes of precedence and of continuation:

$$
\boxed{\cdots\cdots} \; \text{-}\xrightarrow{\mathscr{A}} \; \underline{\alpha} \; \text{-}\xrightarrow{\mathscr{B}} \; \underline{\beta} \qquad \text{or} \qquad \boxed{\cdots\cdots} \; \text{-}\xrightarrow{\mathscr{B}} \; \underline{\beta} \; \text{-}\xrightarrow{\mathscr{A}} \; \underline{\alpha} \; .
$$

Let an observer capture the fact of any distinguishability in the penultimate $\mathscr{A}$. Section 2.1 tells us that this may only be the distinguishability of objects $\{\underline{\alpha}_1, \underline{\alpha}_2, \ldots\}$; hence, this very $\mathscr{A}$ turns into an observation (pt. **O**). The **M**-paradigm then gives rise to the numbers of $\underline{\alpha}$-events $(n_1, n_2, \ldots)$ and, thereupon, their relative frequencies $(\varrho_1, \varrho_2, \ldots)$ by the rule (14). If subsequently micro-observations $\mathscr{B}$ are to follow, then a composite macro-observation $\mathscr{B} \circ \mathscr{A}$ has been formed, and frequencies $\{\varrho_j\}$ cannot impact statistical characteristics of these later $\mathscr{B}$-observation's micro-events. However, being an ongoing ensemble for $\mathscr{B}$, each homogeneous $\{\underline{\alpha}_s \cdots \underline{\alpha}_s\}_{n_s}$ is indistinguishable from an indefinite ensemble $\{\cdots \underline{\Psi} \cdots \underline{\Phi} \cdots\}_{n_s}$ since the concept of "$\approx_{\mathscr{A}}$-sameness" is unknown for $\mathscr{B}$. Instrument $\mathscr{B}$ is "aware of only its own $\approx_{\mathscr{B}}$ and cannot know *what* it destroys", or that the source-object consists of one and the same $\underline{\alpha}_s$. Rejecting this point brings us once again (p. 9) to attempts at "penetrating the black box" of transitions (5), i.e., to attempts at creating the physics of a more primary level. According to pts. **O** and **M**, an instrument produces nothing more than its own "destruction list"; in this case, $(\{\underline{\beta}_1\}_{m_1}, \{\underline{\beta}_2\}_{m_2}, \ldots)$. This list is completely independent of the preceding one since, according to pt. **R$^{\bullet\bullet}$**, there cannot be restrictions on $\mathfrak{T}_{\mathscr{A}}$ and $\mathfrak{T}_{\mathscr{B}}$. In case the set $\{\underline{\alpha}_s \cdots \underline{\alpha}_s\}_{n_s}$ transits into collection $\{\underline{\beta}_k \cdots \underline{\beta}_k\}_{n_s}$, this means that $\underline{\alpha}_s$ has always transited into one and the same $\underline{\beta}_k$ every time (under the convention $\Sigma \rightsquigarrow \infty$), and merely a coincidence $\underline{\alpha}_s = \underline{\beta}_k$ of eigen-primitives in the lists $\mathfrak{T}_{\mathscr{A}}$ and $\mathfrak{T}_{\mathscr{B}}$ takes place.

If $\mathscr{B} \circ \mathscr{A}$ is proceeded with a third observation $\mathscr{C}$, the preceding analysis is repeated recursively with the same result; only the values $\{\varrho_j\}$ will be changed. As a consequence, only the following two ongoing types for macro-scheme (20) are conceivable:

$$
\{\cdots \underline{\Psi} \cdots \underline{\Phi} \cdots \underline{\Theta} \cdots\} \qquad \begin{array}{l} \text{indefinite ensemble} \\ \lceil \text{no statistics} \rceil \end{array}, \tag{22}
$$

$$
\{\{\cdots \underline{\Psi} \cdots \underline{\Phi} \cdots \underline{\Theta} \cdots\}^{(\varrho_1)} \{\cdots \underline{\Psi} \cdots \underline{\Phi} \cdots \underline{\Theta} \cdots\}^{(\varrho_2)} \cdots\} \qquad \begin{array}{l} \text{ensemble mixture} \\ \lceil \text{with statistics } (\varrho_1, \varrho_2, \ldots) \rceil \end{array}. \tag{23}
$$

It is reasonable to regard Case (23) as a "non-interfering" mixture of the system's $\mathscr{A}$-preparations

$$
\{\mathcal{S}^{(\varrho_1)}, \mathcal{S}^{(\varrho_2)}, \ldots\} \quad \Longleftrightarrow \quad \{\underline{\alpha}_1^{(\varrho_1)}, \underline{\alpha}_2^{(\varrho_2)}, \ldots\},
$$

to each of which one assigns the positive number $\varrho_s < 1$ referred to as its statistical weight. These weights—"an element of reality" ([113], p. 649)—are all that is inherited from the preparation $\mathscr{A}$, and subsequent micro-observation acts $\mathscr{B}$ are performed again on indefinite ensembles (22).

It is clear that in the view of transitions $\dashrightarrow$ in scheme (20), this situation is a derivative of (22) and this very type (22) is crucial ([34], p. 53). In other words, if the preparation is regarded as a concept as essential as observation (pt. **O**), we still remain within the framework of the binate essence of the transition:

$$\underline{\Psi} \; \overset{\mathscr{B}}{\dashrightarrow} \; \underline{\beta}, \qquad \underline{\alpha} \; \overset{\mathscr{B}}{\dashrightarrow} \; \underline{\beta}\,.$$

Its left-hand side should always be seen as an undetermined primitive, even though we treat/call it the preparatory (micro)observation. See also "preparation-measurement reciprocity" in [133].

*3.2. Ensemble Brace*

According to pts. **R** and **M**, the representations in (11) must reflect all information about the physics of the problem: primitives/incomes, transitions ("arrows" $\dashrightarrow$), and outgoing statistics. All the data are contained in Scheme (20), which is why the maximum that the model of a future mathematical object—it characterizes everything we obtain while watching the $\mathcal{S}$—can rely on is the *ensemble brace*:

$$(\underline{\Xi}) := \begin{pmatrix} \{\cdots\underline{\Psi}\cdots\underline{\Phi}\cdots\underline{\Theta}\cdots\} \\ \mathscr{A}\;\Big\downarrow\;\downarrow\;\downarrow\;\downarrow\;\downarrow\;\mathscr{A} \\ \{\cdots\underline{\alpha}_1\cdots\underline{\alpha}_2\cdots\} \end{pmatrix} \tag{24}$$

(or a couple of ensemble bunches).

It is immediately seen that (24) carries the radical difference between situation (17) and its "roughening" (19) because of the upper row. The enormous arbitrariness within the brace and arrows $\overset{\mathscr{A}}{\dashrightarrow}$ is "programmed" to give birth to the different processing rules of statistics and to effects that are typical for QM. Thanks to the maximality of (24), it is only this row that encodes all the sought-for cases of distinguishability $\not\approx$. In particular, by varying the upper row while the lower one remains unchanged, we get into a situation when $\underline{\alpha}$-statistics $(f_1, f_2, \ldots)$ are found to be the same for $(\underline{\Xi})$ and $(\tilde{\underline{\Xi}})$, and meanwhile, $(\underline{\Xi}) \not\approx (\tilde{\underline{\Xi}})$.

The problem is thus as follows. With the indefinite $\mathfrak{A}$-ensemble (16) in hand, i.e., with the upper row of (24), is it possible, based on the principles described above, to bring the still incomplete relation $\not\approx$ to the maximal quantum-physical distinguishability of states?

## 4. Why Does Domain $\mathbb{C}$ Come into Being?

… quod ideo sint imaginariae, … quod ideo sint …tum certe forent reales ideoque non imaginariae—L. Euler (1736)

(… this is why they are imaginary. Were they …, they would certainly be real and therefore not imaginary.)

… denn die imaginären Größen existierten doch nicht?—D. Hilbert (1926)

The first priority in the $\not\approx$-distinguishability of objects (24) is to separate the closest and unconditional criterion—the outgoing $\underline{\alpha}$-statistics. To do this, let us split the lower row into families $\big\{\{\underline{\alpha}_1\}_{\infty_1}\{\underline{\alpha}_2\}_{\infty_2}\cdots\big\}$, where

$$\infty_1 + \infty_2 + \cdots = \infty\,, \tag{25}$$

and, subsequently (rather than the reverse, otherwise (23)), taking into account the "arbitrariness of arrows", we also split the upper row:

$$(\underline{\Xi}) = \begin{pmatrix} \big\{\{\cdots\underline{\Psi}\cdots\underline{\Phi}\cdots\underline{\Theta}\cdots\}_{\infty_1} & \{\cdots\underline{\Psi}\cdots\underline{\Phi}\cdots\underline{\Theta}\cdots\}_{\infty_2}\cdot\big\} \\ \downarrow\;\downarrow\;\downarrow\;\downarrow & \downarrow\;\downarrow\;\downarrow\;\downarrow \\ \big\{\;\{\underline{\alpha}_1\cdots\cdots\underline{\alpha}_1\}_{\infty_1} & \{\underline{\alpha}_2\cdots\cdots\underline{\alpha}_2\}_{\infty_2}\;\cdots\big\} \end{pmatrix} \tag{26}$$

(the indication of observation $\xrightarrow{\mathscr{A}}$ is omitted further below since it has been mirrored in primitives $\underline{\alpha}$). Hereafter, infinities $\infty_j$ stand for cardinal numbers (a number of elements, possibly finite) of their own ensembles. Therefore, the extension of distinguishability (15) should be produced by comparing the sub-objects such as

$$
\begin{array}{c}
\{\cdots\underline{\Psi}\cdots\underline{\Phi}\cdots\underline{\Theta}\cdots\}_{\infty_1} \\
\downarrow \quad \downarrow \quad \downarrow \quad \downarrow \\
\{\underline{\alpha}_1\cdots\cdots\underline{\alpha}_1\}_{\infty_1}
\end{array} \quad , \tag{27}
$$

that differ from each other in the upper-row composition.

### 4.1. Continuum of Quantum Phases

The cardinality of the $\mathfrak{T}$-set cannot be finite. This would finitely entail many $\underline{\alpha}$-primitives for all kinds of instruments. However, the finiteness of this number $n_{\mathfrak{T}}$ would mean an exclusivity of its value that does not follow from anywhere. At the same time, all the $\mathfrak{A}$-ensembles (16) are subsets of the set $\mathfrak{T}$ (boolean $2^{\mathfrak{T}}$); any finite portion of it is ruled out. Hence, the endless variety of upper rows in (27) is uncountable.

Aside from the number of $\mathfrak{f}$-statistics, program **R** does also require an association of the numerical objects with each row

$$
\mathfrak{A} = \{\cdots\underline{\Psi}\cdots\underline{\Psi}\,\underline{\Phi}\cdots\underline{\Phi}\,\underline{\Theta}\cdots\underline{\Theta}\cdots\}_{\infty} \quad \Longleftrightarrow \quad \cdots ,
$$

because primitive's symbols must disappear in the ultimate description. To avoid introducing the structures ad hoc, we will produce numbers here—the upper row—in the same manner, in which statistics were producing in Section 2.5—the lower row. Indeed, the genesis of the concept of the number must be single in theory. That is, we should again take into account the presence of copies of primitives and write

$$
\cdots \quad \Longleftrightarrow \quad \Big\{ \underbrace{\{\underline{\Psi}'\}_{\infty'}\{\underline{\Psi}''\}_{\infty''}\cdots}_{K\text{ times}} \Big\} , \tag{28}
$$

and numbers per se will come into being by the $\Sigma$-convention, such as (14), i.e., through the cardinal ratios

$$
\varkappa' := \frac{\infty'}{\infty}, \qquad \varkappa'' := \frac{\infty''}{\infty}, \qquad \ldots . \tag{29}
$$

Now, the discreteness of micro-transition acts is embodied in (28) with the sequence $(\underline{\Psi}', \underline{\Psi}'', \ldots)$, and the uncountability of micro-arbitrariness is inherited by attaching the symbolic "quantities"—"countless" characters $(\infty', \infty'', \ldots)$—to elements of this sequence. The global discreteness says that there are no grounds to assume a more than countable infinity $\aleph_\circ$ for the set $\mathfrak{T}$, i.e., $|\mathfrak{T}| = \aleph_\circ$. The infinity of the family (28), hence, has the type

$$
2^{\aleph_\circ} = \aleph ,
$$

i.e., it is continual [134]. Parenthetically, the $2^{\aleph_\circ}$ is the only known way of introducing the continual (more than discrete) mathematical infinity. Which possibilities exist for the form of row (28)?

The trivial case $\mathfrak{A} = \big\{\{\underline{\Psi}'\}_{\infty'}\big\}$, i.e., $K = 1$ in (28) drops out at once since element $\underline{\Psi}'$ would always go into the same primitive:

$$
\begin{array}{cc}
\begin{array}{c}
\{\underline{\Psi}'\cdots\cdots\underline{\Psi}'\}_{\infty_1} \\
\downarrow \quad \downarrow \quad \downarrow \quad \downarrow \\
\{\underline{\alpha}_1\cdots\cdots\underline{\alpha}_1\}_{\infty_1}
\end{array}
&
\begin{array}{c}
\{\underline{\Psi}'\}_{\infty_1} \\
\mathscr{A}\downarrow\cdots\downarrow\mathscr{A} \\
\{\underline{\alpha}_1\}_{\infty_1}
\end{array}
\end{array} \quad . \tag{30}
$$

However, this is tantamount to the identity $\underline{\Psi}' \equiv \underline{\alpha}_1$, which robs of any meaning the concept of the transition $\underline{\Psi} \xrightarrow{\mathscr{A}} \underline{\alpha}$. We obtain a single number here—the number of $\underline{\alpha}_1$-clicks—and arrive thereby at classical statistics, the physics of which is inadequate with

respect to the interference patterns. Hence, the following options are admissible for the formations (28):

$$\left\{ \underbrace{ \{\underline{\Psi}'\}_{\infty'} \{\underline{\Psi}''\}_{\infty''} }_{K=2} \right\}, \quad \cdots, \quad \left\{ \underbrace{ \{\underline{\Psi}'\}_{\infty'} \{\underline{\Psi}''\}_{\infty''} \{\underline{\Psi}'''\}_{\infty'''} \cdots }_{3 \leqslant K < \infty} \right\}, \quad \cdots$$

$$\cdots, \quad \left\{ \underbrace{ \{\underline{\Psi}'\}_{\infty'} \{\underline{\Psi}''\}_{\infty''} \{\underline{\Psi}'''\}_{\infty'''} \cdots }_{K=\infty} \right\} \quad (31)$$

with minimal $K = 2$. If some of the infinities $(\infty', \infty'', \dots)$ are finite here or countable, this does not change the total continuality $\aleph$. The extreme case $K = \infty$—a countable infinity of continuums—also changes this count because of $\aleph + \aleph + \cdots = \aleph$ [135]. All of these infinities may be even countably duplicated without augmenting the continuum since $\aleph \cdot \aleph \cdots = \aleph^{\aleph_\circ} = \aleph$.

What can one say about relationship of cases (31) between each other? Do we have to deal with their total arbitrariness or with only one of these schemes? The latter case—the sameness/indistinguishability of upper rows in (27)—would correspond to the structural staticity of theory. Otherwise, whether one (unrecognizable upper) row should differ (why?) from another in the number (what?) of defining primitives $\{\underline{\Psi}', \underline{\Psi}'', \dots\}$ (which ones?)?

Suppose the variability of $K$. That is, consider the simultaneous existence of, say, the $K = \{2, 3\}$ rows

$$\left\{ \{\underline{\Psi}'\}_{\infty'} \{\underline{\Psi}''\}_{\infty''} \right\}, \qquad \left\{ \{\underline{\Psi}'\}_{\infty'} \{\underline{\Psi}''\}_{\infty''} \{\underline{\Psi}'''\}_{\infty'''} \right\}.$$

However, each of the 2-row is a particular case of the 3-row with a cardinal number $\infty''' = 0$:

$$\left\{ \{\underline{\Psi}'\}_{\infty'} \{\underline{\Psi}''\}_{\infty''} \right\} = \left\{ \{\underline{\Psi}'\}_{\infty'} \{\underline{\Psi}''\}_{\infty''} \{\underline{\Psi}'''\}_{\{\infty'''=0\}} \right\} \subset \left\{ \{\underline{\Psi}'\}_{\infty'} \{\underline{\Psi}''\}_{\infty''} \{\underline{\Psi}'''\}_{\infty'''} \right\}$$

(the case in point is sets). Therefore, these situations are structurally indistinguishable from each other, and the $K = 2$ theory is a *sub*theory for $K = 3$. So, the cases $K = \{2, 3\}$ are actually not mutually exclusive; rather, they form an embedding. We thus have arrived at the one cumbersome and common construct akin to the Russian dolls $2 \supset 3 \supset 4 \supset \cdots$. Hence, the minimal 2-theory will always be present inside all the higher orders $K > 2$ as an "independent (sub)world". For this reason, the $K = 2$ theory must be created in any way; incidentally, it will enclose the $K = 1$ case.

In the other part, we have no criteria to terminate the sequence $2 \rightarrowtail 3 \rightarrowtail \cdots$ at some intermediate $K < \infty$. Such a cut-off does immediately raise an issue of the questionable empirical exclusivity of a certain "world integer $K \geqslant 3$" that defines the number of "physically inaccessible" $\underline{\Psi}$-objects. Moreover, these options would be related to a certain topological dimension $K \geqslant 3$ that has an unmotivated origin. We thus conclude that the non-minimal options $K = 3$, $K = 4$, ... in (31) should be dismissed.

**Remark 5.** *A few remarks can be made in connection with the case when $K = \infty$. It is related to a conglomerate of infinities, which has the form of a discretely infinite family of continual infinities $\{\varkappa',\varkappa'', \dots\}$, and things would have been even "worse" had the staticity of the schemes (31) been changed to variability. Such formations would need to be equipped with topology and with associated concepts of convergence and of limit. However, all this touches on principally unobservable numerical entities, for which it is not clear how to motivate the further reductions to "finite mathematics" as required: dimensions, finite approximations, finite numbers (which ones?), and the like. More to the point, all of that would pertain to the global structural parameters of the theory prior to constructing it per se, not to mention the physical models. To put it plainly, such an assumption would result not in a theory but in a theory of theories, and so on ad infinitum, which should be somewhere terminated in some way. For these reasons, we leave the case $K = \infty$ aside, though it might be worth elaborating on it. However, in Section 7.6, we will give a further justification that the number domain of the theory is what it has already been known in QM.*

As a result, one has a choice: the structural staticity $K = 2$ or entirely non-structured/ undetermined set of outgoing primitives $\{\underline{\Psi}_j, \underline{\Psi}_k, \ldots\}$, i.e., extremely complex case $K = \infty$. We do choose $K = 2$. This option might have been adopted even before on the ground that the most minimalistic construction, which set-theoretically gives rise, as a minimum, to the minimal numerical object—a single number—corresponds to the minimal $K = 2$ in (31). The maximal case is problematic, while the mid-ones are ruled out. That is to say, all possible assumptions regarding the upper row structure in (27) are indistinguishable from a case just as if the row contained two primitives only $\{\underline{\Psi}', \underline{\Psi}''\} =: \{\underline{\Psi}, \underline{\Phi}\}$. The functionality of the symbol $\cup$, with regards to the inclusion of the $\{\underline{\Psi}, \underline{\Phi}\}$'s copies, is unchanged (see Section 5.1 further below).

We establish in the following writing of Scheme (27) that

$$\{\underline{\Psi} \cdots \underline{\Psi}\}_{\infty_1'} \cup \{\underline{\Phi} \cdots \underline{\Phi}\}_{\infty_1''} \quad \overset{\text{---}}{\overset{\text{---}}{\longrightarrow}} \quad \{\underline{\alpha}_1 \cdots \underline{\alpha}_1\}_{\infty_1}$$

none of the primitives $\{\underline{\Psi}, \underline{\Phi}\}$ coincide with $\underline{\alpha}_1$. Otherwise, the unrestricted adjunction of identical transitions $\underline{\alpha}_1 \dashrightarrow \underline{\alpha}_1$ to (27) would mean indeterminacy of both the number $\varkappa_1$ and the actual statistics $(f_1, f_2, \ldots)$.

Let us take into account that numbers (29) are mathematically generated by the standard scheme: $\lceil$(ordered) integers$\rceil \rightarrowtail \lceil$(ordered) rationals$\rceil \rightarrowtail \lceil$(ordered) continuum $\rceil$. The natural ordering $<$ is always present here and, as is well known ([136], p. 52), can be isomorphically represented by the set-theoretic inclusion $\subset$ on a certain system of sets. That inclusion (= "to be contained in"), in turn, is directly concerned with the semantics of Section 2. The natural-language term "accumulating"—"the old is being nested into the new"—is formalized to create sets by the cumulative ensembles (see Section 5.1).

We now conclude that all kinds of schemes (27) form an $\aleph$-continuum, for which there is no reasonable rationale for equipping it with a topology other than the standard order topology of the one-dimensional real $\mathbb{R}$-axis or its equivalents. Call the quantity $\varkappa \in \mathbb{R}$ *quantum phase*.

It should be added that in considering some two upper rows in (27) as infinite sets

$$\{\underline{\Psi} \cdots \underline{\Psi}\}_{\infty'} \cup \{\underline{\Phi} \cdots \underline{\Phi}\}_{\infty''} \quad \text{and} \quad \{\underline{\Psi} \cdots \underline{\Psi}\}_{\widetilde{\infty}'} \cup \{\underline{\Phi} \cdots \underline{\Phi}\}_{\widetilde{\infty}''} \, ,$$

one can always establish their formal identity. However, physics requires distinguishing the rows, which is what the numerical part of pt. **R** and comparison of cardinals $(\infty', \infty'')$ do "serve".

### 4.2. Statistics + Phases

Thus, the closest reconciliation of Scheme (26) with the **R**•-postulate is an ensemble brace of the form

$$(\underline{\Xi}) = \begin{pmatrix} \{\{\underline{\Psi}\}_{\infty_1'} \{\underline{\Phi}\}_{\infty_1''}\} & \{\{\underline{\Psi}\}_{\infty_2'} \{\underline{\Phi}\}_{\infty_2''}\} & \cdots\cdots \\ \downarrow \ \downarrow \ \downarrow \ \downarrow & \downarrow \ \downarrow \ \downarrow \ \downarrow & \\ \{\underline{\alpha}_1 \cdots\cdots \underline{\alpha}_1\}_{\infty_1} & \{\underline{\alpha}_2 \cdots\cdots \underline{\alpha}_2\}_{\infty_2} & \cdots\cdots \end{pmatrix} \tag{32}$$

followed by the (upper) continual numeration through $\mathbb{R}$-numbers

$$\varkappa_s := \frac{\infty_s'}{\infty_s} \qquad (\infty_s := \infty_s' + \infty_s'') \, . \tag{33}$$

In other words, the quantitative description in the theory is created on the basis of the minimal building bricks

$$\begin{pmatrix} \{\{\underline{\Psi}\}_{\infty'} \{\underline{\Phi}\}_{\infty''}\} \\ \downarrow \ \downarrow \ \downarrow \ \downarrow \\ \{\underline{\alpha} \cdots\cdots \underline{\alpha}\}_{\infty} \end{pmatrix} \qquad \textit{(unitary brace)} \tag{34}$$

with *two* abstract ongoing primitives.

Now, we have had cardinals connected by Relation (25) and Structures (32) and (33). In the above-described context, parentheses { } and symbols $\underline{\Psi}$, $\underline{\Phi}$, $--\rightarrow$ no longer carry meaning at this point. Therefore, we may omit them as "extraneous" and write (32) as

$$(\underline{\Xi}) \quad \Longleftrightarrow \quad \begin{pmatrix} \varkappa_1 & \varkappa_2 & \cdots \\ \infty_1 & \infty_2 & \cdots \end{pmatrix} = \cdots,$$

where $\underline{\alpha}_s$ are well represented by a subscripted numerals; observation $\mathscr{A}$ has been fixed so far. Let us now introduce a statistics from the "embracing infinity" (25):

$$\cdots = \begin{pmatrix} \varkappa_1 & \varkappa_2 & \cdots \\ f_1 \cdot \infty & f_2 \cdot \infty & \cdots \end{pmatrix} = \begin{pmatrix} \varkappa_1 & \varkappa_2 & \cdots \\ f_1 & f_2 & \cdots \end{pmatrix} \cdot \infty', \qquad f_s := \frac{\infty_s}{\infty}.$$

Then, by $\Sigma$-postulate, one arrives at a continually numeral labeling of objects (32):

$$(\underline{\Xi}) \quad \Longleftrightarrow \quad \left\{ \begin{pmatrix} \varkappa_1 \\ f_1 \end{pmatrix}, \begin{pmatrix} \varkappa_2 \\ f_2 \end{pmatrix}, \ldots \right\}.$$

Recall that the arithmetical operations on the emergent pairs $(f, \varkappa)$ are still out of the question, and $\Sigma$-limit does not care the "innards" of $(\underline{\Xi})$. Only one of all the potentially infinite quantities tends to the $\infty$-infinity—the total cardinality (25) of brace (32). What remains "non-extraneous" in (32) is $\underline{\alpha}$'s, and we return them to their place. Hence, from the viewpoint of observation $\mathscr{A}$, the aggregate of the possible brace (24) is indistinguishable from an order-indifferent two-parametric family of data

$$(\Xi) = \left\{ \begin{pmatrix} \varkappa_1 \\ f_1 \end{pmatrix}\underline{\alpha}_1, \begin{pmatrix} \varkappa_2 \\ f_2 \end{pmatrix}\underline{\alpha}_2, \ldots \right\}. \tag{35}$$

We drop a lower bar in the symbolic designation $(\underline{\Xi})$, highlighting the fact that the meaning of the $(\Xi)$-object becomes increasingly divorced from primitives in pt. **S** and gets into the number domain to match program **R**.

As an outcome, despite the freedom of ingoing collection in (26) and quantum micro-arbitrariness, the distinguishability $(\Xi) \not\approx (\tilde{\Xi})$ is *indeed determinable*, it is determinable not only by statistics, and is the $(\mathbb{R} \times \mathbb{R})$-numerical:

$$(\Xi) \not\approx (\tilde{\Xi}), \quad \text{if} \quad (f_s, \varkappa_s) \neq (\tilde{f}_s, \tilde{\varkappa}_s). \tag{36}$$

What is more, the preliminary (classical) $\not\approx$-criterium (15) fits in (35)–(36) as a particular case by omitting the $\varkappa$-numbers and middle link from (17). That is to say, ignoring quantum "$\varkappa$-effects" is only possible via the $(3 \mapsto 2)$ reduction of (17) into (18), with an automatic imposition of the ClassPhys description. A simplified and hypothetical version of QM over $\mathbb{R}^1$ is also ruled out. It would mean a reduction in the two numbers $(f, \varkappa)$ to a single one. However, they have fundamentally different origins. The construct and reasoning in Section 2.1 also tell us that the attempt at a greater "quantum specification" to (5) and (17) is impossible by virtue of the two-row structure—ingoing/outgoing—of the object $(\Xi)$, and distinguishability by numerical pairs (36) is the highest possible.

The $(\Xi)$-objects (35) remain, and they, as a family, exhaustively inherit the problem's physics. The quantities $f_s$ are the really observable (unitless) numbers—the percentage quantity of events—which are declared by instrument/observer to be the distinguishable $\underline{\alpha}$-objects. The quantities $\varkappa_s$ are the internal and unremovable degrees of freedom. Figuratively speaking, the $\varkappa$'s may be speculatively referred to as phases, but they may not be associated with an actual quantity of something. Not only is any material or the classical treatment of these "amounts" impossible, but it is fundamentally *prohibited* since the converse would have meant endowing the nonexistent boxes (5) and (6) with a notional content or asserting the nature of their origin. Justification is only allowed here for the fact of their existence,

which is mirrored by the presence of the left-hand side in the concept of the transition of $\underline{\Psi} \dashrightarrow^{\mathscr{A}} \underline{\alpha}$ (Remark 2).

In view of numerous ongoing discussions of the meaning to the quantum state [21], note that, for the same reason, any (even merely similar) classical/ontological and causal "visualization mechanisms" ([5], p. 137) as the wave function of a certain real matter, of a hypothetical observable, of an "objective knowledge', or of the classical data (whatever this all means) are—and this we stress with emphasis—pointless. This is why, strictly speaking, without further theoretical conventions,

- It is impossible ([12], p. 13) to make/prepare, observe/read-off, transmit or measure/approximate a state, or to endow it with the property of being known/unknown, or physically recognize/compare/distinguish it from the other.

We will be repeatedly turning back to this matter in Sections 6.3.1, 6.4, 6.5, and 10.2. The present thesis has not undergone a change even with regard to the word "statistics" in the Born rule [6], if only because the rule is a substantial—two-to-one—reduction in the $(\mathtt{f}, \varkappa)$-data. The state will itself, when created as a mathematical object, determine the meaning of all of these words (see Section 5.3) with an appropriate concept of the physical distinguishability (Section 2.4). Cf. the works [53,54] and the "methods to directly measure general quantum states . . . by weak measurements" in [137] and, on the other hand, the statements in Section 15.5 of the book [33].

All the $\varkappa_s$ and $\mathtt{f}_s$ are independent of each other, except for relation $\mathtt{f}_1 + \mathtt{f}_2 + \cdots = 1$. Taking into account the admissible renormalization of both $\mathbb{R}$-numbers, the pair $(\mathtt{f}, \varkappa)$ can be topologically identified with a point on the complex plane:

$$\begin{pmatrix} \varkappa \\ \mathtt{f} \end{pmatrix} \rightleftarrows (\lambda, \mu) \in \mathbb{R}^2 =: \mathbb{C} \,.$$

That is, the domain $\mathbb{C}$ is at the moment just a two-dimensional numeric continuum without algebra of complex numbers. Notice that the pairs of $\mathbb{R}$-numbers is a starting point—different from ours—to the QT in ref. [138]. More than that, the impossibility of the real-number QM became a subject of the direct experimental test to distinguish between the complex-number and real-number representations of QT: on photonic systems [139] and the superconducting qubits [140].

The issue of the numerical domain over which the quantum description is being conducted—the real $\mathbb{R}$, the complex $\mathbb{C}$, the quaternions Q, or whatever—is non-trivial and continues to be the subject of study [57,93,138,141,142]. The complexity $\mathbb{C}$ is often motivated by quantum dynamics (Schrödinger's equation) ([36], p. 132; Stueckelberg), [143]; however, such a motivation is inconsistent, and as we have seen, there is no need for it. The rigidity of the $\mathbb{C}$-domain points to the fact that, in particular, the quaternion QM also has no place to originate from ([33], Section 10.1), although it was the object of theoretical constructs in the 1960–1970s [144]. Note that even the most comprehensive works [36] (p. 131), [72] (p. 234), [93] (p. 217), [96,138], and [142] (!) observe a difficulty in the full substantiation of the $\mathbb{C}$-domain in QT. Within the last decade, this theme had also attracted the particular attention of the information-theoretic approach to QT [138,145,146].

The above-outlined emergence of the numerical quantities in theory is a draft at the moment and will be refined further below in Section 7.

## 5. Empiricism and Mathematics

Set theory does not seem today to have . . . organic interrelationship with physics—
P. Cohen and R. Hersh ([147], p. 116)

. . . physics has . . . to say about the foundations of mathematics . . . "if we believe in ZF there is nothing for physics to say" is not right—P. Benioff ([2], p. 31)

Up to this point, we have dealt, roughly speaking, with a single abstract aggregate $(\underline{\Xi})$ isolated from the others. However, the constructional nature of the ensemble brace (32) entails the following closedness relation between them. Every brace $(\underline{\Xi})$ is composed of

some others in infinitely many ways (for remote analogies, see ([40], Section 11.2)), i.e., it is a union

$$(\Xi) = (\Xi') \cup (\Xi'') \,, \tag{37}$$

and, to put it in reverse, any union of two braces is a third object-brace. In assemblages (37), the operation $\cup$, which generates them, is commutative and associative:

$$A \cup B = B \cup A, \qquad (A \cup B) \cup C = A \cup (B \cup C) \,, \tag{38}$$

and these two- and three-term relations not only are not a formal supplement, but should be read as the *structural* properties in general. Let us address the matter more closely.

*5.1. Union of Ensembles*

Consider the lower $\underline{\alpha}$-rows of brace (26) and experimentally forming the new real $\underline{\alpha}$-ensembles from them. Let the procedure of that forming be denoted by $\mathsf{U}(A, B, \ldots)$, where $(A, B, \ldots)$ are the ensembles per se. Its essence is such that it is comprehensively determined by the following minimum. A rule that involves the *fewest* (i.e., two) number of arguments $\mathsf{U}(A, B) = ?$ and a rule of the *repeated* applying $\mathsf{U}$ to itself: $\mathsf{U}(\mathsf{U}(\ldots), \ldots) = ?$. Obviously, we should write

$$\mathsf{U}(A, B) = \mathsf{U}(B, A), \qquad \mathsf{U}\big(\mathsf{U}(A, B), C\big) = \mathsf{U}\big(A, \mathsf{U}(B, C)\big) \,, \tag{39}$$

which is of course merely the empirical rephrasing the standard properties (38) of operation $\cup$. However, the converse is logically preferable: Empiricism (39) is formalized into the abstract properties (38). If we now attach the upper "quantum" primitives to the low $\underline{\alpha}$-rows—a requirement of Section 2.1—then the operationality of actions with the resulting $(\Xi)$-braces would be just like that of $\mathsf{U}$, i.e., (39). In other words, we carry over (and had already used everywhere) properties (39) to the general operation on $(\Xi)$-brace, without distinguishing between the essences of symbols $\cup$ and $\mathsf{U}$. "Micro-operationality' of empiricism and its formalization are confined, at most, by the rules (38) and (39).

Let us temporarily discontinue using the numerical terminology as applied to $(\Xi)$-objects. They differ from each other due to relationships between their "innards", rather than because of our assignment of differed symbols $(\lambda, \mu)$ to them. The brace is comprised of elements that are combined into sets and are added to them. In the language of the abstract logic, we are dealing with the fact that transitions $x$ form the brace $A, B, \ldots$, i.e., they are in the membership relationships $x \in A$, $x \in B$, $\ldots$ or, when accumulated as micro-acts, "get belonged to them". That is to say, the braces themselves and their formation (accumulation of statistics for the $\Sigma$-limit) are equivalent to a huge number of propositional "micro-sentences $x \in A$ or $x \in B$ or $\ldots$". However, again, this is nothing but a logically formal equivalent of the union operation $\cup$:

$$A \cup B = \big\{ x \,\big|\, (x \in A) \vee (x \in B) \big\} \,, \tag{40}$$

which is already being constantly exploited above.

**Remark 6.** *As is well known [134,136], due to properties of logical atoms $\in$ (membership) and $\vee$ (or), the properties of sentences such as (40) are determined precisely by rules (38) for $\cup$. Technically, we should also take an idempotence $A \cup A = A$ into account, however. At the same time, the need to have a number requires that the duplicates in ensembles have to be taken into consideration. Nevertheless, this situation is easily simulated by the set theory itself. Indeed, look first at the lower row in (24) as a strictly abstract set $\{\underline{\alpha}', \underline{\alpha}'', \ldots\} \subset \mathfrak{T}$. Then, instrument $\mathscr{A}$ "asserts" the distinguishable elements $\{\underline{\alpha}_1, \underline{\alpha}_2, \ldots\}$ and those that should be thought of as their equivalents:*

$$\underline{\alpha}_1' \approx \underline{\alpha}_1'' \approx \cdots =: \underline{\alpha}_1, \qquad \underline{\alpha}_2' \approx \underline{\alpha}_2'' \approx \cdots =: \underline{\alpha}_2, \qquad \ldots \,.$$

*This equivalence can be characterized, say, by words "a detector click at one and the same place* $\underline{\alpha}_1$*".*
*Upon such a formalization, one obtains the formation* $\{\underline{\alpha}_1'\underline{\alpha}_1''\cdots\}\{\underline{\alpha}_2'\underline{\alpha}_2''\cdots\}\cdots \approx \{\underline{\alpha}_1\cdots\}\{\underline{\alpha}_2\cdots\}$
$\cdots$*, i.e., the very lower row in* (26)*. It is within this context that we think of the union operation*
*without running into inconsistencies. Accordingly,* $(\Xi)\cup(\Xi) \neq (\Xi)$*, but the standard symbol* $\cup$
*continues to be used for simplicity.*

Therefore if we get back to the numeral labels (35) but ignore the "inner composition"
of $(\Xi)$, i.e., the **M**-paradigm, thus excluding $\cup$ and (38) from the reasoning, then all possible
$(\Xi)$-objects would turn into the semantically "segregated ideograms". Micro-transitions,
their mass nature, arbitrariness, $\napprox$-distinguishability, and the "quantumness" of the task
simply disappear. To illustrate, the obvious statement

$$\text{the brace } \{\underline{\Psi} \xrightarrow{\mathscr{A}} \underline{\alpha}\} =: (\Xi) \text{ has an empirical "kindred"}$$

$$\text{with its duplication } \{\underline{\Psi} \xrightarrow{\mathscr{A}} \underline{\alpha}, \underline{\Psi} \xrightarrow{\mathscr{A}} \underline{\alpha}\} =: (\Xi')$$

becomes pointless because the property $(\Xi)\cup(\Xi) = (\Xi')$ is missing. Furthermore, this
is despite the fact that creating the transition copies in $(\Xi')$ is a primary operation for
generating the objects and reasoning at all. The construction of the theory would then
become possible only with the interpretative introduction of the vanished concepts anew.
Therefore, macro-empiricism necessitates that the relationships (38) be operative rules, and
with that, the quantumness or classicality of consideration is of no significance.

**Remark 7.** *Let us take a closer look at the situation on the opposite—mathematical—side. The*
*union of sets* $\cup$ *is already a fundamental operation at the level of the set-theoretic formalization, e.g.,*
*the Zermelo–Fraenkel (*ZF*) axioms [134]. This is one of the first ways to create sets—the axiom of*
*union. Thus, if we believe in the set-theoretic mode of explaining/creating the quantum rudiments,*
*the quantitative description will inevitably invoke the operationality of the mathematical primitive*
$\cup$ *through rules (38). This would be suffice to declare,*

- *Inasmuch as we have nothing but* $\cup$ *and* $(\Xi)$ *(taboo* **T***), commutativity/associativity of*
  *theory is then postulated from the outset by (38), with the subsequent carrying these structures*
  *over to numerical representations, i.e., to* $\mathbb{R}$ *or* $\mathbb{C}$*.*

  *It is preferable, however, to adhere to the sequence order in ideology more stringently—*⌈*obser-*
  *vation*⌉ $\rightarrowtail$ ⌈*mathematics*⌉, ⌈*empiricism*⌉ $\rightarrowtail$ ⌈*numerical representation*⌉*—without substituting it*
  *for the opposite. At least, if we rely upon the comprehension of the empiricism as a formalization of*
  *the zero-principium of* QT *(Remark 2):*

- *Our primordial perceptions are formalized only into sets and set-theoretic* $\cup$*-abstraction (40).*

*See also [2] (p. 178), [58] (Ch. 3), [78] (p. 323), [96,104], [148] (pp. 12, 86, Ch. 4), [149] (Section V.9),*
*and Section 11.1.*

Summing up, we detect a kind of junction point: the physical and mathematical
fundamentality of operation $\cup$ for describing the elementary acts. That is to say, the
mathematics of $(\Xi)$-brace (32) and of objects (35) may not inherently be exhausted by them
as "bare" sets without structures.

Recalling now pr. **II**, we draw a conclusion regarding the very construction of the
theory.

- The reconciliation of the **R**-paradigm with empiricism must transform itself into
  *rewriting* the primary ensemble $\cup$-constructions (26), (32), (34), and relationships
  between them into the language of numerical symbols.

  More formally, we have the following continuation of pt. **R•**.

**R⁺**　*Homomorphism of the ensemble-brace properties "onto numbers"*: mutual $\cup$-relationships
(38) between the $(\Xi)$-brace should be carried over to relations between their numerical
$(\Xi)$-representations (35).

Thereby, we once again fix the maximum that is available for the building up of quantum mathematics. One may only handle the $\cup$-aggregates of transitions—constructions (32), (35)—and the minimal modules (34).

*5.2. Semigroup*

In line with (37), let us split the unitary brace (34) into two or combine two brace into one, then delete the symbols of primitives $\underline{\Psi}$ and $\underline{\Phi}$ from there. As was pointed out above, they are not necessary at this stage. By replacing the notation of upper cardinals (34) with pairs $(\infty_1', \infty_2')$ and $(\infty_1'', \infty_2'')$, upon the union, one obtains

$$(\infty_1', \infty_2') \cup (\infty_1'', \infty_2'') = (\infty_1' + \infty_1'', \infty_2' + \infty_2'') . \tag{41}$$

Here, addition $+$ obviously satisfies the properties (38). If the cardinal "$\infty$-coordinates" are replaced with the "finite percentages" $(\varkappa, \mathfrak{S})$ introduced above, i.e., if one puts

$$\left\{ \varkappa = \frac{\infty_1}{\infty_1 + \infty_2}, \quad \mathfrak{S} = \infty_1 + \infty_2 \right\}, \qquad \left\{ \infty_1 = \varkappa \mathfrak{S}, \quad \infty_2 = (1 - \varkappa)\mathfrak{S} \right\} \tag{42}$$

as in (33), then Rule (41) acquires the form of a number composition:

$$(\varkappa', \mathfrak{S}') \circ (\varkappa'', \mathfrak{S}'') = \left( \frac{\varkappa'\mathfrak{S}' + \varkappa''\mathfrak{S}''}{\mathfrak{S}' + \mathfrak{S}''}, \ \mathfrak{S}' + \mathfrak{S}'' \right) . \tag{43}$$

The commutativity/associativity properties of operation $\circ$ hold here due to the birationality of (42). Then, the formal application of $\Sigma$-postulate $\mathfrak{S}' + \mathfrak{S}'' \to \infty$ breaks, however, the symmetry $(') \leftrightarrow ('')$ and associativity of $\circ$ since

$$(\varkappa', \mathfrak{S}') \circ (\varkappa'', \mathfrak{S}'') \ \longmapsto \ \varkappa' \circ \varkappa'' = s \cdot \varkappa' + (1 - s) \cdot \varkappa'', \qquad s := \frac{\mathfrak{S}'}{\mathfrak{S}' + \mathfrak{S}''} \tag{44}$$

and $s$ is an undefined parameter. The consequence of the same kind holds true for the f-components of pairs (35), for which a convex $w$-combination of the statistical weights does arise:

$$(\mathtt{f}_1', \mathtt{f}_2', \ldots) \circ (\mathtt{f}_1'', \mathtt{f}_2'', \ldots) =: (\overline{\mathtt{f}}' \circ \overline{\mathtt{f}}'') = w \cdot \overline{\mathtt{f}}' + (1 - w) \cdot \overline{\mathtt{f}}'', \qquad w := \frac{\Sigma'}{\Sigma' + \Sigma''} . \tag{45}$$

At the same time, the splitting (41) is no more than an "intrinsic reshuffle" of one and the same $(\underline{\Xi})$-brace, which "knows nothing" about the concept of a number (numbers $s, w$), much less about the concept of observation or its numerical form. Therefore, mathematics of the ensemble structures should be independent of any representation for (37) by such operations as (43). Composition $(\underline{\Xi}') \circ (\underline{\Xi}'') = (\underline{\Xi})$ should be determined solely by its constituents $(\mathtt{f}', \varkappa')$ and $(\mathtt{f}'', \varkappa'')$, i.e., such numbers as $(s, w)$ must not appear here.

**Remark 8.** *In classical statistics, the foregoing has an analog as indifference of data on events to the way of gathering and layout thereof. For example, $(2, 3) + (1, 4) \equiv (0, 6) + (3, 1) \equiv \cdots =: \mathtt{data}$. Then, the observation proper is being created by the scheme $\mathtt{data} \rightarrowtail (3, 7) \mapsto \left( \frac{3}{3+7}, \frac{7}{3+7} \right) = (0.3, 0.7) = (\mathtt{f}_1, \mathtt{f}_2) =: \mathtt{observ}$. Parameters such as $w$ can appear in $(\underline{\Xi})$ only if, prior to any of the $\cup$-unions (37), a construction similar to (23) has been fixed. That is, the invariantly number-free brace (37) has been supplemented by an external number $w$ and ratio $w : (1 - w)$. The correction $(\underline{\Xi}) \rightarrowtail \left\{ (\underline{\Xi}')^{(w)}, (\underline{\Xi}'')^{(1-w)} \right\}$ of the theory, related to this number and to arrays (23), is very well known. This is a $w$-statistical mixture $\left\{ (w; \psi'), (1 - w; \psi'') \right\}$ of wave functions, accompanied by a formalization in terms of the statistical operator $w \cdot |\psi'\rangle\langle\psi'| + (1 - w) \cdot |\psi''\rangle\langle\psi''|$.*

Now, to ensure that numerical $(\mathtt{f}, \varkappa)$-realization (35) of ensemble brace (32) inherits quantum empiricism $(\mathbf{O}, \mathbf{M})$ and structural properties (37) and (38) properly, we reas-

sign the quantities $(\mathtt{f}, \varkappa)$ with a "percentage meaning" and replace them with different numbers $[\lambda, \mu]$:

$$(\Xi) = \left\{ \begin{bmatrix} \mu_1 \\ \lambda_1 \end{bmatrix} \underline{\alpha}_1, \; \begin{bmatrix} \mu_2 \\ \lambda_2 \end{bmatrix} \underline{\alpha}_2, \; \dots \right\} \tag{46}$$

(this important move will be touched upon once again in Section 7.1). In so doing, each pair $\begin{bmatrix} \mu' \\ \lambda' \end{bmatrix}$, $\begin{bmatrix} \mu'' \\ \lambda'' \end{bmatrix}$ behaves as a whole, and, under coinciding $\underline{\alpha}_s$, the pairs are endowed with a composition $\begin{bmatrix} \mu' \\ \lambda' \end{bmatrix} \oplus \begin{bmatrix} \mu'' \\ \lambda'' \end{bmatrix}$ that is to be commutative. Along with this, if symbol $\uplus$ denotes a composition of objects (46), it should obviously copy properties (38):

$$(\Xi) \uplus (\Psi) = (\Psi) \uplus (\Xi), \qquad \big( (\Xi) \uplus (\Psi) \big) \uplus (\Phi) = (\Xi) \uplus \big( (\Psi) \uplus (\Phi) \big) .$$

The finite ensembles are vanishingly small in their contribution into infinite ones ($\Sigma$-postulate), i.e., elements of the $(\underline{\Xi})$-family, as infinite sets, are considered modulo finite ensembles. Once again, the "finitely many" is forbidden in theory. As soon as we put the numbers of $\underline{\alpha}_1$, of $\underline{\alpha}_2$, ... to be finite, we immediately obtain the numerical distinguishability $n_1 \neq n_2, \dots$, i.e., the act of macro-observation. Let us designate the image of finite ensembles as $(0)$, and, due to property $(\Xi) \uplus (0) = (\Xi)$, it is naturally referred to as zero. The collection (46) itself has also been formed by the $\cup$-combining the ingredients

$$\left\{ \begin{bmatrix} \mu_1 \\ \lambda_1 \end{bmatrix} \underline{\alpha}_1, \; \begin{bmatrix} \mu_2 \\ \lambda_2 \end{bmatrix} \underline{\alpha}_2, \; \dots \right\} \equiv \left\{ \begin{bmatrix} \mu_1 \\ \lambda_1 \end{bmatrix} \underline{\alpha}_1 \right\} \cup \left\{ \begin{bmatrix} \mu_2 \\ \lambda_2 \end{bmatrix} \underline{\alpha}_2 \right\} \cup \cdots = \cdots ,$$

which is why the same symbol $\uplus$ may be freely used between objects with different $\underline{\alpha}_s$:

$$\cdots = \left\{ \begin{bmatrix} \mu_1 \\ \lambda_1 \end{bmatrix} \underline{\alpha}_1 \right\} \uplus \left\{ \begin{bmatrix} \mu_2 \\ \lambda_2 \end{bmatrix} \underline{\alpha}_2 \right\} \uplus \cdots .$$

For the sake of brevity, we omit the redundant curly brackets further, redefining

$$(\Xi)_{\mathscr{A}} := \begin{bmatrix} \mu_1 \\ \lambda_1 \end{bmatrix} \underline{\alpha}_1 \uplus \begin{bmatrix} \mu_2 \\ \lambda_2 \end{bmatrix} \underline{\alpha}_2 \uplus \cdots . \tag{47}$$

As a result, we have had that the set-theoretic prototypes (26) and (27), (32) of states (11) do invariantly exist in the form of every possible $\cup$-decomposition. Thus, in dealing with the only instrument $\mathscr{A}$, one reveals the following property.

- For each observation $\mathscr{A}$, the set of $(\Xi)_{\mathscr{A}}$-objects forms an infinite commutative semi-group $\mathfrak{G}$ with respect to operation $\uplus$.
  An internal (beyond the observation) nature of $(\Xi)_{\mathscr{A}}$-objects (47) is characterized by commutative superpositions $(\Xi')_{\mathscr{A}} \uplus (\Xi'')_{\mathscr{A}}$ thereof, which are independent of the classical composition of observational $\mathtt{f}$-statistics.

*5.3. Measurement*

The described above numerical $(\Xi)$-version of the $(\underline{\Xi})$-brace "$\cup$-phenomenology" makes it possible now to preliminarily formalize a concept, the absence of which deprives the theory of its basis. Namely, *measuring statistics by observation $\mathscr{A}$ over $\mathcal{S}$*:

$$\text{QM-measurement:} \qquad \big( [\lambda_1, \mu_1], [\lambda_2, \mu_2], \dots \big) \; \longmapsto \; (\mathtt{f}_1, \mathtt{f}_2, \dots) . \tag{48}$$

That is, the $[\lambda, \mu]$-collection gets mathematically mapped into the $\mathtt{f}$-statistics. This is a maximum of information provided by observation $\mathscr{A}$. Mapping (48) annihilates the pairs $[\lambda, \mu]$. Therefore, the inheritance/homomorphism of operations $\cup$ and $\uplus$ onto anything at all is eliminated. Upon operation (48), both the $(\mathtt{f}, \varkappa)$-sets and $\cup$-unions thereof, $\uplus$-operations, and the semigroup $\mathfrak{G}$ per se disappear. As a well-known result, the distinctive feature subsequently referred to as a superposition will also disappear after measurement. The new numbers $\{\mathtt{f}_s\}$ may be "added up" only as required by the different, i.e., the classical rule: forming the convex combinations (45). We note that the formalization of the measurement does not now depend on how the mathematical map $[\lambda, \mu] \rightarrowtail (\mathtt{f})$ would be further implemented—it is a separate job [6]—or how the $t$-dynamics would be introduced.

**Remark 9.** *The incorporation of t-dynamics into the theory is still impossible due to the absence of mathematics to be applied to instants $t_1$, $t_2$. Accordingly, no physical t-process, a temporal imitation of the measuring, or its dynamical description may correspond to the mathematical mapping shown in Mapping* (48). *The known "conceptual" problems with the collapse dynamics [1,13,18,45,115] are actually non-existent [15,87,93]. More precisely, they stem from the blurring of meaning that we typically give to the words "states" (what is that?), "ensembles" (what are they comprised of?), and "dynamics/collapse" (of what?). In regard to the latter, the authors of the book [58] speak out in a most definitive manner—the "fairy tales". See also Section* 10.3 *further below.*

In Section 2.1, the fundamental premise of the $\underline{\alpha}$-symbol-based distinguishability $\not\approx$ was the foundation of the entire subsequent language; "two clicks are never identical" ([126], p. 761). One then observes that the measurement or its outcome will essentially remain a vacuous term "for microsystems nothing can be directly measured" ([92], p. 304) until it invokes the concept of a QM-state, i.e., the $(\Xi)$- and $\underline{\alpha}$-objects. In the following, we shall see that, as a rough guide, everything that is observable whatsoever is a function of the state and of the state space.

Once again, it is stressed that the concept of the state must precede the notion of measurement, rather than the reverse. "[J.] Bell fulminated against the use of the word "measurement" as a primary term when discussing quantum foundations" ([30], p. 262). See also the entire chapter 23—"Against "measurement'"—in ([28], pp. 213–231).

*5.4. Covariance with Respect to Observations ("the same")*

Up to this point, we had had no need for the matching of observation $\mathscr{A}$ with observation $\mathscr{B}$, although it is clear that a description based on a certain specified $\mathscr{A}$ will inevitably be non-invariant with respect to the tool $\{\mathscr{A}, \mathscr{B}, \ldots\}$—"observation space"—and unacceptable (pt. **R**) due to the impermissible exclusivity of the set $\{\underline{\alpha}_1, \ldots\}$. At the same time, we do not have anything but $\{\mathscr{A}, \mathscr{B}, \ldots\}$ and micro-acts (12) (pts. **T** and **M**). In the brace, this fact has already been present; transitions $-\overset{\mathscr{A}}{\rightarrow}$ are combined into integrities (24). Logically, however, the $(\Xi)_{\mathscr{A}}$-, $(\Xi)_{\mathscr{B}}$-objects are incomparable and isolated from each other as carriers of statistics of different origins.

On the other hand, "*the same* is observed by instrument $\mathscr{B}$, as by instrument $\mathscr{A}$". Although this context has not yet been invoked, without it, the application of set-theoretic constructs to physics is devoid of meaning, just like the union of the speeds of an electron and of the Moon into a set $\{v_{\mathrm{e}}, v_{\mathrm{M}}\}$, with the subsequent creating a certain physical characteristic of this "two-body system"—say, the mean velocity $\frac{1}{2}(v_{\mathrm{e}} + v_{\mathrm{M}})$. Indeed, "The statements of quantum mechanics are meaningful and can be logically combined *only* if one can imagine a *unique experimental context*" ([40], p. 115).

Thus, the global structuredness is required in the set of various $(\Xi)$-data according to the context "the same, identical" or its negation. Apparently, this addition implies such entities as "the same particle", "in the same preparation/state", "under the same temperature and **M**-environment (12)", "the same closed system $\mathcal{S}$", "in the same external field", "in the same interferometer" with "the same detectors/solenoids", the like [33,40,44]; the short and generalized notation $\langle\!\langle \mathcal{S}, \mathbf{M}, \ldots \rangle\!\rangle$. All the notions here, including the state, are physical conventions, yet their formalization and modeling are called for the creation of a theory (Section 2.4).

The notion "with the same initial data" falls under the same category, if the intention is to use the term time $t$. Again, the very creation of the $(\Xi)$-brace as a set "by the piece" is from the outset thought of as (Section 2.4) a creation on the assumption of common $\langle\!\langle \mathcal{S}, \mathbf{M}, \ldots \rangle\!\rangle$. For instance, the $\mathscr{A}$-statistics $(\Xi)_{\mathscr{A}}$ are gathered within "the same" $\langle\!\langle \mathcal{S}, \mathbf{M}, \ldots \rangle\!\rangle$ as the $\mathscr{B}$-statistics $(\Xi)_{\mathscr{B}}$. On its part, any variation is sufficient to obtain "not the same", even if we *envision it as null* in the spirit of the widely known "without in any way disturbing a system" ([131], p. 234). To take an illustration, equipment of interferometer (Section 6.5) with additional "which-slit" detectors is already at variance with the notion of "the same

$\mathcal{S}''$. In similar cases, we end up in situations similar to Case (23) since the detectors cause an $\underline{\alpha}$-distinguishability.

Notice that the notions "the same" and "distinguishable" (Remark 2), while antonymous, mutually exclude each other. Semantically, one without the other makes no sense, which closely resembles Bohr's conception of complementarity [78].

It follows from the above that in order to match $\mathscr{A}$ and $\mathscr{B}$, the *metatheoretical* [149] category $\langle\!\langle \mathcal{S}, \mathbf{M}, \dots \rangle\!\rangle$ is required; however, we are only in possession of the ensemble brace $(\Xi)_{\mathscr{A}}$ and $(\Xi)_{\mathscr{B}}$ (pt. **T**). On the other hand, without joint consideration of *the two instruments*, i.e., without introducing a mechanism for the mathematical matching $(\Xi')_{\mathscr{A}} \rightleftarrows (\Xi')_{\mathscr{B}}$, $(\Xi'')_{\mathscr{A}} \rightleftarrows (\Xi'')_{\mathscr{B}}$, $\dots$, the segregation of the $(\Xi)$-objects is absolute. (It is clear that the matching of single micro-events $\underline{\Psi} \overset{\mathscr{A}}{\dashrightarrow} \underline{\alpha}_s$ and $\underline{\Psi} \overset{\mathscr{B}}{\dashrightarrow} \underline{\beta}_j$ is also futile.) It is impossible to associate physics with the abstractly segregated $(\Xi)_{\mathscr{A}}$-brace. Otherwise, the solitary object $(\Xi)_{\mathscr{A}}$, generating nothing more than statistics provided by the single instrument $\mathscr{A}$, would yield a description of everything, which is absurd by pt. $\mathbf{R}^{\bullet\bullet}$. The physical contents (to come) arise precisely through the above-mentioned matching (see Section 6.4 below).

As a result, we adopt a kind of the relativity-principle analogue—a tenet on the quantum observational covariance.

**III**　　Theory should introduce a means of equating the macro-observations (pts. **O** + **M**) by differing instruments $\{\underline{\alpha}_1, \dots\}_{\mathscr{A}} \neq \{\underline{\beta}_1, \dots\}_{\mathscr{B}}$ under a common (the same) experimental environment $\langle\!\langle \mathcal{S}, \mathbf{M}, \dots \rangle\!\rangle$.

(*The third principium of quantum theory*)

Cf. [22] (p. 632) and mathematical analogies [85,86].

5.4.1. Semantic Closedness and the Equal Sign $=$

We are currently returning once again to Section 2.1, falling into a situation when the case in hand does not just entail fundamental theory in the form of $\lceil$math$\rceil$ + $\lceil$physical "bla-bla-bla"$\rceil$, while, continuing on an informal note, the mathematics of physics—quantum mathematics—is being created "from scratch". When building up this math, it is impossible to forego the physical conventions $\langle\!\langle \mathcal{S}, \mathbf{M}, \dots \rangle\!\rangle$; meanwhile, any preliminary and the formal characterization for $\langle\!\langle \mathcal{S}, \mathbf{M}, \dots \rangle\!\rangle$ is ruled out.

Indeed, the attempts to mathematically formalize the physical context of observation, rather than observation itself, will not logically manage without another "more fundamental" observation, in this case, of the very experimental environment. The semantic cycling is apparent here, and any of its mathematization will lead to a retrogression of definitions into infinity, which is known as the "von Neumann catastrophe" ([80], pp. 158–$\dots$)) or as "trying to swallow itself by the tail" ([28], p. 220). Which is why, once again, the "Box (6) method" prohibitions are required. See also a paragraph containing the capitalized emphasize "CANNOT IN PRINCIPLE" on p. 418 of the work [15]. Sooner or later, it will have to be declared that mathematics will be created for *the* convention $\langle\!\langle \mathcal{S}, \mathbf{M}, \dots \rangle\!\rangle$ and that this mathematics will be a mathematical model for this $\langle\!\langle \mathcal{S}, \mathbf{M}, \dots \rangle\!\rangle$. The analogous argument—"mathematics is there to serve physics, and not the other way round" ([16], p. 242; L. Hardy)—has already long been met in the literature [23,33,40]. In connection with the "general contextual models", see the books [64,150] (the Växjö-model, "quantum contextuality") and bibliography therein.

**Remark 10** (semantic). *To avoid the just mentioned linguistic closedness—a kind of mathematical "pathology" of the physical and natural languages—a description that lays claim to the role of an unambiguous/rigorous theory requires a careful separation of the object- and meta-languages. For more detail, see [105] (Sections 14–16), [106] (Section V.1), and [136] (Section 3.9). For this reason, the constructs should track the blending of the object QM-domain (syntax) and the meta-domain (semantics) and, more generally, the penetration of extra-linguistic elements of thinking [148] into QM. The notion of "the same $\langle\!\langle \mathcal{S}, \mathbf{M}, \dots \rangle\!\rangle$", which is intuitive in the natural language, should explicitly be indicated as an external and fundamental category (pr. III), and its circular re-interpretations/re-translations within the theory should be banned. That is, re-stating "the sameness $\langle\!\langle \mathcal{S}, \mathbf{M}, \dots \rangle\!\rangle$, the*

*identical $\langle\!\langle \mathcal{S}, \mathbf{M}, \ldots \rangle\!\rangle$" by way of word or of the equality symbol $=$ between some other entities is forbidden.*

- *"The same" may no have a definition in terms of anything else. It exists prior to theory and has only a meaning ($=$ verbal context), though its natural-language descriptions may be of great variety and be "presented to us in wildly different ways" ([86], p. 2 and the whole of the Section "The awkwardness of equality").*

  *One could, e.g., accept the typical verbal vehicle "a complex of conditions, which allows of any number of repetitions" (quotation from the literature). It is clear that the words "complex . . . allows . . . repetitions" here are just another semantic equivalent to the word "the same $\langle\!\langle \mathcal{S}, \mathbf{M}, \ldots \rangle\!\rangle$". The physics terminology per se (Sections 6.4, 6.5, and 9.1) will become accessible when physical concepts are introduced via the originating—and obligatorily very ascetic—quantum-mechanics language. See also the selected thesis on page 70.*

It is crucial to immediately note that, in the same manner, the classical description contains the cited arguments in their entirety. It is easy to convince that such a description also implies implicitly that which is designated above as $\langle\!\langle \mathcal{S}, \mathbf{M}, \ldots \rangle\!\rangle$; otherwise, the physical reasoning would be entirely impossible. "[W]e often prefer to regard a number of outcomes of distinct physical operations as registering the same property, . . . representing the same measurement. . . . permitting an unrestricted identification of outcomes would lead to "grammatical chaos"" (Foulis–Randall ([111], p. 232)). More to the point, the physics and mathematics not merely have been closely interwoven with each other. Any recursive procedure of definitions will inevitable result in either a cyclic definition at some level, or a definition that refers outside not only of the physics but even of the math. Hence, the hierarchical arrangement of notions/. . ./definitions—a property that is frequently uncontrolled and violated in the human thinking—can only be meaningful if at least one knot in the definition network is externally defined. In this work, that basic points are, as a rough guide, the brace $\underline{\Psi} \stackrel{\mathscr{A}}{\dashrightarrow} \underline{\alpha}$ and the notion of "the same $\langle\!\langle \mathcal{S}, \mathbf{M}, \ldots \rangle\!\rangle$", motivated in Remarks 2 and 10, respectively.

**Remark 11.** *Here, the situation is similar to the role of the axiom of choice in the ZF-system [134,147]. It has been well known for a long time that the axiom is often subconsciously implied ([149], Chs. II, IV); it can also not be either circumvented or ignored. Another counterexample to "infinite retrogression and circularity" in logic comes from the very same system. This is a ban on infinite chain of set memberships $\in$ on the left*

$$\| \cdots \in X_n \in \cdots \in X_2 \in X_1 \in X_0$$

*(the regularity axiom $[\forall x \in X, \ x \cap X \neq \varnothing] \Rightarrow [X = \varnothing]$) under the permissibility of the infinite $(\in)$-continuing to the right:*

$$X_0 \in X_1 \in X_2 \in \cdots \in X_n \in \cdots \in \cdots$$

*(not rigorously, the infinity axiom) [120,134].*

*The obvious parallels here are the famous Russell paradox [149] or a chaos in the computer file system when the "hard links" from a folder to the parent folder are allowed. Thus, the relations $\in$ "downwards" to the left and necessarily terminates in something, i.e., in a set $X_0$ that contains nothing: $\varnothing = X_0 \in \cdots \in X_n \in \cdots$. Therefore, one needs to give "meaning" to the only set—the empty one $\varnothing$. Incidentally, it is these axioms that guarantee the existence of infinitely many ordinal numbers (106) and the uniqueness of this structure. The ordinals and numbers have yet to be dealt with further below in more detail.*

All that remains is to add that no theory in physics is feasible without re-calculations of physical units and of vectors/tensors without transformations in the fiber superstructures over manifolds, etc. Accordingly, the considerations on invariance and on transformations should be present in the quantum case as well, but it, which is its principal difference from

the classical case, still lacks the concepts of physical quantities/properties (see Section 6.4). Therefore, such argumentation may only be applied to those objects that we have at our disposal, i.e., to the $(\Xi)$-brace. The renunciation of pr. **III** would actually be tantamount to the inability to make the physics theories whatsoever.

Now, pr. **III** and the "quantum diversity of the reference frames" $\{\mathscr{A}, \mathscr{B}, \dots\}$ require a kind of factorization of the entire family $\{(\Xi)_{\mathscr{A}}, (\Xi)_{\mathscr{B}}, \dots, (\Xi)'_{\mathscr{A}}, (\Xi)'_{\mathscr{B}}, \dots\}$ with respect to the conception $\langle\!\langle \mathcal{S}, \mathbf{M}, \dots \rangle\!\rangle$, i.e., the introduction of an operation of equating the results $(\Xi)_{\mathscr{A}}, (\Xi)_{\mathscr{B}}$ that came when observing $\mathcal{S}$. $(\Xi)_{\mathscr{A}} \overset{?}{=} (\Xi)_{\mathscr{B}}$ should not be immediately put since these braces are simply different sets. That is why, with isolated semigroups

$$\Big\{ \underbrace{\overset{\mathscr{A}}{\uplus}; (\Xi')_{\mathscr{A}}, (\Xi'')_{\mathscr{A}}, \dots}_{\mathfrak{G}_{\mathscr{A}}} \Big\}, \quad \Big\{ \underbrace{\overset{\mathscr{B}}{\uplus}; (\Xi')_{\mathscr{B}}, (\Xi'')_{\mathscr{B}}, \dots}_{\mathfrak{G}_{\mathscr{B}}} \Big\}, \quad \dots$$

at our disposal, we have to conceive of them as elements of a new set $H$ of objects having a single nature, 1) to carry out the mapping $\{\mathfrak{G}_{\mathscr{A}}, \mathfrak{G}_{\mathscr{B}}, \dots\} \mapsto H$, assigning new representatives $|\Xi_{\mathscr{A}}\rangle \in H$ to the $(\Xi)$-brace, and 2) to equip $H$ with an equivalence relation $|\Xi_{\mathscr{A}}\rangle \approx |\Xi_{\mathscr{B}}\rangle$ (the concept "the same" above). Let us implement all of that by the scheme

$$(\Xi)_{\mathscr{A}} := \begin{bmatrix} \mu_1 \\ \lambda_1 \end{bmatrix} \underline{\alpha}_1 \overset{\mathscr{A}}{\uplus} \begin{bmatrix} \mu_2 \\ \lambda_2 \end{bmatrix} \underline{\alpha}_2 \overset{\mathscr{A}}{\uplus} \cdots \quad \longmapsto \quad \begin{bmatrix} \mu_1 \\ \lambda_1 \end{bmatrix} |\alpha_1\rangle \pm \begin{bmatrix} \mu_2 \\ \lambda_2 \end{bmatrix} |\alpha_2\rangle \pm \cdots =: |\Xi_{\mathscr{A}}\rangle \in H,$$

$$(\Xi)_{\mathscr{B}} := \begin{bmatrix} \mu_1 \\ \lambda_1 \end{bmatrix} \underline{\beta}_1 \overset{\mathscr{B}}{\uplus} \begin{bmatrix} \mu_2 \\ \lambda_2 \end{bmatrix} \underline{\beta}_2 \overset{\mathscr{B}}{\uplus} \cdots \quad \longmapsto \quad \begin{bmatrix} \mu_1 \\ \lambda_1 \end{bmatrix} |\beta_1\rangle \pm \begin{bmatrix} \mu_2 \\ \lambda_2 \end{bmatrix} |\beta_2\rangle \pm \cdots =: |\Xi_{\mathscr{B}}\rangle \in H,\tag{49}$$

$$\dots\dots\dots \qquad\qquad \dots\dots \qquad\qquad \dots\dots\dots$$

In this, the new addition $\pm$ must of course homomorphically inherit operations $\overset{\mathscr{A}}{\uplus}, \overset{\mathscr{B}}{\uplus},$ $\dots$, and the extension of this definition throughout $H$ is then made with the aid of the very equivalence $\approx$:

$$|\Xi'_{\mathscr{A}}\rangle \pm |\Xi''_{\mathscr{B}}\rangle = \Big| |\Xi''_{\mathscr{B}}\rangle \approx |\Xi''_{\mathscr{A}}\rangle \Rightarrow \Big| = |\Xi'_{\mathscr{A}}\rangle \pm |\Xi''_{\mathscr{A}}\rangle = |\Xi'_{\mathscr{B}}\rangle \pm |\Xi''_{\mathscr{B}}\rangle.$$

The negation $\not\approx$ of the relation $\approx$, e.g., $|\Xi'_{\mathscr{A}}\rangle \not\approx |\Xi''_{\mathscr{A}}\rangle$, is exactly the very same distinguishability that was discussed in Sections 2 and 3.

For the sake of convenience, we adopt the regular sign $=$ for $\approx$ in order not to introduce yet a further homomorphism, which are already numerous, with more to come. In other words, the physics $\langle\!\langle \mathcal{S}, \mathbf{M}, \dots \rangle\!\rangle$ is "concentrated" in the sign $=$, turning the empirical structures (49) into the $\mathscr{A}$-, $\mathscr{B}$-implementations of the object $|\Xi\rangle \equiv |\Xi_{\mathscr{A}}\rangle = |\Xi_{\mathscr{B}}\rangle$ under construction. The adequate term for it—the `Info/Data-Source` or "representative of information" (Č. Brukner (2014))—corresponds to the preliminary prototype of the concept of a state, but we will remain within the standard term, disregarding its variance.

## 6. Quantum Superposition

How come the quantum? … No space, no time—J. Wheeler (1989)

… postulation of something as a Primary Observable is itself a sort of theoretical act and may turn out to be wrong—T. Maudlin ([151], p. 142)

### 6.1. Representations of States

Let us simplify notation according to the rule $\begin{bmatrix} \mu \\ \lambda \end{bmatrix} =: \mathfrak{a}$. The sought-for relationships between $\mathscr{A}, \mathscr{B}, \dots$ then turn into the key point of further construct—the equalities

$$\begin{bmatrix} \text{representations} \\ \text{of } |\Xi\rangle\text{-state} \end{bmatrix} : \quad \mathfrak{a}_1 |\alpha_1\rangle \pm \mathfrak{a}_2 |\alpha_2\rangle \pm \cdots \overset{\text{"the same"}}{=\!=\!=} \mathfrak{b}_1 |\beta_1\rangle \pm \mathfrak{b}_2 |\beta_2\rangle \pm \cdots = \cdots. \tag{50}$$

They furnish *representations* $|\Xi_{\mathscr{A}}\rangle, |\Xi_{\mathscr{B}}\rangle, \dots$ *of quantum state* $|\Xi\rangle$ *of system* $\mathcal{S}$. By design, this `DataSource` object $|\Xi\rangle$ carries data $(\Xi)_{\mathscr{A}}, (\Xi)_{\mathscr{B}}$ and, more generally, $(\Xi)$-data (47) from arrays of *any* observations, including the imaginary ones. That is what eliminates

the initial need for the $(\Xi)_{\mathscr{A}}$-brace (24) to come from the observation $\mathscr{A}$, which is reflected in the shortening of the term "representation of state" to simply "state" $|\Xi\rangle$. It should be added that the straightforward storing of objects $\{|\Xi_{\mathscr{A}}\rangle, |\Xi_{\mathscr{B}}\rangle, \ldots\}$ in a certain set $H$, but with the independence of operations $\{\pm^{(\mathscr{A})}, \pm^{(\mathscr{B})}, \ldots\}$ preserved, would not differ from the tautological substitution of symbols. Accordingly, the semantic autonomy of $(\underline{\Xi})$-brace would also be inherited, whereas covariance **III** requires an elimination of precisely this autonomy. What is more, the set-theoretic original copy for operations $\{\overset{\mathscr{A}}{\uplus}, \overset{\mathscr{B}}{\uplus}, \ldots\}$ and $\pm$ is one and the same—the union $\cup$.

The symbols $|\alpha_s)$ and $|\beta_s)$ in (50) are no more than symbols. Hence, the objects' property (50) of being identical must be reflected in terms of their coordinate $\mathfrak{a}, \mathfrak{b}$-components (pt. $\mathbf{R}^\bullet$). This means that any aggregate $(\mathfrak{a}_1, \mathfrak{a}_2, \ldots)$ is unambiguously calculated by means of a certain transformation $\widehat{U}$ into any other $(\mathfrak{b}_1, \mathfrak{b}_2, \ldots)$ when the two aggregates represent a common $|\Xi\rangle$:

$$(\mathfrak{a}_1, \mathfrak{a}_2, \ldots) = \widehat{U}(\mathfrak{b}_1, \mathfrak{b}_2, \ldots) .$$

The $\widehat{U}$ then becomes an isomorphism between these aggregates (a preimage of the future unitary transformation ([6], p. 14)) and, accordingly, their lengths must coincide. This length—a certain single constant—will be symbolized as D.

### 6.2. Representations of Devicesand Spectra

Naturally, the instrument is converted to the *H*-structure language along with $(\Xi)$-objects. It is a set of symbols $\{|\gamma_1), |\gamma_2), \ldots\}$ in place of the previous $\{\gamma_1, \gamma_2, \ldots\}$. As has just been shown, their number for any $\mathscr{C}$-instrument should be equal to $\bar{\mathrm{D}}$. However, generally speaking, $|\mathfrak{T}_{\mathscr{A}}| \neq |\mathfrak{T}_{\mathscr{B}}|$ since $\mathfrak{T}_{\mathscr{A}}$ and $\mathfrak{T}_{\mathscr{B}}$ are assigned in an arbitrary way (pt. **O**). Therefore, if we take an illustration $\mathscr{A}\{\underline{\alpha}_1, \underline{\alpha}_2\}$ and $\mathscr{B}\{\underline{\beta}_1, \underline{\beta}_2, \underline{\beta}_3\}$, then *H*-representation of instrument $\mathscr{A}$ should appear at least as $\{|\alpha_1), |\alpha_2), |\alpha_3)\}$. Clearly, the already present distinguishability $\underline{\alpha}_1 \not\simeq \underline{\alpha}_2$ (Section 2.2) is automatically converted into an abstract distinguishability of new symbols $|\alpha_1) \neq |\alpha_2)$, and empirical $\mathscr{A}$-distinguishability is confined exclusively by these two symbols. In that case, for the purpose of noncontradiction, the added third symbol $|\alpha_3)$, as an adjunction to the abstract relations $|\alpha_3) \neq |\alpha_1)$ and $|\alpha_3) \neq |\alpha_2)$, should be complemented with the notion of its physical *in*discernibility from $|\alpha_1)$ or $|\alpha_2)$. By an extension of this argument, one obtains that every $\mathscr{A}$-instrument should be endowed with the (non)equivalence relation $(\simeq / \not\simeq)$ in terms of the *H*-structure by its formal $\{|\alpha_1), \ldots\}$-representations. How do we do this?

Let us proceed further from a self-suggested extension of pt. **R**. Let us declare—and it is more than natural—that the number representations $\alpha_s$ are linked not only to observations but to instruments as well. Each $\alpha_s$ is the new object of a numerical type: a number or a collection of numbers. Then, indiscernibility, say $|\alpha_3) \simeq |\alpha_1)$, is recorded by coincidence of the numeral labels $\alpha_3 = \alpha_1$ attached to the symbols $|\alpha_3)$ and $|\alpha_1)$, respectively. The abstract ("old") distinguishability $|\alpha_3) \neq |\alpha_1)$, meanwhile, remains as it is. From here, we have the following formalization of the relationship between $\simeq$ and $=$ by means of dropping/adding the brackets $|\ )$:

$$\left.\begin{array}{ccc} |\alpha_s) \neq |\alpha_k) & \Longleftrightarrow & \alpha_s \neq \alpha_k \\ |\alpha_s) \simeq |\alpha_k) & \Longleftrightarrow & \alpha_s = \alpha_k \end{array}\right\} \quad \text{under} \quad |\alpha_s) \neq |\alpha_k) . \tag{51}$$

Call the quantity $\alpha_s$ (numerical) the *spectral label/marker* of eigen-element $|\alpha_s)$. Then, by the *H-representation* $[\mathscr{A}]$ of instrument $\mathscr{A}$, we will mean the set of objects $\{|\alpha_1), \ldots, |\alpha_\mathrm{D})\}$ supplemented with the spectral structure (51):

$$[\mathscr{A}] := \left\{ |\alpha_1)_{\lfloor\alpha_1}, |\alpha_2)_{\lfloor\alpha_2}, \ldots \right\} . \tag{52}$$

It is not difficult to see that if $|\alpha_1) \neq |\alpha_2)$, then either $|\alpha_3) \simeq |\alpha_1)$ or $|\alpha_3) \simeq |\alpha_2)$. Otherwise, spectral markers $\lfloor\alpha_1 = \lfloor\alpha_2$ should coincide, and primary primitives $\underline{\alpha}_1 \not\simeq \underline{\alpha}_2$ lose

their empirical distinguishability in contrast to (7). The multiple coincidence of $\lfloor\alpha_s$-markers is admissible.

In the presence of relations (51), it is natural to state that instrument $\mathscr{A}$ is coarser (more symmetrical) than $\mathscr{B}$ and, terminologically, to declare that the degeneration of the spectral-label values takes place. In cases of embeddability such as $\mathscr{A}_2\{\underline{\alpha}_1, \underline{\alpha}_2\} \subset \mathscr{A}_3\{\underline{\alpha}_1, \underline{\alpha}_2, \underline{\alpha}_3\}$, instrument $\mathscr{A}_2$ can even be called the same as (coinciding with) $\mathscr{A}_3$, but with a more rough scale. Conversely, $\mathscr{A}_3$ is a more precise extension of $\mathscr{A}_2$. In particular, the natural notion of a device resolution fits here.

All instruments may then be mathematically imagined as having the same resolution, but, perhaps, with degeneration of spectra. The non-coinciding instruments may be interpreted as non-equivalent reference frames $\mathscr{A} \neq \mathscr{B}$ in an observation space. According to pts. **R$^{\bullet\bullet}$** and **III**, they are mandatorily present in the description. The spectral degenerations are also always present since element $\underline{\alpha}_1$ can always be removed from $\mathfrak{T}_{\mathscr{A}}$, and there are no logical foundations to prohibit an observational instrument with family $\mathfrak{T}_{\mathscr{A}} - \{\underline{\alpha}_1\}$. Hence, it follows that introducing the spectra—instrumental readings—is required even formally, without physics. It is of course implied here that spectral (in)discernibility is realized in the same manner as its statistical counterpart in Sections 2.6 and 4, i.e., by numbers. Incidentally, such a property of $\lfloor\alpha_s$—i.e., of being a numerical object—is not at all necessary at the moment. The spectrum $\{\lfloor\alpha_1, \lfloor\alpha_2, \ldots\}$ may be thought of as an abstract set of labels attached to the eigen-elements. As numbers, it is introduced for the subsequent creation of models to classical/macroscopic dynamic, and they are numerical.

Returning to D, we note that, in any case, the toolkit $\{\mathscr{A}, \mathscr{B}, \ldots\} =: \mathcal{O}$ in real use has always been defined, fixed, and is finite. Consequently, the constant

$$\mathrm{D} \geqslant 2 \tag{53}$$

has also been defined and fixed, and it becomes the globally static observable characteristic—an empirically external parameter. Meanwhile, the entire scheme internally contains the natural method of its own extension $\mathrm{D} \mapsto \mathrm{D} + 1$, and the potentially all-encompassing choice $\mathrm{D} = \infty$ may be considered the universally preferable one in QT. By freezing the different $\mathrm{D} < \infty$, the theory makes it possible to create models, and they are not only admissible but also well-known. Their efficiency is examined in experiments. Once again:

- The D-constant concept of spectra and their degenerations is created by the $(\mathscr{A}, \mathscr{B})$-covariance requirement, i.e., by principium **III**.

As a result, the structure of $H$-representations of states and of instruments are liberated from the arbitrariness in assigning the subsets $\mathfrak{T}_{\mathscr{A}}$ in (8). The statistical unitary pre-images (34) and $H$-elements of the form $\mathfrak{c}\lfloor\gamma_s)$ can be associated with any "eigen symbol" $\lfloor\gamma_s)$. They are always available because every possible brace (32) is known to contain subfamilies when ongoing $\underline{\Psi}, \underline{\Phi}$-primitives get to a single one, e.g., to $\underline{\gamma}_1$. Therefore, every representation $\mathfrak{a}_1\lfloor\alpha_1) \pm \cdots$ is always equivalent to a $(\underline{\Xi})_{\mathscr{C}}$-brace for some observation $\mathscr{C}$ with a homogeneous outgoing ensemble $\{\underline{\gamma}_1 \cdots \underline{\gamma}_1\}$. That is, one may always write

$$\mathfrak{a}_1\lfloor\alpha_1) \pm \mathfrak{a}_2\lfloor\alpha_2) \pm \cdots = \mathfrak{c}_1\lfloor\gamma_1) \pm 0\lfloor\gamma_2) \pm \cdots =: \mathfrak{c}_1\lfloor\gamma_1), \tag{54}$$

while naturally referring to $\mathfrak{c}_1\lfloor\gamma_1)$ as one of the *eigen-states of instrument $\mathscr{C}$*, with an appropriate adjustment of the similar definition in pt. **O**. The construction of the representation-state space is far from being complete since it is still a "bare" semigroup $H$.

### 6.3. Superposition of States

Since writings (50) exist for any ensemble $(\underline{\Xi})$-brace, let us consider the following two representations:

$$\mathfrak{a}_1\lfloor\alpha_1) \pm \mathfrak{a}_2\lfloor\alpha_2) = \mathfrak{b}_1\lfloor\beta_1) \pm \mathfrak{b}_2\lfloor\beta_2) \pm \cdots,$$
$$\mathfrak{a}_2\lfloor\alpha_2) = \mathfrak{b}_1'\lfloor\beta_1) \pm \mathfrak{b}_2'\lfloor\beta_2) \pm \cdots. \tag{55}$$

Comparison of these equalities tells us that the second one is a solution of the first one with respect to $\mathfrak{a}_2|\alpha_2\rangle$. Hence, the semigroup operation $\dotplus$ admits a cancellation of element $\mathfrak{a}_1|\alpha_1\rangle$. This means that there exists an *H*-element $\tilde{\mathfrak{a}}_1|\alpha_1\rangle$ such that

$$\left\{\tilde{\mathfrak{a}}_1|\alpha_1\rangle \dotplus \mathfrak{a}_1|\alpha_1\rangle\right\} \dotplus \mathfrak{a}_2|\alpha_2\rangle = \tilde{\mathfrak{a}}_1|\alpha_1\rangle \dotplus \left\{\mathfrak{b}_1|\beta_1\rangle \dotplus \mathfrak{b}_2|\beta_2\rangle \dotplus \cdots\right\}$$
$$\Downarrow$$
$$0|\alpha_1\rangle \dotplus \mathfrak{a}_2|\alpha_2\rangle = \tilde{\mathfrak{a}}_1|\alpha_1\rangle \dotplus \mathfrak{b}_1|\beta_1\rangle \dotplus \mathfrak{b}_2|\beta_2\rangle \dotplus \cdots$$
$$\Downarrow \quad \text{(due to (54))}$$
$$\mathfrak{a}_2|\alpha_2\rangle = \mathfrak{b}_1'|\beta_1\rangle \dotplus \mathfrak{b}_2'|\beta_2\rangle \dotplus \cdots$$
$$\Downarrow \qquad\qquad \Downarrow$$
$$|0\rangle := 0|\alpha_1\rangle = \tilde{\mathfrak{a}}_1|\alpha_1\rangle \dotplus \mathfrak{a}_1|\alpha_1\rangle, \qquad \tilde{\mathfrak{a}}_1|\alpha_1\rangle \dotplus \mathfrak{b}_1|\beta_1\rangle \dotplus \cdots = \mathfrak{b}_1'|\beta_1\rangle \dotplus \cdots,$$

where $|0\rangle$ stands for a zero in the semigroup *H* (image $(0)$ of the finite-length brace $(\Xi)$) and $0$ in $0|\alpha_1\rangle$ is a symbol of its $[\lambda, \mu]$-coordinates. By canceling out $\mathfrak{a}_s|\alpha_s\rangle$, one by one, if necessary, one deduces that any element of *H* does have an inversion. That is, *H* is actually a group. We re-denote inverse elements $\tilde{\mathfrak{a}}_s|\alpha_s\rangle$ by $(-\mathfrak{a}_s)|\alpha_s\rangle$ and inversions of sums are formed from $(\dotplus)$-sums thereof. Moreover, all the $[\lambda, \mu]$-pairs turn into a set $\{\mathfrak{a}, \mathfrak{b}, \ldots\}$ equipped with the above-mentioned composition $\oplus$, which follows from an obvious property of unitary brace:

$$\mathfrak{a}|\alpha_1\rangle \dotplus \mathfrak{b}|\alpha_1\rangle = (\mathfrak{a} \oplus \mathfrak{b})|\alpha_1\rangle \tag{56}$$

(inheritance of clossedness under the $\cup$-operation). This composition is also a $\oplus$-operation of a group and of a commutative one:

$$\mathfrak{a} \oplus \mathfrak{b} = \mathfrak{b} \oplus \mathfrak{a}, \qquad (\mathfrak{a} \oplus \mathfrak{b}) \oplus \mathfrak{c} = \mathfrak{a} \oplus (\mathfrak{b} \oplus \mathfrak{c}), \qquad \mathfrak{a} \oplus 0 = \mathfrak{a}, \qquad \mathfrak{a} \oplus (-\mathfrak{a}) = 0 . \tag{57}$$

Therefore, the group nature of semigroup *H* and the group (57) come from the scheme

$$
\begin{bmatrix} \text{single observations} \\ \mathscr{A}, \mathscr{B}, \ldots \end{bmatrix} \Rightarrow \begin{bmatrix} \text{semigroups} \\ \mathfrak{G}_{\mathscr{A}}, \mathfrak{G}_{\mathscr{B}}, \ldots \end{bmatrix} \longmapsto
$$
$$
\longmapsto \begin{bmatrix} (\mathscr{A}, \mathscr{B})\text{-covariance}, \\ \langle\!\langle \mathcal{S}, \mathbf{M}, \ldots \rangle\!\rangle \text{ and principium } \mathbf{III} \end{bmatrix} \Rightarrow \lceil \text{group } H \rceil
$$

and, technically, from equatings/identifyings (50), i.e., from conception "the same" (Section 5.4). For its part, it is this very structure of algebraic operations—the two- and three-term (and nothing else) axioms of commutativity/associativity, i.e., the group—that comes from properties (39). All of this provides an answer to the key question: where do the (semi)group and the minus sign come from and why?

Thus, handling the $|\Xi\rangle$-objects breaks free from its ties to the notion of observation, and the objects admit the formal writings $\mathfrak{a}|\Psi\rangle \dotplus \mathfrak{b}|\Phi\rangle \dotplus \cdots$. Call them *superpositions*. However, as soon as they or the state are associated in meaning with the word "readings" (this is discussed at greater length in Sections 6.4 and 6.5), this term should be replaced with a non-truncated one, i.e., a representation of the state with respect to a certain observation. Specifically, the statistical weights $\mathfrak{f}_j$ are extracted from such expressions only after their conversion into a sum over eigen-states of the form (50); a task of the subsequent mathematical tool.

No superposition $\mathfrak{a}|\Psi\rangle \dotplus \mathfrak{b}|\Phi\rangle \dotplus \cdots$, including (54), has any physical sense in and of itself [5] (p. 137), [12] nor is it preferable to any other one. It merely mirrors the closedness of states with respect to operation $\dotplus$ since any $|\Xi\rangle$ is re-recorded as a sum of various $\{\mathfrak{a}|\Psi\rangle, \mathfrak{b}|\Phi\rangle, \ldots\}$ in a countless number of ways and is linked to any other such sum. Without a system of $|\alpha_s\rangle$-symbols for instrument $\mathscr{A}$, nothing observable is extractable out of the aggregate of coefficients $\{\pm\mathfrak{a}, \pm\mathfrak{b}, \ldots\}$ (and, of course, of the $|\Psi\rangle$-letters themselves) in any imaginable way. Accordingly, it is incorrect to speak of—a widespread misconception—

the destruction of the superposition or of the "relative-phase information" ([119], p. 253), associating the word destruction with the physical/observational meanings or processes.

As a result, *even without having a numerical theory yet* and without recourse to the concept of a physical quantity, superposition may not address whatever physical concepts, we arrive at the paramount property, which characterizes the most general type of micro-observation's ensembles (17).

- *Superposition principle*

  A ($\pm$)-composition of quantum states $\mathfrak{a}|\Psi\rangle$ and $\mathfrak{b}|\Phi\rangle$, which are admissible for system $\mathcal{S}$, is an admissible state

$$\mathfrak{a}|\Psi\rangle \pm \mathfrak{b}|\Phi\rangle = \mathfrak{c}|\Xi\rangle \tag{58}$$

  and, with that, the set $\left\{\mathfrak{a}|\Psi\rangle, \mathfrak{b}|\Phi\rangle, \mathfrak{c}|\Xi\rangle, \dots\right\} =: H$ forms a commutative group with respect to operation $\pm$. The family $\{\mathfrak{a}, \mathfrak{b}, \dots\}$ of coordinate $\mathbb{R}^2$-representatives of states (50) is also equipped with the same group structure under the $\oplus$-operation (57) and with the rule of carrying the operation $\pm$ over to $\oplus$:

$$\mathfrak{a}|\Psi\rangle \pm \mathfrak{b}|\Psi\rangle = (\mathfrak{a} \oplus \mathfrak{b})|\Psi\rangle. \tag{59}$$

Let us clarify the transferring of (56) to (59). The union of the state prototypes $\mathfrak{a}|\Psi\rangle$, $\mathfrak{b}|\Psi\rangle \in H$ is known to belong to $\mathfrak{G}$. Thus, the composition $\mathfrak{a}|\Psi\rangle \pm \mathfrak{b}|\Psi\rangle$ should be identical to a certain element $\mathfrak{c}|\Psi\rangle \in H$. It is clear that $\mathfrak{c}$ depends on $\mathfrak{a}$, $\mathfrak{b}$ and, hence, $\mathfrak{a}|\Psi\rangle \pm \mathfrak{b}|\Psi\rangle = \mathfrak{c}(\mathfrak{a}, \mathfrak{b})|\Psi\rangle$. The exhaustive properties of dependence $\mathfrak{c}(\mathfrak{a}, \mathfrak{b})$ are given by Formulas (57) and (59) under notation $\mathfrak{c}(\mathfrak{a}, \mathfrak{b}) =: (\mathfrak{a} \oplus \mathfrak{b})$.

6.3.1. "Physics" of Superposition

Besides the essentially unphysical nature of the ($\pm$)-superpositions, i.e., "we cannot recognize them" ([12], p. 13), the primary and salient property of quantum addition is in the fact that, due to the group subtraction, it is possible to experimentally obtain a "quantum zero" in statistics from "non-zeroes'. With that, these "seem to be" positive, but there are "negative non-zeroes", i.e., negative numbers (Section 9.2). Subtraction manifests by the typical obscurations in interference pictures. S. Aaronson adds to this: "We have got minus signs, and so we have got interference" ([20], p. 220). No classical composition

$$w\varrho_1 + (1 - w)\varrho_2 \tag{60}$$

of non-zero statistics $\varrho_1$, $\varrho_2$ can provide a zero value since the zero will never be obtained via the $\cup$-unions. The same is true for the pre-superposition in isolated brace $(\Xi)_{\mathscr{A}}$, i.e., when one instrument is in question.

**Remark 12.** *One cannot help but mention yet another counterexample to the superposition's "physicality": the (in)famous "quantum cat". Any combination of the dead and living animal is meaningless as a statement about new/nonclassical entity such as a "(half-)dead/alive cat" or such statements about particles as "their being neither here nor there but everywhere", especially with the stress on "at the same point in time" (see pr. **I**). It makes absolutely no sense to add (allegedly in accord with the character +) to each other the nature's phenomena and notions that have not yet been created and are dynamical ("alive") at that.*

- **What is being added is states, not their denominations** or verbal descriptions of envisioned *("fantasized", "fantastic phantoms" ([12], p. 15))* physical properties *such as spin up/down or dead/alive. Cf. [151] (pp. 134 (!), 135).*

*The "cat-box open" is a click, not state, without a notion of "a cat". Accordingly, the word combination "the quantum objects exist in "superpositions" of different possibilities" (a representative excerpt from the literature) is at most an interpretative allegory (Section 10) without physical and mathematical content. That is to say, strictly speaking,*

- *No quantum (micro)system has ever been/dwelled in any state, much less in a superposition one, and much less at an instant t. Ludwig, on pp. 16 and 78 of the book [58], insists that it is a "myth" and "a fairy tale, ... the very widespread idea that each microsystem has a real state ... represented by a vector in a Hilbert space", and M. Nielsen remarks in [21] that "Saying $0.6|0\rangle + 0.8|1\rangle$ is simultaneously 0 and 1 makes about as much sense as Lewis Carroll's nonsense poem* Jabberwocky: ...". *K. Svozil does also underscore that "'coherent superpositions' just correspond to improper, misleading representations of non-existing aspects of physical reality. They are delusive because they confuse ontology with epistemology" ([152], p. 26).*

*The meaning of the word "add" is still being created, including an implementation at objects to be thought of as the "atomic irreducible" entities—the numbers (Section 7).*

*T. Maudlin notes on p. 133 of the work [151]: "Our job ... is to invent mathematical representations ..., rather than merely linguistic terms such as "z-up." ... we are in some danger of confusing physical items with mathematical items" (italics supplied). Here is an example of confusion. If we are going to measure the z-spin in one of the $(\rightleftarrows_x)$-beams in a Stern–Gerlach device, then why and when does this "observable" certainty—say $|\rightarrow_x\rangle$—get turned into a $(\uparrow\downarrow_z)$-uncertainty $|\uparrow\rangle + |\downarrow\rangle$? (see [153] (p. 232)). However, what if we are about not to do this? We come up against the question:*

- *What does one mean by an equal-sign $=$ in the orthodox notation $|\rightarrow\rangle = |\uparrow\rangle + |\downarrow\rangle$?*

*Which state does the system "intend" to fall into: the z-uncertainty or the x-determinacy? Which of the states is it in, after all? Examples to the "physicality of states" may be continued endlessly [154].*

A statement about QM-superposition (without $\mathbb{C}$-numbers) as a non-independent axiom can be found in the book ([36], p. 108) but arguments given there are circular: ⌈Hilbert space⌉ ↣ ⌈quantum logic of propositions⌉ ↣ ⌈superposition principle⌉. Similarly, in the works [122] and [72] (p. 164), all of that is "derived" from modular lattices [155]. However, the lattices are known to enter QM from the Hilbert space structure and, on the other hand, the purging quantum rudiments of such a space' axiomatics constitutes Birkhoff's 110-th problem ([155], p. 286). Note also that, in connection with the formal logic approaches to the theory construction [9,36,72,111,122,156,157], the issue of vindicating the *matters* that this logic deals *with* (logic of what?) [58,93,158] should not be neglected. What we mean here is the questions on logic: of propositions? [105,106] of relations? of (math-logic) classes/sets? [120] of phenomena/properties? (which ones?) of quantum/classical events? ...? "For example, would one have to develop a quantum set theory?" ([110], p. 17). "If by "logic" we mean something like "correct reasoning," then it would make no sense to think of logic as "just another theory."" ([73], p. 258). The more abstract micro-events and Boolean logic we have used in metamathematical reasoning at the moment ([87], pp. 189, 193) contain nothing that depends on classical physics. That is, quantum foundations do not require [58] a different quantum/non-classical logic. See also [74] (p. 29).

6.3.2. When and What Is Non-Commutativity?

Yet another fact that results from the above constructs is that the availability of a superposition math-structure (58) reflects the presence of at least *two* $\mathscr{A}$, $\mathscr{B}$ with *non-coinciding* families of eigen-primitives $\{\underline{\alpha}_s\}$, $\{\underline{\beta}_k\}$. This consequence of pt. $\mathbf{R}^{\bullet\bullet}$ should be particularly emphasized since it will manifest in the *non-commutativity* of operators $\hat{\mathscr{A}}$ and $\hat{\mathscr{B}}$ in the future. Although the present work does not get to operators as a mathematical structure, it is clear that the emergent eigen-states and spectra have a direct bearing on them. In this context, the "commuting instruments" $\{|\alpha_1\rangle, |\alpha_2\rangle, \ldots\} = \{|\beta_1\rangle, |\beta_2\rangle, \ldots\}$ can be treated, roughly speaking, as coinciding because this fact is independent of the specific spectra $\{\lfloor\alpha_1, \lfloor\alpha_2, \ldots\}$, $\{\lfloor\beta_1, \lfloor\beta_2, \ldots\}$ assigned to them. If they differ, this is merely a different (numerical) graduation of the spectrum scale. It is the same for all instruments, and its length is the parameter D.

Notice that the definition of an $\mathscr{A}$-observation is not different from the formal assignment of the family $\mathfrak{T}_{\mathscr{A}}$ (pt. **O** and (8)), which is why the non-coinciding sets $\mathfrak{T}_{\mathscr{A}}$, $\mathfrak{T}_{\mathscr{B}}$ do always exist. This provides a kind of abstractly deductive existence's proof for the non-commutativity, QM-interference—see Section 6.5 further below—and for the utmost low-level finality of QM altogether [13,33,93]. The whys and wherefores of theory do not require invoking the physical conceptions; cf. [10] (p. 2).

Of no small importance is that this point entails an independence of the (existence/presence of) classical physics or of its formal deformation, which are yet to be created from the quantum one (cf. a selected thesis on page 15). In particular, no use is required of the notion of a certain pretty small—again the classical/physical term—quantity, i.e., the Plank constant $\hbar$ ([123], Section 6.5). (Parenthetically, no numerical value of this constant matters here; it is not dimensionless and its zero limit is not meaningful.) What is more, the quantum paradigm (17)–(19) tells us that the classical description begins, i.e., we do create/introduce, with the notions of a micro-event's average and of time, whereas these conceptions are still absent at the moment and in the present work. Similarly for the notions of locality, causality, the classical event, and the classical object.

### 6.4. Physical Properties

Now, the "general physics" $\langle\!\langle \mathcal{S}, \mathbf{M}, \ldots \rangle\!\rangle$ is mathematized into representations (50) of states $|\Xi\rangle$ of system $\mathcal{S}$. There is, however, an ambiguity, the source of which is the fact that the natural/classical language also lays claims to a similar formulation. This refers to the belief in the existence of mathematics ("bad habit" [3]; see also [38], [58] (p. 122), and [159]) that describes $\mathcal{S}$ as an individual object with properties regardless of observation; an observation that is not a *functioning* attribute of the mathematics itself. In classical description, it is specified by definitions: point $\mathcal{P}$ of a phase space, $(q, p)$-coordinatization of the point (manifold), and statistical distribution $\varrho(q, p)$.

On the other hand, quantum empiricism provides nothing more to us besides the ensemble brace and $|\Xi\rangle$-states (pt. **T**). Preordained definienda with physical contents are unacceptable, i.e., $\mathcal{S}$ should not be conceived as "something with *physical* properties" or as an "*individual* object"[93,94], [113] (p. 645). However, since the observational data (in the broadest sense of the word) may not originate from anywhere but a certain $|\Xi\rangle$-object, there should subsequently create:

(1) The very concept of physical objects and properties ([160], pp. 211–230);
(2) Their numerical values/characteristics, i.e., the "physical attributes of objects" ([131], p. 238; N. Bohr).

This is habitually referred to as elements/images of reality [27] (p. 194), [40] (Section 10.2), [94] (Section XIII.4.8)—Bell's "beables" [28]—or what we have been calling attributes of a physical system.

- "The very notion of 'phenomenon' or of 'the appearance of things,' ... is a cognitive and perceptual act of abstraction"

<div align="right">M. Wartofsky ([160], p. 220)</div>

That is to say, the physical phenomena per se do not exist [92] (p. 310), [127].

Indeed, the primary ideology of Sections 1.3 and 2.1 tells us that an invasion of physically self-apparent images into the theory should be avoided ([87], p. 69) because "quantum theory not only does not use—it does not even dare to mention—the notion of a "real physical situation"" ([27], p. 198; E. Jaynes). Continuing a quotation from R. Haag on page 5, one requires "the renunciation of the absolute significance of conventional physical attributes of objects" ([131], p. 238; N. Bohr) and of concomitant and accustomed logic in reasoning. In fact, we are led to (re)build the language of the classical description. Therefore, everything, with no exceptions, should be created mathematically: coordinates, momenta, energies, optical spectra, device readings, lengths/distances and time, extension and lifetime of objects, the language of particles, their number/numeration (Fock space),

(in)discernibility/individuality (bosons/fermions), the notions of a subsystem of system $\mathcal{S}$ (see (23)), and even a notion of the physical rigor (in reasoning), etc.

Degrees of freedom, the concepts of the field/body/mass/inertia/interaction, the numerical labeling the space-time continuum, Newtonian mechanics with its equations and the concepts of the force, interaction, and the causality of classical events, thermodynamics, the very term "the classical state", the numerical labels of the space-time continuum and numerical forms of what is known as the classical reference frames—coordinates on manifolds—need to also be created. Once more to underscore, the numerical forms of the classical space coordinates and the time (e.g., the metric tensor $g_{\alpha\beta}(x)dx^\alpha dx^\beta$) have a quantum empirical origin. The latter fact is required for carefully posing the questions of quantum gravity, and it should be noted in passing that the simultaneity is an ill-defined term not only in the (general) relativity theory; in QT it is even worse. In common with the simultaneous measurability, this term appears to have come from the classical framework, which is why it is illegal as a quantum-theoretical primitive (pr. I and [87]).

6.4.1. Waves/Particles?

The concept of a (non-elementary) particle, which is conceptually close to the notion of a subsystem/part, is also a physical convention and can only arise from the $|\Xi\rangle$ or its models: Bose-condensates, deformation excitations in crystal lattices, quasi-particles in a superfluid phase, quantum theories of various fields (relativistic or non), and more. Here, by particle we mean the classical kinematic conception. "What do we detect? The presence of a particle? Or the occurrence of a microscopic event?" wondered R. Haag (2013). H. Zeh and G. Ludwig do answer: "There are no particles in reality" [161], "we must abandon the notion of a microscopic "object", one to which we have been accustomed" ([87], p. 69).

Clearly, the QFTs is a subclass of QM rather than its extension; not that we have yet given a definition of QM. In particular, it is common knowledge in QFT that there is no logical way to distinguish a particle from a certain state—normally, a vacuum excitation. One word should therefore be used for both. To this extent, the familiar "dualism of ... the *particle picture* and the *wave picture*" [78] (Section 7.2), [91], [108] (p. 28) simply disappears. K. Popper is rather emphatic concerning this "problem" and puts it, in their "thirteen theses" [108], quite rightly in the following terms: "*the great quantum muddle*", "alleged "duality" or "complementarity", ... *this* kind of "understanding" is of little value", "has not the slightest bearing on either physics, ...", "fashionable among quantum theorists, ... a vicious doctrine", and the like. As a matter of fact, both the particles and waves are the classical terms [61] and, in quantum language, they turn into the derivatives of the concepts of state and mixture (23).

- Like waves, *the particle is already an appearance*—an observable one (phenomenology, derivative)—rather than a logical primitive or a fundamental substance, which is why it may not exist [161] prior to theory's principles ([126], p. 762 (!)). Paraphrasing Heisenberg, Haag remarks, in the context of their "event theory", that "Particles are the roof of the theory, not its foundation" ([88], p. 300).

Both these notions should be superseded by a mathematics of clicks.

The f-statistics also falls under observable quantities, and constant D, if declared finite, is an example of an already created characteristic: the dimension of a state space to come. A tensorial structure of this space—compound systems—also pertains to the physical properties, but we do not touch upon this point here. As an aside, this compositional structure will provide the means of distinguishing the aforementioned models under $D = \infty$.

In other words, the logic of the above constructs prohibits not only endowing the phraseology "internal state of an individual object $\mathcal{S}$" and "the system is in a (definite) state [4,58,93,94] with a meaning but also indirectly using its numerical forms. That would work in the circumvention of empiricism, assuming the a priori availability of mathematical structures that do not rest on the state space. L. Ballentine remarks in this regard: "the habit of considering an individual particle to have its own wave function is hard to break"

([34], p. 238); cf. "To speak of a single possible initial apparatus state is pure fantasy" ([80], pp. 241–242; N. Graham).

*6.5. Interference*

Let us go on with comments as to involving the *physics*-related argumentation to explicate the quantal behavior. We have already mentioned above that for this purpose there is simply no language of physics (Sections 2.1 and 6.4) and of mathematics yet (Sections 2.3 and 5). That is why analogies of this sort are not only deceptive but must be prohibited for exactly the same reasons that accompanied boxes (5). The typical examples in this connection are the simultaneous measurability mentioned above and the two-slit interference [5,17].

First and foremost, the two cases—whether one or two slits are open—are utterly "different experimental arrangements" [153] (p. 236), [64] (p. 58):

$$\langle\!\langle \mathcal{S}, \mathbf{M}, \ldots \rangle\!\rangle' \neq \langle\!\langle \mathcal{S}, \mathbf{M}, \ldots \rangle\!\rangle'' \,.$$

There is nowhere to seek a means of their comparison or the transference of one into another ([153], p. 236). Nonetheless, the classical approach, when opening another slit $\langle\!\langle \mathcal{S}, \mathbf{M}, \ldots \rangle\!\rangle_2'$ together with the first one $\langle\!\langle \mathcal{S}, \mathbf{M}, \ldots \rangle\!\rangle_1'$, does literally envision properties for $\langle\!\langle \mathcal{S}, \mathbf{M}, \ldots \rangle\!\rangle''$ (see Section 6.4). In doing so, the transference method itself—"addition of the two 1-slit $\langle\!\langle \mathcal{S}, \mathbf{M}, \ldots \rangle\!\rangle'$-physicae" by the rule of arithmetical addition of statistics (60)—is meanwhile considered self-apparent. Thus, natural questions arise, such as "why/where are the zeroes coming from, they should not be there". In accordance with the aforesaid, everything here is erroneous, including the "natural" questions. There are no rules at the outset whether (non)classical and even quantum, just as there is no addition per se. An a priori assumption that stem from the obvious images for $\langle\!\langle \mathcal{S}, \mathbf{M}, \ldots \rangle\!\rangle_1'$ and $\langle\!\langle \mathcal{S}, \mathbf{M}, \ldots \rangle\!\rangle_2'$ is actually a declaration of the physical properties for $\langle\!\langle \mathcal{S}, \mathbf{M}, \ldots \rangle\!\rangle''$, but they do not follow from anywhere [17], [64] (p. 55), [162]. The (illegal) assumption of the "negligible effect of which-slit detectors" were mentioned on p. 28 is identical with a declaration of a physical property, as well as a solenoid's switch-on/off in the Aharonov–Bohm effect.

Taken alone, the $\varrho$-distributions—separate for $\langle\!\langle \mathcal{S}, \mathbf{M}, \ldots \rangle\!\rangle_1'$ and $\langle\!\langle \mathcal{S}, \mathbf{M}, \ldots \rangle\!\rangle_2'$—are entirely correct observational pictures, but introducing the rule (60) is indistinguishable from "invention" of physics—a logically prohibited operation. As Slavnov had put it, "to invent the physical exegesis of a … mathematical scheme" ([76], p. 304). "Our custom of seeing classical mechanics as a no-nonsense description of 'reality as it is' does not seem to be justified. This custom is actually based on a confusion of categories …" (W. de Muynck ([4], p. 89)). In other words, the mere fact of non-adherence to this rule means that the grammatical conjunction of the verbs "to understand/deduce" with the noun "micro-phenomena" is unacceptable even linguistically. It is the point **T** that prohibits predefined (classical) semantics, and this was faithfully summarized by C. Fuchs: "badly calibrated linguistics is the predominant reason for quantum foundations continuing to exist as a field of research" ([2], p. xxxix). To figure out or deduce (from mathematics) that quantal phenomena are unfeasible ([5], p. 111) and are *"absolutely* impossible, to explain in any classical way" (quotation by Feynman). Just as with the elucidation of the nature of the quantum state on p. 23, any (circum-)classical justification or even motivation are guaranteed to fail here since they are based on significant and implicit assumptions.

The classical theory is a theory of *observational* objects with *observational* properties expressed by *observational* numbers. We possess none of the three items required to create the quantum (= correct) description (Section 2). The adjective "observational" itself is a linguistic notion of the classical vocabulary (Section 2.2). Accordingly, the description can only be changed "to describe in newly created terms". A. Leggett notes [95] that which is understood as common-sense should also be changed (see also [12] (p. 10)). The reason is clear.

- Common-sense operates—and that is perfectly normal—with observational categories rather than with structureless "microscopy" (9) and ∪-abstractions of Section 5.1;

cf. Bohr's correspondence principle [78]. In effect, we have dealt with a "fundamental chasm" between the right description—"what is really going on?" ([12], p. 12)—and our ability to give a (naturally speaking) explanation in terms of these categories:

> "All our intuition, all our sense of what constitutes concreteness are based upon our everyday experience, and the terms used to describe a phenomenon concretely are necessarily drawn from that experience. There is no indication that such a language could be used without contradictions for phenomena which are as far removed from it as those of microscopic physics" (A. Messiah. *Quantum mechanics*).

The total dismissal of this has to be at the heart of quantum reconstructing.

6.5.1. Detector Micro-Events

For similar reasons, we may not think or envision that a particle in an interferometer "flies through the slit", "has (not) arrived", "is located somewhere in the region of space" ([26], p. 7)], "here, not there", "now/later", that "the choice of a detector has been delayed" ([27], Wheeler), [62], or that a "photon ... interferes ... with itself" ([26], p. 9), and that, generally, "something is flying along a trajectory", and "something" is a particle at an intuitive understanding. Cf. Dirac's description of "the translational states of a photon" in Section 3 of [26].

- "Photons are just clicks in photon detectors; nothing real is traveling from the source to the detector" (ascribed to A. Zeilinger),

and this point is supported by all the known varieties of interferometers. There has to be an amendment here.

The clicks themselves are not the clicks *of photons/particles*, just "merely clicks". "[T]he click is no ... produced by a particle. ... nothing takes place in the source that could be a cause of the click ..., the genuinely fortuitous click comes without a cause and has no precursor" ([126], pp. 758, 765). Nothing really interferes inside interferometers, nor is anything superposed/reinforced. For example, the fact that the path of "photons" is not represented by trajectories was impressively demonstrated with the nested Mach–Zehnder experimental setup in the work [163]. Asking "where the photons have been" [163] is also the matter of a certain $\underline{\alpha}$-distinguishability. An interferometer—the entire installation—should be perceived as nothing more than a black box $\langle\!\langle S, \mathbf{M}, \dots \rangle\!\rangle$—the box (5)—outside of space and time. This is a kind of irreducible element that produces the only entity—distinguishable $\underline{\alpha}$-events, and no other. The box contains no "flying particles". Exempli gratia, none of the words in the typical sentence "photon propagates a definite path" are well-defined. Any assessment of the screen flashes observed within the interferometer, e.g., "is zero statistics possible in any spot?", lacks meaning until the theory's numerical apparatus is presented.

**Remark 13.** *Thus, Young's interference of the light beams (1803) is* inherently *the quantum not the classical effect: a micro-events' accumulation is usually termed as the light intensity. The classical electromagnetism and optics, in an exact sense, do not explain, only describe, the phenomenon quantitatively with the use of the numerical concepts of the positive and, which is important, the negative values of observable strength-fields $\vec{E}$ and $\vec{H}$. (The negative numbers are specifically discussed further in Section 9.2.) Accordingly, operations of their addition/subtraction "rephrase" the effect in words "superposing, suppressing, waves, intensities"', and we call this "the explanation". In a quantum way of looking at it, all of these concepts are not yet available, and the phenomenon per se is no more than statistics of the "positively accumulative" quantal clicks:*

- *There are no particles, waves, or subtractions there.*

*The same macroscopic effect, which is visible with the naked eye and "explainable by waves", would take place if we had a "laser" of, say, mono-energetic very slow electrons (a proposal for*

*experimentalists). To put it more precisely: a gun or emitter of something we envision as the "tiny bodily formations" the electrons, molecules, microbes, and the like. It is self-evident that we would have seen the wave-like manifestation even from a single slit.*

Criticism of the typical (a common event-space) examination of the two-slit experiment [164] is already abundant in the literature. See, for example, the works [17] (Sections V.1, VI.1–2), [64] (pp. 55–58), [66] (Ch. 2), and [121] (p. 93), [162] (!), [165].

By way of continuing the last sentence in Remark 3, we add the following. To force an electron-click to happen each time at the same (or predictable) place is no different from "completely describing *everything* that we have", i.e., from the precise setting of "the same" and of macro-context $\langle\!\langle \mathcal{S}, \mathbf{M}, \ldots \rangle\!\rangle$. It is amply evident that this is a manifest absurdity. Hence, it immediately follows that *the unpredictability of microscopic events must exist in principle* and macro-determinism may be only an idealization through a (math) model: the model description of the $\langle\!\langle \mathcal{S}, \mathbf{M}, \ldots \rangle\!\rangle$ itself.

Summing up, it is not the quantum interference that requires interpretative comprehension but its classic "roughening". In other words, a scheme that latently presumes the rule (60) of extrapolation of what is observed in macro and micro ([160] (!), last sentence on p. 101). It is this scheme and not the quantum approach that contradicts the logic and experience. Pauli characterizes this as habits "known as 'ontology' or 'realism'". More than that, the chief component of constructs—$\lceil$observation $\rightarrowtail$ state$'\rceil$—is cast out and replaced with (19) under such a transformation. The `DataSource` object (p. 31) begins to be identified with *observational* and *numerical* characteristics (see a paragraph preceding Remark 4), while the logic of the micro-world requires precisely distancing these two concepts, with no need for the characteristics themselves.

Thus, we should not be deriving the physics of one phenomenon from another ([58], p. 92) and making (super)generalizations, as soon as the incorrectness of the previous derivation method was established.

- Quantum-mathematics is not a physical theory—and that is its distinguishing feature—but rather a single syntactical (meta)*principle of forming the mathematical models being subsequently turned into* (the physical) *theories*. This principle is not subject to any physical validation.

Scott Aaronson was likely the first to advance the line of thought about non-physicality. On page 110 of the book [20], he writes that "it's *not* a physical theory in the same sense as electromagnetism or general relativity ... quantum mechanics sits at a level *between* math and physics ... *is the operating system* ...". Fuchs–Peres provoke: "quantum theory does *not* describe physical reality" ([43], p. 70).

To create the models, we already have a good deal of *latitude*: the toolkit $\mathcal{O} = \{\mathscr{A}, \mathscr{B}, \ldots\}$, the parameter D, the families $\{\mathfrak{T}_{\mathscr{A}}, \mathfrak{T}_{\mathscr{B}}, \ldots\}$, numbers $\{\varrho_s\}$ of mixtures (23), and—thanks to the notion of covariance **III**—spectra, a structure of a group, and the concept of (different) representations of a mathematical structure. This liberty will be subsequently augmented with the key notions of a mean and of time $t$ and also with the composite systems, the classical Lagrangians/Hamiltonians, their symmetries, gauge fields, and phenomenological constants. This is what is currently termed the quantum phenomenology or a *quantization procedure* of the classical models: the path-integrals, $S$-matrix, etc.

All that remains is to examine the numerical constituent of quantum mathematics. The further strategy (Sections 7–9) lies in the fact that the numbers need to be created at first as a theoretical concept—arithmetic—and then as the "numerical values for observable quantities"—the observations numbers. Sections 9.1 and 9.2 contain some more explanations along these lines.

## 7. Numeri

By number we understand not so much a multitude of unities, as the abstracted ratio of any quantity to another quantity of the same kind, which we take for unity—I. Newton (1707)

### 7.1. Replications of Ensembles

In connection with the emergence of a group, the numerical representation of brace also undergoes a change since the "doubling" of a semigroup into a group through adjoining the inversions deprives coordinate $\mathfrak{a}$ of its distinction in comparison with the inversion $-\mathfrak{a}$. Given the involution

$$- (-\mathfrak{a}) = \mathfrak{a} \, , \tag{61}$$

it makes no difference what to call an element and what to call its inversion in the pair $\{\mathfrak{a}, -\mathfrak{a}\}$. This doubling is formally known as a symmetrization of the commutative associative law (monoid) [166]. Curiously, under commutativity and associativity ([167], Section 1.10), the solution to the problem of embedding is unique ([166], pp. 15–17), and otherwise, no solution, in general, exists. There exist the classes (Mal'cev (1936)), which are not axiomatized by finitely many $\forall$-formulas ([168], pp. 216–217).

The aforesaid is best demonstrated by another way of "numeralizing" the empiricism, which is realized as the infinite replication of finite ensembles

$$\{\{\underline{\Psi}\}_n \{\underline{\Psi}\}_n \cdots \} = \{\{\underline{\Psi}\}_n\}_\infty =: \{\underline{\Psi}\}_{n\infty} \, . \tag{62}$$

That is, empirically, any infinite ensemble is thought of as created by repetitions (copies) of the finite objects $\{\underline{\Psi}\}_n$. It is in this sense, and in this sense alone, that one should read the writing $\Sigma \rightsquigarrow \infty$ for the infinity postulate (14) because, at the moment, we possess neither the mathematics nor the topological concepts, such as a passage to the limit $\lim_{\Sigma \to \infty}$. For example, the expression $\Sigma \times \infty$ can be viewed as a conjunction of the actual and potential infinity [105,169]. Simply put, the case in point is not an *axiomatic* act—an imposition of the *math*-existence condition for numbers $\{\mathfrak{f}_j\}$ in (14). The latter has been typically criticized as an idea of the stable limiting frequencies in QM [8] (pp. 15, 183, 211, ...), [23], [170] (pp. 97–99), [171]. Rather, we claim that the only way to consistently incorporate the language notions of infinity and of the finite (observational) numbers in theory—"to cross an abyss" (Poincaré)—is the above semantics and correspondence between symbols $\{n_j, \mathfrak{f}_j; \Sigma, \rightsquigarrow, \times, \infty\}$. See also subsection "StatLength and infinity" in the work [6].

In turn, the above-mentioned copies $\{\underline{\Psi}\}_n$ are replications of the atomic primitive $\{\underline{\Psi}\}_1$. Replication is thus an operation of the same significance as $\cup$ and $\uplus$. With this point, the $(\Xi)$-brace is characterized by the "numerical" combination

$$(35) \quad \longmapsto \quad \big\{ [n_1\infty, m_1\infty], \ [n_2\infty, m_2\infty], \ \ldots \big\} \ \rightleftarrows \ (\Xi)$$

(indices label the $\underline{\alpha}_s$-primitives), which has been created from the unitary brace by the scheme

$$(34) = \begin{pmatrix} \{\{\underline{\Psi}\}_{\infty'}\{\underline{\Phi}\}_{\infty''}\} \\ \downarrow \ \downarrow \ \downarrow \ \downarrow \\ \{\underline{\alpha} \cdots \cdots \underline{\alpha}\}_\infty \end{pmatrix} \quad \longmapsto \quad \begin{pmatrix} \{\{\underline{\Psi}\}_{n\infty}\{\underline{\Phi}\}_{m\infty}\} \\ \downarrow \ \downarrow \ \downarrow \ \downarrow \\ \{\underline{\alpha} \cdots \cdots \underline{\alpha}\}_{(n+m)\infty} \end{pmatrix} \quad \longmapsto \quad [n\infty, m\infty]\underline{\alpha} \, . \tag{63}$$

The semigroup union $(\Xi') \uplus (\Xi'')$ is then conformed with the writing

$$\big\{ [n_1'\infty, m_1'\infty], \ [n_2'\infty, m_2'\infty], \ \ldots \big\} \uplus \big\{ [n_1''\infty, m_1''\infty], \ [n_2''\infty, m_2''\infty], \ \ldots \big\} =$$
$$= \big\{ [(n_1' + n_1'')\infty, (m_1' + m_1'')\infty], \ [(n_2' + n_2'')\infty, (m_2' + m_2'')\infty], \ \ldots \big\} \, . \tag{64}$$

Moreover, the n-, m-quantities may be freely thought of as real ones due to the $\mathbb{R}^2$-continual infinity of ensembles proven above (Section 4). The empirical rationale of this is apparent; namely, fractions of the arbitrarily large ensembles $\{\underline{\Psi}\,\underline{\Psi}\cdots\}$.

This way of matching the infinity with $\Sigma$-postulate automatically inherits translation of associativity/commutativity because the "percentages", such as $s$ and $w$, just as the rules (44) and (45) themselves, do not even emerge. There, these numbers originated from $\Sigma$-postulate, but it, in turn, was *demolishing the pair* $(\varkappa, \mathfrak{S})$ *itself* in (43): $\mathfrak{S} \to \infty$. It is clear

that, according to (64), the semigroup structure $\mathfrak{G}$ is also inherited, turning into the addition of the numerical pairs

$$(n', m') \oplus (n'', m'') = (n' + n'', m' + m'') . \tag{65}$$

Returning to the group, we observe that the "negative symbols" $(-n, -m)$ might be initially taken as the semigroup $\mathfrak{G}$ being duplicated, with equal success and with the same arithmetical addition $\oplus$, while the positive $(n, m)$ could be thought of as inversions thereof.

Summing up, let us specify the rules of passing to the numerical representations

$$(34) \quad \Longleftrightarrow \quad \{\pm\overset{\Psi}{\mathsf{p}}, \pm\overset{\Phi}{\mathsf{q}}\}\underline{\alpha}, \qquad (\mathsf{p}, \mathsf{q}) \in \mathbb{R}^2 \tag{66}$$

and, to avoid ambiguity, replace the binary-composition symbols $\{\uplus, \pm\!\!\!\downarrow\}$ with a new symbol $\hat{+}$ for objects (66):

$$\{\overset{\Psi}{\mathsf{p}}, \overset{\Phi}{\mathsf{q}}\}\underline{\alpha} \,\hat{+}\, \{\overset{\Psi}{\mathsf{n}}, \overset{\Phi}{\mathsf{m}}\}\underline{\alpha} .$$

The previously dropped primitives $\underline{\Psi}$, $\underline{\Phi}$ have been restored here since they will be further needed for the theory's covariance (Sections 7.4 and 7.5), although they are still unnecessary at the moment.

It is not accidental that we spoke of "numerally labeling" the brace (p. 22) since the question of arithmetic on them had not yet arisen. Although $\mathfrak{f}$-statistics—the real $\mathbb{R}$-numbers—are already involved, their use was based on an accustomed perception of the number. In accordance with pr. **II**, the numerical formalization of ensemble empiricism should be considered in greater detail.

### 7.2. The Number as an Operator

Let us take up the "process of manufacturing" the numbers (pr. **II**). We begin with the classical simplification

$$\mathfrak{A} = \{\{\underline{\Psi}\}, \{\underline{\Psi}\,\underline{\Psi}\}, \{\underline{\Psi}\,\underline{\Psi}\,\underline{\Psi}\}, \{\underline{\Psi}\,\underline{\Psi}\,\underline{\Psi}\,\underline{\Psi}\}, \dots\} , \tag{67}$$

and the notion of the number does not yet appear in any form.

The mathematical abstracting the observation micro-acts is an employment of the operation $\cup$ and of its closedness (see Section 5.1). For example, $\{\underline{\Psi}\} \cup \{\underline{\Psi}\,\underline{\Psi}\,\underline{\Psi}\} = \{\underline{\Psi}\,\underline{\Psi}\,\underline{\Psi}\,\underline{\Psi}\}$. All the symbols in (67), as well as the character $\cup$, is of course merely a convention, and they may be changed. By writing (67) in symbols such as $\{a, b, c, d, \dots\}$ and $+$, this set should be supplemented with identities as $a + b = c$, $b + b = d$, $\dots$, i.e., with a binary construction $+$. Then, (semi)group and commutative superpositions arise. Though note that introducing the numbers at this point—even if only as symbols—is not necessary. It would reduce to re-notating the set's elements, to be precise. However, the empirical description calls for their unification, as manifested in the numerical notation as $\{\underline{\Psi}\} =: 1\{\underline{\Psi}\}$, $\{\underline{\Psi}\,\underline{\Psi}\} =: 2\{\underline{\Psi}\}$, $\dots$. It is precisely this pattern that was implicitly kept in mind in procedures (34), (35), (62), and (63), i.e., when introducing the numbers $\mathsf{n}$ by means of replication of finite or infinite ensembles:

$$\{\underline{\Psi} \cdots \underline{\Psi}\}_{\mathsf{n}} \iff \mathsf{n}\{\underline{\Psi}\}, \qquad \{\{\underline{\Psi}\}_{\infty} \cdots \{\underline{\Psi}\}_{\infty}\}_{\mathsf{n}} \iff \mathsf{n}\{\underline{\Psi}\} .$$

The symbol $\iff$ should read here as "the same thing as". Clearly, the very idea of the conjunction of the two entities—empirical brace (34) and the concept of a (quantitative and ordinal) number (Sections 5.1 and 5.2)—is not otherwise implementable. That is to say:

- We have no any means of translating the aggregates of micro-acts $\overset{\mathscr{A}}{\dashrightarrow}$ (i.e., macro-observations **M**) into the numerical language other than through the counting of things [172], i.e., through the natural-language notion of the "quantity of something":

$$
\begin{array}{c}
\lfloor\cdots\downarrow\downarrow \;\; \mathscr{A}\text{-transitions} \;\; \downarrow\downarrow\cdots\rfloor \\
\Downarrow \qquad \Downarrow \\
\lceil\text{quantity of}\rceil \; \lceil\text{something}\rceil \quad (\text{replication}) \\
\Downarrow \qquad \Downarrow \\
\lceil\text{numbers}\rceil \; \lceil\underline{\Psi}\text{-primitives, ensembles}\rceil \\
\searrow \quad \swarrow \\
\mathsf{n}\{\underline{\Psi}\}
\end{array}
\tag{68}
$$

Heisenberg stresses an obligatory relationship with "the natural language because it is only there that we can be certain to touch reality" ([98], pp. 201–202). Otherwise, the quantitative theory would have nowhere to originate even at the level of calculating the natural entities by the $\mathbb{N}$-number tokens. It may be added that arising the numbers is a permanently present (innate) process of creating the thought objects by an abstraction in the human brain: the mental suppressing/neglecting of the inessential and identifying the distinguishable entities—perceptual objects—irrespective of their nature ([84], "forming collections, ... putting objects together"; pp. 99, 251). It is something that humans do all the time without even realizing they are doing it. This process, say,

$$
\lceil\text{language, words}\rceil \cdots \rightarrowtail \{\text{sheep}, \underline{\Psi}, \text{verb}, \underline{\Psi}, \text{theory}, \dots\} \rightarrowtail
$$
$$
\{\text{a sheep, a } \underline{\Psi}, \text{a verb}, \dots\} \cdots \rightarrowtail \lceil\text{something/thing/}\dots\text{/Stücke}\rceil \cdots \rightarrowtail
$$
$$
\{\bullet\text{Stück}, \bullet\text{Stück}, \bullet\text{Stück}, \bullet\text{Stück}, \bullet\text{Stück}, \dots\} \rightarrowtail \{\bullet, \bullet, \bullet, \bullet, \bullet; \text{Stücke}\} \rightarrowtail
$$
$$
\{1, 1, 1, 1, 1; \text{Stücke}\} \rightarrowtail 5\text{Stück} \rightarrowtail 5 \; \sout{\text{Stück}} \rightarrowtail 5 \rightarrowtail \lceil\text{abstraction 5}\rceil \,,
$$

is akin to Cantor's concept of a Menge ([136], Ch. 1, Section 1.1) and has *no* the mathematical (math-logic) nature. Rather, the math of numbers does originate from it [84] (Ch. 3); see also [173] (Section 2.4.5.1 ARITHMETIC).

Incidentally, the "inessential and identifying" just mentioned have the nature just like the "the same" in Section 5.4. It is with these notions—a key feature of the natural/physical language and of speech—that any abstracting begins: the "abstracting from ...".

On the other hand, the numerical tokens are "affixed" not only to the "atom" $\{\underline{\Psi}\}$ but also to other objects, *any* at that; for more details, see Remark 16 further below. Therein lies the primary meaning of this still proto-mathematical concept [174] ("Psychologie du nombre"). One might even say, a definition according to which this notion has been conceived ("20 Stück", "half an hour" ...) and is being used universally. Here are a few examples:

$$
\{\underline{\Psi}\,\underline{\Psi}\,\underline{\Psi}\} \equiv 3\{\underline{\Psi}\}, \qquad \{\underline{\Theta}\,\underline{\Theta}\,\underline{\Phi}\,\underline{\Phi}\} \equiv 2\left(\{\underline{\Theta}\} \cup \{\underline{\Phi}\}\right)
$$
$$
\{\underline{\Phi}\,\underline{\Psi}\} \overset{2}{\rightarrowtail} \{\underline{\Phi}\,\underline{\Psi}\,\underline{\Phi}\,\underline{\Psi}\}, \qquad a \overset{3}{\rightarrowtail} 3a, \qquad c \overset{1}{\rightarrowtail} 1c
\tag{69}
$$

Accordingly, in between the elements, there arise identities such as $2b \equiv 4a, 3a \equiv c, 1c \equiv c$. In other words, as we complete Simplification (67),

- While abstracting the empirical contents of the number entities into math-symbols, they should be defined as unary operations $\{\hat{1}, \hat{2}, \dots, \widehat{{}^{3}\!/\!_{4}}, \dots, \hat{\pi}, \dots\}$ that take action at $\mathfrak{A}$-set (67) as automorphisms: $\{\hat{2}b = \hat{4}a, \hat{1}c = c, \dots\}$.

That said, replication is formalized as an operator $\hat{n}$ with its numerical symbol $\mathsf{n}$:

$$
\psi \overset{\hat{n}}{\mapsto} \mathsf{n}\psi, \qquad \psi, \mathsf{n}\psi \in \mathfrak{A}, \qquad \mathsf{n} \in \mathbb{R} \,,
\tag{70}
$$

where $\psi$ is understood to be any (sub)ensemble/(sub)set. In the language of the ZF-theory, $\mathsf{n}\psi$ would be formally organized as an ordered pair $(\mathsf{n}, \psi) := \{\{\mathsf{n}\}, \{\mathsf{n}, \psi\}\}$ [134], where $\mathsf{n}$

is a cardinality of a set consisting of copies of the object/set $\psi$. We will refer to these facts as the implementation of a replication operator by numbers.

Attention is drawn to the fact that the case in point at the moment *is not* a math-logical *definitio/formalization* of the concept of a number, such as (106), but is an introduction of what is understood by number in the empirical/physical theory (**II**). For example, Chomsky says with regard to this point: "When multiplying numbers in our heads, we depend on many factors beyond our intrinsic knowledge of arithmetic" ([132], p. 3).

### *7.3.* QM *and Arithmetica*

We immediately observe the following properties.

The operators are applicable to each other. Being a family $\{\hat{n}, \hat{m}, \hat{p}, \ldots\}$, they are closed with respect to their composition $\hat{n}\,(\hat{m}\psi) = (\hat{n} \circ \hat{m})\psi = \hat{p}\psi$, and among them, there is an identical operator $\hat{\mathbb{1}}\psi = \psi$. The empirical meaning of the concept indicates a fractional portion of the ensemble (see (62)) requires that for each $\hat{n}$ there exists its inversion $\hat{n}^{-1}$. Hence, the composition of replications $\hat{n} \circ \hat{n}^{-1}$ must return the former "quantity": $(\hat{n} \circ \hat{n}^{-1})\psi = \hat{\mathbb{1}}\psi$. Therein lies "the actual meaning of division. ... this [operator] construction really corresponds to division" ([85], p. 37). By virtue of the fact that family $\{\hat{n}, \hat{m}, \hat{p}, \ldots\}$ provides automorphisms of the $\mathfrak{A}$-set, these operators entail the associative identities $((\hat{n} \circ \hat{m}) \circ \hat{p})\psi = (\hat{n} \circ (\hat{m} \circ \hat{p}))\psi$. This point is a property, and it has a proof [175] (Section I.1.2). The common nature of the replication and of the $\cup$-union also signifies that there are relations in place that mix the actions of the unary $\hat{n}$'s and the binary union of ensembles. At a minimum, suffice it to define the action of the replicator on a "$\cup$-sum" of replications. Clearly, the case in point is the distributive coordination of $\circ$ and $\cup$:

$$\hat{p}\,(\hat{n}\psi \cup \hat{m}\psi) = (\hat{p} \circ \hat{n})\psi \cup (\hat{p} \circ \hat{m})\psi\,.$$

We now observe that the indication of $\psi$ everywhere in the identities above loses the necessity, and the $\psi$-label becomes a semblance of a dummy index or the unit symbol (kg), which can be changed. As we omit it, the theory is freed of $\psi$ as a "calculation unit". Then, the last relation, as an example, acquires the form of a property between the operator n-symbols (70), if $\{\cup, \circ\}$ are replaced with the symbols of binary operations $\{+, \times\}$:

$$\mathsf{p} \times (\mathsf{n} + \mathsf{m}) = (\mathsf{p} \times \mathsf{n}) + (\mathsf{p} \times \mathsf{m})\,. \tag{71}$$

Supplementing this relation with other empirically determining properties, one infers that the *unary* operationality of $\hat{n}$-replications (70) is *indistinguishable* from the *binary* operationality on their n-symbols. The latter, in turn, acquires the multiplicative structure of a commutative group

$$\mathsf{n} \times \mathsf{m} = \mathsf{m} \times \mathsf{n}, \qquad (\mathsf{n} \times \mathsf{m}) \times \mathsf{p} = \mathsf{n} \times (\mathsf{m} \times \mathsf{p}), \qquad \mathsf{n} \times 1 = \mathsf{n}, \qquad \mathsf{n} \times \mathsf{n}^{-1} = 1\,, \tag{72}$$

and, as for the addition $+$, it is already binary and commutative due to properties of $\cup$ (Section 5.1):

$$\mathsf{n} + \mathsf{m} = \mathsf{m} + \mathsf{n}, \qquad (\mathsf{n} + \mathsf{m}) + \mathsf{p} = \mathsf{n} + (\mathsf{m} + \mathsf{p}), \qquad \mathsf{n} + 0 = \mathsf{n}\,. \tag{73}$$

Incidentally, the three-term multiplicative associativity relation in (72) has the same operatorial nature and origin as operations $\{\cup, \cup\}$ do in (39). We have already commented on the additive analog in this situation—a determinative structure of the binary operation—after Formula (57).

It is also clear that Rules (71)–(73) must be supplemented with the concept of a negative number

$$\mathsf{n} + (-\mathsf{n}) = 0\,, \tag{74}$$

for such numbers have been fully justified in the superposition principle.

After having acquired Properties (71)–(74)—call them *arithmetica*—symbols $\{n, m, \ldots\}$ turn into abstract numbers, although their operator genesis does not go away and is yet to be involved. This is where a full list of requirements for the concept of a real number should be added, and which have to do with ordering $<$, completeness/continuality, *and* their relations with Rules (71)–(74). We will assume that this is conducted axiomatically ([176], pp. 35–38), although the algebraic constituent of this "axiomatics", as we have seen, is not axiomatical but deducible from empiricism. Multiplication $\times$, and also the subsequent $\odot$-multiplication of $\mathbb{C}$-numbers (82), is a most nontrivial part in deriving the structure from "the arithmetic".

As an outcome, we reveal an essential asymmetry in the genesis of the standard binary structures $+$ and $\times$ (cf. [84] (p. 60)), and thereby a greater primacy of QM-consideration even over the (seemingly self-evident) arithmetic. Indeed, binarity may come only from operation $\cup$, which is primordially unique and, thereby, is inherited only to the one natural prototype—addition.

- Multiplication is not featured in the superposition principle, nor does it arise directly as a binary structure. The absence of a multiplication symbol in (58) and (59) is no accident.

The multiplication originates in the closedness of replications $\hat{n} \circ \hat{m}$, and they are required according to the **M**-paradigm (12). In effect, any non-operatorial way of introducing the n-numbers is not self-evidence for empiricism. An operator nature of the number is precisely that which gives rise to the second binary operation. Moreover, without such a comprehension of the number, the "linear nature" of QM (Section 8.1) will remain axiomatic at all times, and, as will be seen below, quantum foundations will be doomed to never-ending interpreting the mathematical symbols. However, the pure axiomatic declaration of Arithmetic (71)–(74) will, in one way or another, require a (reciprocal to (68)) treatment of the number in a context of "the quantity of what?", while its empirical pre-image always appears in the pair $\lceil$the quantity of$\rceil$ $+$ $\lceil$something$\rceil$. Another way to put it is:

- In the foundations of theory, there arises a predecessor/analog to the notion of a physical unit,

though the ultimate description is a description in terms of binary structures in Arithmetic (71)–(74). It is carried out by dropping/attaching the symbols such as $\psi$, which is a quantum generalization to the independence of a physical theory, from the measurement units.

Certainly, when formalized, the $\hat{n}$-replication and its binary n-twin become universally abstract. For example, the $\hat{n}$-operator (70) may be applied to the quantum case in which the object $\psi$ has already an internal structure associated with the presence of $\underline{\Psi}, \underline{\Phi}$-primitives. This changes no the essence of the matter. Another example is when numbers n give birth to really observable quantities. See also Section 5.3, Remark 16, and additional discussion in Section 9. Let us now proceed from the fact that the comprehension/relation of the number and its operator has been formalized as described above. This is "Axioms" (71)–(74).

As concerns the philosophical literature, the issue of numbers was likely discussed [177–179] (see also [172] and non-philosophical book [174]), and it would be appropriate to quote T. Maudlin: "... numbers: they can be added to one another, *perhaps* multiplied by one another, .... However, it is typically obscure what sort of *physical* relation these mathematical operations could possibly represent" ([151], p. 138; first emphasis ours, second in original). Cf. Einstein's remarks regarding the "concepts and propositions" and "the series of integers" on p. 287 in [180].

*7.4. Two-Dimensional Numbers*

A number in and of itself, as a replication operator, may be applied to any ensemble and to anything at all. However, in the quantum case, the "upper" primitives are attached to every "lower" $\underline{\alpha}$-event. These primitives, as was noted above, have to be discarded. At the same time, the minimal structure associated with the homogeneous array $\{\underline{\alpha}_s \cdots \underline{\alpha}_s\}$

as a whole is a unitary brace $\{\overset{\Psi}{n}, \overset{\Phi}{m}\}\underline{\alpha}_s$ containing two "upper" primitives $\underline{\Psi}$, $\underline{\Phi}$. Their order, however, is arbitrary there. That is to say, given $(n, m)\underline{\alpha}$, there are two quite equal objects $\{\overset{\Psi}{n}, \overset{\Phi}{m}\}\underline{\alpha}$ and $\{\overset{\Phi}{n}, \overset{\Psi}{m}\}\underline{\alpha}$ that are subjected to a replication. Each of them should be in a relationship (see Section 5.1) to any other brace (63), which is already apparent in the example of "one-dimensional" versions $(n, 0)\underline{\alpha}$ and $(n', 0)\underline{\alpha}$. We mean that for each pair $\{(n, 0)\underline{\alpha}, (n', 0)\underline{\alpha}\}$, there always exists the number $m$ such that $\hat{m}(n, 0)\underline{\alpha} = (n', 0)\underline{\alpha}$, i.e., $m \times n = n'$.

As in the classical case (69), the sought-for generalizations of replicators are the transitive automorphisms on *unitary $\underline{\alpha}$-brace* (66), but they *are not* abstract and *not* arbitrary. They are strictly bound to the declared meaning of the number: $\hat{N}$-operation of creating the copies. Therefore, by virtue of the equal rights of $\underline{\Psi}$ and $\underline{\Phi}$, it is imperative to bring the two one-fold copying acts $\hat{N}\{\overset{\Psi}{n}, \overset{\Phi}{m}\}\underline{\alpha}$ and $\hat{M}\{\overset{\Phi}{n}, \overset{\Psi}{m}\}\underline{\alpha}$ into play, which differ in the permutation of primitives $\underline{\Psi} \rightleftarrows \underline{\Phi}$. This point will determine a quantum extension of the replication.

As a result, since we have nothing but the copying $\hat{N}$ and "union" $\hat{+}$, the most general transformation of the brace $\{\overset{\Psi}{n}, \overset{\Phi}{m}\}\underline{\alpha}$ into (any) brace $\{\overset{\Psi}{n'}, \overset{\Phi}{m'}\}\underline{\alpha}$, which has been in a quantum-replication relation with it, is determined by the rule

$$\{\overset{\Psi}{n}, \overset{\Phi}{m}\}\underline{\alpha} \quad \xrightarrow{\widehat{(N,M)}} \quad \{\overset{\Psi}{n'}, \overset{\Phi}{m'}\}\underline{\alpha} \qquad \Rightarrow \qquad \{\overset{\Psi}{n'}, \overset{\Phi}{m'}\}\underline{\alpha} = \hat{N}\{\overset{\Psi}{n}, \overset{\Phi}{m}\}\underline{\alpha} \ \hat{+} \ \hat{M}\{\overset{\Phi}{n}, \overset{\Psi}{m}\}\underline{\alpha} . \quad (75)$$

This is the quantum version of Operators (69) and (70), and the foregoing ideology of $\hat{N}$-operators and of liberation from the $\underline{\Psi}$-symbols remains in force and entails the following. The numeral implementation of replicating the unitary brace (66), along with the $(n, m)$-representation of itself, is also determined by a certain pair $(N, M) \in \mathbb{R}^2$, i.e., by an operator symbol $\widehat{(N, M)}$.

The aforesaid means that the numerical form $(n, m) \xrightarrow{\widehat{(N,M)}} (n', m')$ of Transformation (75) is indistinguishable from a composition of pairs

$$(N, M) \odot (n, m) = (n', m') ,$$

where $\odot$ is a designation for the new binary operation. Its resultant structure is derived from the arithmetical nature (72) of the one-dimensional replication (69) described above, i.e., from the rules

$$\hat{N}\{\overset{\Psi}{n}, \overset{\Phi}{m}\}\underline{\alpha} = \{N\overset{\Psi}{\times}n, \ N\overset{\Phi}{\times}m\}\underline{\alpha}, \qquad \hat{M}\{\overset{\Phi}{n}, \overset{\Psi}{m}\}\underline{\alpha} = \{M\overset{\Phi}{\times}n, \ M\overset{\Psi}{\times}m\}\underline{\alpha} . \quad (76)$$

Here, a positivity/negativity of symbols $(n, m)$ in (66) should also be taken into account. Having regard to the foregoing, Rules (75) and (76) generate the Ansatz

$$(N, M) \odot (n, m) = (\pm Nn \pm Mm, \ \pm Nm \pm Mn) , \quad (77)$$

wherein all four signs $\pm$ are independent of each other, and the $(\times)$-multiplication of one-dimensional numbers in (72) and (76) have been re-denoted by the habitual standard $Nm := N \times m$. What should the pair-composition rule (77) be?

As was the case previously, the just-emerged binarity for $\odot$ should inherit—due to its operator origin—associativity, the existence of unity $\mathbb{1}$, and inversions. Namely, if the $(n, m)$-pairs are identified with the notation (57) according to the convention

$$(n, m) =: \mathfrak{a} , \quad (78)$$

then the following properties should be declared:

$$(\mathfrak{a} \odot \mathfrak{b}) \odot \mathfrak{c} = \mathfrak{a} \odot (\mathfrak{b} \odot \mathfrak{c}), \qquad \mathfrak{a} \odot \mathbb{1} = \mathfrak{a}, \qquad \mathfrak{a} \odot \mathfrak{a}^{-1} = \mathbb{1} . \quad (79)$$

From (75) and (76), it is not difficult to see that the combining (79) with (57) leads to a distributive coordination of operations $\oplus$ and $\odot$:

$$\mathfrak{c} \odot (\mathfrak{a} \oplus \mathfrak{b}) = (\mathfrak{c} \odot \mathfrak{a}) \oplus (\mathfrak{c} \odot \mathfrak{b}) . \tag{80}$$

However, the direct examination of this property shows that Ansatz (77) satisfies it automatically. More than that, we can even consider Ansatz (77) with parameters $\{\alpha, \beta, \gamma, \delta\}$ instead of $(\pm)$-signs:

$$\mathfrak{a} \odot \mathfrak{b} = (\mathsf{N}, \mathsf{M}) \odot (\mathsf{n}, \mathsf{m}) = (\alpha \mathsf{N}\mathsf{n} + \beta \mathsf{M}\mathsf{m}, \ \gamma \mathsf{N}\mathsf{m} + \delta \mathsf{M}\mathsf{n}) .$$

Then, the straightforward calculation shows that Distributivity (80) holds under the arbitrary $\{\alpha, \beta, \gamma, \delta\}$.

In turn, the examination of associativity—the first equality in (79)—under the same meaning for $\{\alpha, \beta, \gamma, \delta\}$ yields $\alpha = \gamma = \delta$ and free $\beta$. Returning to the $(\pm)$-values of these parameters, this associativity particularizes Ansatz (77) into the expression

$$(\mathsf{N}, \mathsf{M}) \odot (\mathsf{n}, \mathsf{m}) = \pm(\mathsf{N}\mathsf{n} \pm \mathsf{M}\mathsf{m}, \mathsf{N}\mathsf{m} + \mathsf{M}\mathsf{n});$$

now, with two independent signs $\pm$. Moreover, in passing, we reveal the commutativity

$$\mathfrak{a} \odot \mathfrak{b} = \mathfrak{b} \odot \mathfrak{a} , \tag{81}$$

though it was not presumed prior to that.

The search for unity $\mathbb{1}$ and subsequent finding of an inversion of the element $(\mathsf{n}, \mathsf{m})$ yield:

$$\mathbb{1} = (\pm 1, 0), \qquad (\mathsf{n}, \mathsf{m})^{-1} = \left(\frac{\mathsf{n}}{\Delta}, -\frac{\mathsf{m}}{\Delta}\right), \qquad \Delta := \mathsf{n}^2 \pm \mathsf{m}^2 .$$

Both the $(\pm)$-symbols continue to be independent here. The choice $\Delta = \mathsf{n}^2 - \mathsf{m}^2$ results in the absence of inversions $(\mathsf{n}, \mathsf{n})^{-1}$. This is in conflict with the group property (79) and also causes the unmotivated exclusivity of the unitary brace $\{\overset{\Psi}{\mathsf{n}}, \overset{\Phi}{\mathsf{n}}\}\underline{\alpha}$. There remains the case $\Delta = \mathsf{n}^2 + \mathsf{m}^2$, and it reduces the scheme to the form

$$\mathbb{1} = \pm(1, 0), \qquad (\mathsf{N}, \mathsf{M}) \odot (\mathsf{n}, \mathsf{m}) = \pm(\mathsf{N}\mathsf{n} - \mathsf{M}\mathsf{m}, \mathsf{N}\mathsf{m} + \mathsf{M}\mathsf{n})$$

with a single symbol $\pm$. It is a simple matter to see that the choice of sign $+$ or $-$ leads to the models that are isomorphic in regard to which of representatives $(+1, 0)$ or $(-1, 0)$ should be assigned for the identical replication $\hat{\mathbb{I}}$. By virtue of (61), it does not matter, and we declare

$$\boxed{\mathbb{1} := (1, 0), \qquad (\mathsf{N}, \mathsf{M}) \odot (\mathsf{n}, \mathsf{m}) = (\mathsf{N}\mathsf{n} - \mathsf{M}\mathsf{m}, \mathsf{N}\mathsf{m} + \mathsf{M}\mathsf{n})} . \tag{82}$$

This is nothing more nor less than the canonical multiplication of complex numbers $\mathsf{n} + \mathsf{i} \cdot \mathsf{m} = \mathfrak{a} \in \mathbb{C}$, if the following identifications are performed:

$$(1, 0) \rightleftarrows \mathbb{1}, \qquad (0, 1) \rightleftarrows \mathsf{i}, \qquad \{\oplus, \odot\} \rightleftarrows \{+, \cdot\}, \qquad (\mathsf{n}, \mathsf{m}) \rightleftarrows (\mathsf{n} + \mathsf{i} \cdot \mathsf{m}) . \tag{83}$$

Notice that the known fully matrix (over $\mathbb{R}$) equivalent to (82)

$$(\mathsf{n} + \mathsf{i} \cdot \mathsf{m}) \mapsto (\mathsf{n}, \mathsf{m}) \mapsto \begin{pmatrix} \mathsf{n} \\ \mathsf{m} \end{pmatrix} \mapsto \begin{pmatrix} \mathsf{n} & -\mathsf{m} \\ \mathsf{m} & \mathsf{n} \end{pmatrix}, \qquad \begin{pmatrix} \mathsf{n}' & -\mathsf{m}' \\ \mathsf{m}' & \mathsf{n}' \end{pmatrix} = \begin{pmatrix} \mathsf{N} & -\mathsf{M} \\ \mathsf{M} & \mathsf{N} \end{pmatrix} \circ \begin{pmatrix} \mathsf{n} & -\mathsf{m} \\ \mathsf{m} & \mathsf{n} \end{pmatrix}$$

does directly reflect the above ascertained operator essence

$$(\widehat{\mathsf{n}', \mathsf{m}'}) = (\widehat{\mathsf{N}, \mathsf{M}}) \circ (\widehat{\mathsf{n}, \mathsf{m}})$$

of both the number multiplication $\circ$ and the $\mathbb{C}$-number itself.

In view of the paramount importance of the $\mathbb{C}$-number field in QT [96,138,142], let us provide additional substantiations to the rigidity of the emergence of this specific number structure, i.e., of the axiom collection (57), (79)–(82). Among other things, the transpositions $\underline{\Psi} \rightleftarrows \underline{\Phi}$ used above fit more general reasoning.

*7.5. Involutions and $\tilde{\mathbb{C}}^*$-Algebra*

Apart from a freedom in ordering the primitives $\underline{\Psi} \rightleftarrows \underline{\Phi}$ in brace $\{\overset{\underline{\Psi}}{n}, \overset{\underline{\Phi}}{m}\}_{\underline{\alpha}}$, there is one more arbitrariness: reappointing them ($\underline{\Psi} \rightarrowtail \underline{\Theta}, \dots$) as elements of the set $\mathfrak{T}$. However, no physics predetermines any of these degrees of freedom. For, if other ingoing $\mathfrak{T}$-elements $\underline{\Theta}$ $\underline{\Omega}$ were present in (32) instead of $\underline{\Psi}, \underline{\Phi}$, then the theory of semigroup $\mathfrak{G}$, strictly, should be declared the segregated theories $\mathfrak{G}_{\underline{\Psi}\underline{\Phi}}$, $\mathfrak{G}_{\underline{\Theta}\underline{\Omega}}$, etc. It is clear that the labeling the theories, or a family thereof, is a manifest absurdity, and they should be thus factorized with respect to all kinds of ways to label them by $\mathfrak{T}$-primitives. The liberation from the $\underline{\Psi}, \underline{\Phi}$-icons and reconciliation of the result with pt. $\mathbf{R}^+$ (p. 25) are then performed by the scheme $\lceil$primitive has changed$\rceil \rightarrowtail \lceil$a number character is changing$\rceil$.

Inasmuch as declaring the $\{\underline{\Psi}, \underline{\Phi}, \underline{\Theta}, \dots\}$ to be ongoing primitives in (32) is a replacement of one to another, any such an appointment boils down to permutations of no more than *pairs* with two types (inner/outer):

$$\hat{\mathbb{J}}_{\underline{\Psi}\underline{\Phi}}: \quad (\underline{\Psi}, \underline{\Phi}) \overset{\underline{\Psi} \leftrightarrow \underline{\Phi}}{\rightleftharpoons} (\underline{\Phi}, \underline{\Psi}), \qquad \hat{\aleph}_{\underline{\Phi}\underline{\Theta}}: \quad (\underline{\Psi}, \underline{\Phi}) \overset{\underline{\Phi} \leftrightarrow \underline{\Theta}}{\rightleftharpoons} (\underline{\Psi}, \underline{\Theta}) . \tag{84}$$

However, it is immediately obvious that these reappointments change nothing in the $\cup$-relationships between (32) and are defined by the structural relations $\hat{\mathbb{J}}_{\underline{\Psi}\underline{\Phi}}^2 = \hat{\mathbb{I}}$, $\hat{\aleph}_{\underline{\Phi}\underline{\Theta}}^2 = \hat{\mathbb{I}}$. Then, the need to indicate the primitives themselves, as required, is eliminated, and their symbols may be thrown away if semigroup $\mathfrak{G}$ is properly furnished with the two abstract involutions $\hat{\mathbb{J}}$ and $\hat{\aleph}$. The $\mathfrak{G}$ itself, of course, also possesses involution (61) that turns it into the group *H*, but this involution has already had a numerical representation (66) by signs $\pm$. To be precise, it suffices to identify here the term "numerica" with the group arithmetic of the $\oplus$-addition (57) coming from the superposition principle realized on pairs (65) and (66). Therefore, the operators' actions (84) should be carried over onto objects defined in precisely this manner; nothing more needs to be assumed.

Operator $\hat{\mathbb{J}}_{\underline{\Psi}\underline{\Phi}}$ is immediately translated into a numerical form independently of the property that the objects $\{\overset{\underline{\Psi}}{n}, \overset{\underline{\Phi}}{m}\}_{\underline{\alpha}}$ form a (semi)group. Indeed, since the swap $\underline{\Psi} \rightleftarrows \underline{\Phi}$ in the unordered pair

$$\hat{\mathbb{J}}_{\underline{\Psi}\underline{\Phi}}: \qquad \{\overset{\underline{\Psi}}{n}, \overset{\underline{\Phi}}{m}\} \rightarrowtail \{\overset{\underline{\Phi}}{n}, \overset{\underline{\Psi}}{m}\} = \{\overset{\underline{\Psi}}{m}, \overset{\underline{\Phi}}{n}\} \qquad \cdots$$

(the $\underline{\alpha}$-label is dropped here as superfluous) is indistinguishable from the permutation of numbers n $\rightleftarrows$ m, the symbols $\underline{\Psi}$ and $\underline{\Phi}$ may be thrown away, organizing the numbers themselves into ordered pairs

$$\cdots \quad \Rightarrow \quad (n, m) \overset{\hat{\mathbb{J}}}{\rightarrowtail} (m, n) .$$

When required, the $\underline{\alpha}$-symbol returns hereinafter.

Let us now proceed to the outer involution $\underline{\Phi} \rightleftarrows \underline{\Theta}$ in (84):

$$\hat{\aleph}_{\underline{\Phi}\underline{\Theta}}: \qquad \{\overset{\underline{\Psi}}{n}, \overset{\underline{\Phi}}{m}\} \rightarrowtail \{\overset{\underline{\Psi}}{n}, \overset{\underline{\Theta}}{m}\} .$$

It is indifferent to the (first) $\underline{\Psi}$-element of the pair, and, by extracting it by the rule

$$\{\overset{\underline{\Psi}}{n}, \overset{\underline{\Phi}}{m}\} = \{\overset{\underline{\Psi}}{n}, \overset{}{0}\} \hat{+} \{\overset{}{0}, \overset{\underline{\Phi}}{m}\} ,$$

the question boils down to finding a representation of the transformations

$$\left(\{\overset{\underline{\Psi}}{n}, \overset{}{0}\} \hat{+} \{\overset{}{0}, \overset{\underline{\Phi}}{m}\}\right) \rightarrowtail \left(\{\overset{\underline{\Psi}}{n'}, \overset{}{0}\} \hat{+} \{\overset{}{0}, \overset{\underline{\Theta}}{m'}\}\right) \qquad (\overset{?}{n'}, \overset{?}{m'}) .$$

The component $\{\overset{\Psi}{\mathsf{n}},\overset{\Phi}{0}\}$ must go into itself since the symbol $\underline{\Psi}$ attached to it has not changed. It means that $\mathsf{n}' = \mathsf{n}$, and one is left with the task

$$\{\overset{\Psi}{0},\overset{\Phi}{\mathsf{m}}\} \overset{?}{\rightleftarrows} \{\overset{\Psi}{0},\overset{\Theta}{\mathsf{m}'}\} \,.$$

However, operation $\hat{\aleph}_{\Phi\Theta}$ recognizes only the primitive's symbols rather than their numbers. That is, replications $\hat{m}\{\overset{\Psi}{0},\pm\overset{\Phi}{1}\} = \{\overset{\Psi}{0},\pm\overset{\Phi}{\mathsf{m}}\}$ do formally commute with $\hat{\aleph}_{\Phi\Theta}$. Hence, by omitting the letters $\{\underline{\Psi},\underline{\Phi},\underline{\Theta}\}$, it will suffice to look for the representation of $\hat{\aleph}$ by numerical pairs $(0,\pm\mathsf{m})$ factorized with respect to replications $\hat{m}$, i.e., by the set $\{(0,1),(0,-1)\}$. It, for its part, remains to be transformed into itself, and the replication operators $\hat{n}, \hat{m}$ will recreate the generic case. The identical transformation $(0,\pm1) \rightarrowtail (0,\pm1)$ is ruled out since $\hat{\aleph}_{\Phi\Theta} \neq \hat{\mathbb{I}}$; therefore, $(0,\pm1) \overset{\hat{\aleph}}{\rightarrowtail} (0,\mp1)$. Restoring all the symbols that were dropped, the effect of $\hat{\aleph}$ reduces to the sign change for the second element of the coordinate pair:

$$(\mathsf{n},\mathsf{m}) \overset{\hat{\aleph}}{\rightarrowtail} (\mathsf{n},-\mathsf{m}) \,. \tag{85}$$

There is no need to change sign for the first element, as this change is the operator $-\hat{\mathbb{I}} \circ \hat{\aleph}$. Furthermore, one observes that the already existing group inversion $-\hat{\mathbb{I}}$ coincides with composition

$$(\hat{\aleph} \circ \hat{\beth})^2 = -\hat{\mathbb{I}} \,, \tag{86}$$

and we may even "forget" about (the "old") subtraction, leaving the equipment

$$\{\oplus, \hat{\mathbb{I}}, \hat{m}, \hat{\aleph}, \hat{\beth}\} \tag{87}$$

of semigroup $\mathfrak{G}$ as an irreducible set of mathematical structures over it.

In this connection, yet another—more formal—motivation of the passage $\lceil$semigroup $\rightarrowtail$ group$\rceil$ and thus of the superposition principle does arise. Indeed, the derivation of $\hat{\aleph}$ above engaged the inversion (61), but the reappointment of primitives $\underline{\Phi} \rightleftarrows \underline{\Theta}$ in (84) is a fully independent act. Therefore, if we forget about "$(-)$-copies of the positive pairs" $(0,\mathsf{m})$, the involutory nature of automorphism $\hat{\aleph}_{\Phi\Theta}$ would still reproduce the semigroup $\mathfrak{G}$ in numbers by "duplication" $\mathsf{m} \rightarrowtail \pm\mathsf{m}$, i.e., create the negative pairs $(0,-\mathsf{m})$, thus turning $\mathfrak{G}$ into a group *H*. An analogous reasoning on the symbol "$-$" could be cited even earlier, when the $\mathbb{C}$-field was being derived.

Now, remembering the above-described move to the binarity of $\odot$-multiplication on the $(\mathsf{n},\mathsf{m})$-pairs, we arrive at the problem of matching it with structures (87). Clearly, one needs only to ascertain the functionality of operators $\hat{\beth}$ and $\hat{\aleph}$ that were not available yet.

Relation (86) immediately gives us the correspondence $\hat{\aleph} \circ \hat{\beth} \rightleftarrows \mathsf{i}$ since $\mathsf{i}^2 = -1$. Hence, one of these operators, say $\hat{\beth}$, manifests itself in the imaginary unit $\mathsf{i}$. The origin of this operator—permutation $\hat{\beth}_{\Psi\Phi}$ in (84)—is the very same permutation $\underline{\Psi} \rightleftarrows \underline{\Phi}$ that generated the i-object in algebra (82) and (83). The second operator, i.e., (85), as is directly seen, is also not related to the binary $\oplus$ and $\odot$ but determines the change $\mathsf{i} \rightarrowtail -\mathsf{i}$. This means that the QM-consideration does not just give birth to the field $\mathbb{C}$ but to a division $\tilde{\mathbb{C}}^*$-algebra, which is equipped with two *non-binary* operations

$$\mathfrak{a} \overset{\hat{\aleph}}{\rightarrowtail} \mathfrak{a}^*, \qquad \mathfrak{a} \overset{\hat{\beth}}{\rightarrowtail} \tilde{\mathfrak{a}} \,.$$

Informally, it defines all the basic actions as "complex quantities" and thereby determines a QM-extension/generalization to the intuitive and habitual arithmetical manipulations (68)–(74) with real things. Consequently, the four binary arithmetical operations—addition/subtraction/multiplication/division—should be supplemented with the two unary ones: conjugation $\hat{\aleph}$ and swap $\hat{\beth}$.

**Remark 14.** *A curious observation for formal complex-number mathematics is appropriate here. None of these operators boil down to involution $-\hat{\mathbb{I}}$. We mean that each of the pairs $(\hat{\aleph}, -\hat{\mathbb{I}})$ or*

$(\hat{\mathbb{J}}, -\hat{\mathbb{I}})$ *is expressible through* $(\hat{\mathbb{J}}, \hat{\aleph})$ *and not the reverse; see* (86)*. To put it plainly, the self-suggested going from the natural sign-change (i.e.,* $-\hat{1}$ *over* $\mathbb{R}$*) to the inversion of the two-dimensional* $\oplus$*-addition (i.e.,* $-\hat{\mathbb{I}}$ *over* $\mathbb{C}$*) deprives the involution* $-\hat{\mathbb{I}}$ *of its primary character, as it has taken in the one-dimensional domain* $\mathbb{R}$*. Furthermore, the second operation* $\hat{\mathbb{J}}$ *is, in a sense, more "primitive" than the complex conjugation* $\hat{\aleph}$*, as this operation has had to conduct it with a formal pair* $(n, m)$—*merely transposes it—and does not invoke an arithmetic action, as does* $\hat{\aleph}$ *when changing the sign* $m \mapsto -m$ *in* (85)*.*

*The relationship between the operators is by binary multiplication:* $\tilde{\mathfrak{a}} = i \odot \mathfrak{a}^*$*. By virtue of this relation, it makes no odds which one of these unary operators is left for* $\mathbb{C}$*-algebra.*

We note—and this is important [6]—that the observational statistics $f_j$ are unchanged upon both operations $\hat{\mathbb{J}}$ and $\hat{\aleph}$.

*7.6. Naturalness of* $\mathbb{C}$*-Numbers*

Thus, the $\mathfrak{T}$-set primitives have been entirely banished from the theory, with the exception of the eigen-state $\underline{\alpha}_s$-markers, which are needed only for distinguishability (Section 2.1) in $\mathscr{A}$-observations. These markers may be interchanged, but permutability $\underline{\alpha}_j \rightleftarrows \underline{\alpha}_k$ is already reflected by the superposition's commutativity. Taking now into account the fact that reassigning the $\underline{\alpha}$-labels does not touch on the concept of the number, one infers: the covariance attained above is exhaustive. As a result, we draw the following conclusion.

- The coordinate representatives $\{\mathfrak{a}, \mathfrak{b}, \mathfrak{c}, \ldots\}$ of states and superpositions thereof (58) form a complex number field $\tilde{\mathbb{C}}^*$ equipped with the structures of conjugation and swap:

$$(n + im) \overset{*}{\mapsto} (n - im), \qquad (n + im) \overset{\sim}{\mapsto} (m + in) . \tag{88}$$

Statistical weights $f_j$ in object (35) are invariant with respect to both the involutions $f_j(\mathfrak{a}^*) = f_j(\mathfrak{a}) = f_j(\tilde{\mathfrak{a}})$ for each component $\mathfrak{a}_s$ independently.

What is more, the commentary on the primacy of QM over the abstract arithmetic (see p. 46) has a logical continuation.

- Quantum-theoretic description invokes no $\mathbb{C}$-numbers, nor does it introduce them. It *does create* them together with the $\tilde{\mathbb{C}}^*$-algebra. The $\mathbb{C}$-numbers are in and of themselves the *quantum* numbers.

This fact is remarkable in its own right because the "two-dimensional" numbers arise at the lowest empirical level, not from the need for solving any mathematical problems. Mathematics is still lacking. Therefore, pt. $R^+$ (p. 25) could have even been weakened by replacing ⌈homomorphism onto numbers⌉, roughly, with the ⌈homomorphism onto continuum⌉. Our minimal points of departure are replications and the ingoing/outgoing structure of brace (32). The imaginary part of the complex number—as a supplement to the real one—comes, as a rough guide, from the left-hand side of the conception $\underline{\Psi} \dashrightarrow \underline{\alpha}$. The theory does not depend on the meanings that will be later attached to the physical concepts—observables, measurement, spectra, means, etc.—their interpretations, or rigorous definitions. At the same time, the interferential "effects of subtraction and of zeroes" are intrinsically present within the construct's foundation itself.

Let us add, in conclusion, two more formal vindications of rigidity of the emerging $\mathbb{C}$-structure. In doing so, one assumes that we have already had the $\mathbb{R}$-numbers.

Unitary brace contain pairs of the form $\{\overset{\Psi}{n}, \overset{\Phi}{0}\}$. The binary operations $\{\oplus, \odot\}$ on their numerical representatives $(n, 0)$ are closed and, as easily seen, form a commutative field, which is isomorphic to $\mathbb{R}$. It is a subset of the generic pair set $(n, m)$. From the operator nature of $\odot$, it follows that these pairs form a certain distributive ring with general-group properties (79) and (80). The presence of the field $\mathbb{R}$ contained in it tells us that these pairs can be realized by the elements $n + mx$ of, at most, associative algebra $A$ over $\mathbb{R}$. Here,

$n, m \in \mathbb{R}$, $x$ is a generator of any ring's element beyond $\mathbb{R}$, and the habitual $+$ replaces the sign $\oplus$. The multiplication of two such elements

$$(n + mx) \odot (n' + m'x) = nn' + (mn' + nm')x + mm'x^2 = \cdots$$

immediately shows that the result does not depend on the order of factors, i.e.,

$$\cdots = (n' + m'x) \odot (n + mx),$$

due to the permutability of $\{n, m, n', m'\}$ between each other and of any $x$ with itself. This is a direct consequence of the two-dimensionality of the algebra $A$; it must be commutative. Invoking now the well-known Frobenius theorem on associative and commutative structures containing the field $\mathbb{R}$ [181], we arrive once again at a multiplication of the form (82). Körner puts this point as follows: "The complex numbers constitute the largest system of objects that most people are content to call numbers" ([172], p. 230).

Topologies on Numbers

Yet another reasoning about exclusivity of $\mathbb{C}$-numbers follows from matching the topological and algebraic properties of the general number systems ([182], Section 27). The case in hand is the uniqueness and non-arbitrariness in the emergence of the topological field $\mathbb{C}$; Pontryagin (1932). In our case, we have two continuums: the numerical symbols $n$ and $m$, each of which, by the very method of constructing the $(\Xi)$-objects (34), is equipped only with the natural ordering $<$. Since we do not have any more math-structures yet, the topology, continuity, and limits on each of the continuums can already be introduced with respect to this relation. For example, there is no need to introduce the topology a priori by creating the arithmetical operation of multiplication/divisibility of rationals (and a concept of the prime integer), as is conducted in the $p$-adic approaches to QM [66,183,184]. The "non-naturalness" of multiplication as compared with addition was already noted above. Moreover, in the $p$-adic versions for a numerical domain, the topologically and physically required matching between the natural ordering, connection, and continuity ([182], Ch. 4) is destroyed, and the approaches themselves stipulate the existence of the *observations* numbers with a comprehensive arithmetic. At the same time, questions about the "structure" of the physical $x$-space at Planck's scale and about measurements by rationals (see motivation in [183,184]) have not yet emerged because we are not relying on physical conceptions and are not yet introducing these notions as numerical. The $x$-space itself is as of yet absent, and D. Mermin [3] overtly claims along these lines that "when I hear that spacetime becomes a foam at the Planck scale, I do not reach for my gun". From the low-level empiricism standpoint, any objects, apart from the $\mathbb{R}^2$-continuality and frequencies $f$, call for independent axioms. In turn, the primary nature of the $\mathbb{R}$-continuality itself follows directly from the boolean $2^{\Xi}$ (p. 19) and $\Sigma$-postulate of infinity.

## 8. State Space

Quantum states . . . cannot be "found out"—W. Zurek ([8], p. 428)

. . . quantum theory refuses to offer any picture of what is actually going on out there—D. Mermin (1988)

### 8.1. Linear Vector Space

Once the replication $(\widehat{N, M})$ of brace $\{\overset{\yen}{n}, \overset{\Phi}{m}\}\underline{\alpha}$ has acquired a binary character

$$(\widehat{N, M})(\{\overset{\yen}{n}, \overset{\Phi}{m}\}\underline{\alpha}) \quad \Longleftrightarrow \quad ((N, M) \odot (n, m))|\alpha\rangle = \mathfrak{a}|\alpha\rangle, \tag{89}$$

the difference between "what is replicated" and "how many times" disappears. A symbol $|\alpha\rangle$ of the eigen-state has been attached to the abstract $\mathbb{C}$-number $\mathfrak{a}$. Construing this point as a quantum analog of re-choosing (liberation of) the measurement units (p. 46), we

obtain that the two formal states $\mathfrak{a}|\alpha)$ and $\mathfrak{b}|\alpha)$ are always connected by a certain number operator $\hat{\mathfrak{p}}$:

$$\mathfrak{b}|\alpha) = \hat{\mathfrak{p}}\left(\mathfrak{a}|\alpha)\right), \qquad \hat{\mathfrak{p}} \rightleftarrows \mathfrak{b} \odot \mathfrak{a}^{-1} \,.$$

Manipulating the numbers becomes independent of symbols $|\alpha)$. The way to formalize this is to think of generic states $\mathfrak{a}|\Psi) \in H$ as the "solid characters"

$$\mathfrak{a}|\Psi) \rightarrowtail |\Xi\rangle \in \mathbb{H} \,, \tag{90}$$

i.e., as the $|\Xi\rangle$-elements of a new set $\mathbb{H}$, which is equipped with the $\hat{\mathfrak{p}}$-replication images represented by the $\mathfrak{p}$-family ($\mathfrak{p} \in \mathbb{C}$) of maps

$$\mathbb{C} \times \mathbb{H} \overset{\cdot}{\mapsto} \mathbb{H} : \qquad \mathfrak{p} \cdot |\Xi\rangle = |\Phi\rangle \in \mathbb{H} \,, \tag{91}$$

and which is obliged to inherit the structure (89). This inheritance says that the coordination of $\odot$-multiplication in (89) with the replication's $\mathfrak{p}$-realization is performed by a new operation $\cdot$ of the unary kind on $\mathbb{H}$, i.e., (91), which should be subordinated to the rule

$$\mathfrak{p} \cdot \left(\mathfrak{a} \cdot |\Psi\rangle\right) = (\mathfrak{p} \odot \mathfrak{a}) \cdot |\Psi\rangle \qquad (\mathfrak{p}, \mathfrak{a} \in \mathbb{C}, \quad |\Psi\rangle \in \mathbb{H}) \,. \tag{92}$$

Due to this connection between operations $\odot$ and $\cdot$, the latter is usually referred to as "multiplication" as well; however, such an intuition with dropping the word "unary" may have implications ([130], Section 6.2). An analogous rule had already occurred in the relationship (59) between the $\oplus$-number $\mathbb{C}$-structure and the $(\pm)$-group superposition, i.e., when the multiplicative structures $\{\odot, \cdot\}$ were not available yet.

Among replication operators $\hat{\mathfrak{p}}$, there exists an identical transformation

$$\hat{\mathfrak{p}} = \hat{\mathbb{I}} : \qquad \mathfrak{a}|\Psi) \overset{\hat{\mathbb{I}}}{\rightarrowtail} \mathfrak{a}|\Psi) \,,$$

to which a symbol of the numerical unity $\mathfrak{p} = \mathbb{1}$ corresponds. From this, in accordance with (90) and (91), there follows the rule

$$\mathbb{1} \cdot |\Xi\rangle = |\Xi\rangle, \qquad \forall |\Xi\rangle \in \mathbb{H} \,.$$

It is clear that the $(\cdot)$-multiplication needs to be agreed with the $\uplus$-union. Let us make use of the fact that an object of (quantum) replication may be not only the unitary brace $\{\overset{\Psi}{\mathsf{n}}, \overset{\Phi}{\mathsf{m}}\}\underline{\alpha}$, which is equivalent to the eigen-element $\mathfrak{a}|\alpha)$, but a $(\hat{+})$-sum of the like objects and, in general, any constituents of quantum ensembles (see p. 44). Therefore, the $\hat{\mathfrak{p}}$-replication

$$\hat{\mathfrak{p}}\left(\mathfrak{a}|\alpha) \dotplus \mathfrak{b}|\beta)\right) = \cdots \tag{93}$$

is known to have its twin-sum

$$\cdots = \mathfrak{a}'|\alpha) \dotplus \mathfrak{b}'|\beta) = \cdots \tag{94}$$

with certain coefficients $\mathfrak{a}'$, $\mathfrak{b}'$.

Let us, for the moment, give back (93) to the initial language of operators/brace according to the scheme

$$\underbrace{(\widehat{N, M})}_{\mathfrak{p}}, \qquad \underbrace{\{\overset{\Psi}{\mathsf{n}'}, \overset{\Phi}{\mathsf{m}'}\}\underline{\alpha}}_{\mathfrak{a}} \,\hat{+}\, \underbrace{\{\overset{\Psi}{\mathsf{n}''}, \overset{\Phi}{\mathsf{m}''}\}\underline{\beta}}_{\mathfrak{b}} \,. \tag{95}$$

Take into account a pre-image of operation $\hat{+}$ on objects (93), i.e., the $\dotplus$. Then, (95) and the content of Sections 7.4 and 7.5 certainly show that the expression (94) must be of the form

$$\cdots = (\mathfrak{p} \odot \mathfrak{a})|\alpha) \dotplus (\mathfrak{p} \odot \mathfrak{b})|\beta) = \cdots \,.$$

Reconverting, by (92), expressions such as $(\mathfrak{p} \odot \mathfrak{a})|\alpha)$ into the operatorial $\hat{\mathfrak{p}}(\mathfrak{a}|\alpha))$, we complete the ellipsis

$$\cdots = \hat{\mathfrak{p}}\left(\mathfrak{a}|\alpha)\right) \hat{\pm} \hat{\mathfrak{p}}\left(\mathfrak{b}|\beta)\right) \, .$$

Passing now to the $\mathfrak{p}$-number and to the $|\Xi\rangle$-objects (90), i.e., replacing $\mathfrak{a}|\alpha) \rightarrowtail |\Psi\rangle$ and $\mathfrak{b}|\beta) \rightarrowtail |\Phi\rangle$, one derives an additivity of operation $\cdot$ when acting on a sum:

$$\mathfrak{p} \cdot \left(|\Psi\rangle \hat{+} |\Phi\rangle\right) = \mathfrak{p} \cdot |\Psi\rangle \hat{+} \mathfrak{p} \cdot |\Phi\rangle \, .$$

Here, the *H*-addition $\hat{\pm}$ has been carried over to the group $\mathbb{H}$ as a new symbol $\hat{+}$. This is nothing but a distributive coordination of the $(\cdot)$-multiplication with the group addition $\hat{+}$.

In a similar way, through a number operator, one establishes yet another relation

$$\mathfrak{a} \cdot |\Xi\rangle \hat{+} \mathfrak{b} \cdot |\Xi\rangle = (\mathfrak{a} \oplus \mathfrak{b}) \cdot |\Xi\rangle$$

between $\cdot$ and $\hat{+}$. Its origin is equivalent to (59). From the constructs above, it is not difficult to see that we have examined all the possibilities of $\mathbb{C}$-replicating the superpositions (58) or their constituents, which is why we have exhausted all the compatibility rules that stem from the two fundamental operations—replication and union ($\hat{\pm}$).

Thus, having considered the passage (90) and (91) as a final homomorphism of the *H*-group elements $\mathfrak{a}|\Psi)$ onto the objects $|\Xi\rangle \in \mathbb{H}$, i.e., adjusting the previous concept of a state and of `DataSource` (p. 31), we infer the following.

- The minimal and mathematically invariant bearer of the observation's empiricism is an abstract space $\mathbb{H}$ of states $|\Psi\rangle$ of the system $\mathcal{S}$. The structural properties

$$\mathbb{H} := \left\{|\Psi\rangle, |\Phi\rangle, \dots\right\} \quad \lceil\text{commutative group under operation } \hat{+}\rceil \, , \qquad (96)$$

$$\mathbb{C} := \{\mathfrak{a}, \mathfrak{b}, \dots\} \quad \lceil\text{field of complex numbers (57), (78)–(82)}\rceil \, ,$$

$$\hat{\mathfrak{a}}|\Psi\rangle =: \mathfrak{a} \cdot |\Psi\rangle \in \mathbb{H} \quad \lceil\text{closedness under operation } \cdot \qquad\qquad (97)$$
$$\Longleftrightarrow \text{operator automorphism } \hat{\mathfrak{a}}|\Psi\rangle\rceil \, ,$$

$$\mathfrak{a} \cdot \left(\mathfrak{b} \cdot |\Psi\rangle\right) = (\mathfrak{a} \odot \mathfrak{b}) \cdot |\Psi\rangle, \qquad \mathfrak{a} \cdot |\Psi\rangle \hat{+} \mathfrak{b} \cdot |\Psi\rangle = (\mathfrak{a} \oplus \mathfrak{b}) \cdot |\Psi\rangle \, ,$$
$$\mathbb{1} \cdot |\Psi\rangle = |\Psi\rangle, \qquad\qquad \mathfrak{a} \cdot \left(|\Psi\rangle \hat{+} |\Phi\rangle\right) = \mathfrak{a} \cdot |\Psi\rangle \hat{+} \mathfrak{a} \cdot |\Phi\rangle \qquad\qquad (98)$$

of the space coincide with the axioms of a linear vector space (LVS) over the field $\mathbb{C}$.

Attention is drawn to the fact that this is the first place in our construct where the word "*linear*" has appeared, and even the superposition principle, page 35, was formulated without using this term. The "axiom list" (96)–(98) should also be complemented with a declaration of the global D-number value (53) established above.

In a nutshell, the nature of the quantum state space is two-fold: group superposition (58) and operator nature of the "$\mathfrak{a}$-numbering" the elements of the group. It admits the $\mathbb{C}$-field scalars as operators. Relations (98) describe the rules of "interplay" between all the objects. It is known that such formations, while being implemented by a binary algebra of numbers, turn into the vector spaces and modules [175] (Ch. 5), [181] (Sections I(7.1–2), II(13.4)). Concerning the consistency of these rules—say, of numerical distributivity (80)—with relations (98), see the work [185].

**Remark 15.** *A certain oddity is in place.* QM-*empiricism is such that the standard definition of* LVS *by the all-too-familiar axioms (96)–(98) is more "non*physical*" by its nature than the "generalistic" abstraction of a group with operator automorphisms of the group H-structure itself [166] (Section I(4.2)), [175]. A point like this might be expected, though. This is because, as noted in Section 1.2, meaning all of the tokens in (1) and their origin are entirely unknown, and the linearity of* QM *is radically different from other "linearities" in physics.*

*All told, the appearance sequence of the mathematical structures is as follows:*

$\ulcorner$*sets, union* $\cup$, ...$\urcorner$ $\rightarrowtail$ $\ulcorner$*semigroup (Section 5.2)*$\urcorner$

$\quad\rightarrowtail$ $\ulcorner$*group H (Section 6.3)*$\urcorner$ $\rightarrowtail$ $\ulcorner$*numbers & arithmetics (Section 7)*$\urcorner$

$\quad\rightarrowtail$ $\ulcorner$*compatibility of the group and numbers (Section 8.1)*$\urcorner$

$\quad\rightarrowtail$ $\ulcorner$*the abstract LVS and its bases*$\urcorner$ .

*This sequence is rigid, such as the box-diagram in Section 1.3, so the structure of LVS cannot be weakened because we have the two fundamental principia (**II** and **III**) in between the semigroup, the group, and the vector space.*

### 8.2. Bases, Countability, and Infinities

From a ban on transitions $\underline{\alpha}_s \dashrightarrow \underline{\alpha}_n$ under $s \neq n$, it follows that unitary $\underline{\alpha}_s$-brace (34) corresponds to vectors $\mathfrak{a}_s \cdot |\boldsymbol{\alpha}_s\rangle$ that are linearly inexpressible through each other. Aside from the general ensemble brace (32), no other elements exist, and all of them are in one-to-one correspondence with the vector representations $\mathfrak{a}_1 \cdot |\boldsymbol{\alpha}_1\rangle \,\hat{+}\, \mathfrak{a}_2 \cdot |\boldsymbol{\alpha}_2\rangle \,\hat{+}\, \cdots$. Each such vector has a statistical pre-image (32), and vice versa; there are no gaps. This means that the system of vectors $\{|\boldsymbol{\alpha}_1\rangle, |\boldsymbol{\alpha}_2\rangle, \ldots\}$ forms a basis of $\mathbb{H}$ as the basis of LVS—*basis of an observable $\mathscr{A}$ or $\mathscr{A}$-basis*—and the number of symbols $|\boldsymbol{\alpha}_s\rangle$ is its dimension: $\dim \mathbb{H} = \mathrm{D}$. The $\mathrm{D} = \infty$ case, just like anything associated with infinity, cannot be formalized without topology, and its presence is presumed, but this discussion is dropped. We just remark that even earlier, when arising the two-dimensional continuum, we have silently assumed the $(\mathbb{R} \times \mathbb{R})$-product topology on it. This supposition is natural, inasmuch as it does not involve additional constructions/requirements. Thus, if Properties (96)–(98) are directly accepted as empirical, then the mathematical rigors augment them axiomatically on the outside because one constructs the mathematical theory.

The micro-transition $\xrightarrow{\mathscr{A}}$ in Section 2.1 is a solitary entity. This means that the number of eigen $\underline{\alpha}_s$-primitives for an actual instrument may be either finite or discretely unbounded. We base this on the fact that continual formations are products of mathematics rather than empiricism (see also [58] (p. 35)). The $\mathfrak{T}$-set, as an example, is also non-continual, but that premise may even be given up because only a discrete portion of this set is present at arguments (transitions $\xrightarrow{\mathscr{A}}$). Notice incidentally that continuum, along with the number, does not feature in the ZF-axioms [134] but is also created, just as "an infinity is actually not given to us at all, but is ... extrapolated through an intellectual process." [105] (p. 55; Hilbert–Bernays); see also the book [84] for the conceptualization of infinity. One obtains a countability of the $\mathscr{A}$-basis. Hence, it follows a completeness of $\mathbb{H}$ and countability of dimension (53), as of the number LVS-invariant:

$$\mathrm{D} = 2, 3, \ldots, \aleph_\circ \qquad (= \dim \mathbb{H}) . \tag{99}$$

Finally, let us mention the following. The basis is a term that in no way is present in the abstract axiomatics (96)–(98), and LVS on its own account does not contain a motive for introducing that concept. However, empirically, the $\mathbb{H}$-space is arising entirely and ab initio in all possible linear combinations over $|\boldsymbol{\alpha}_j\rangle$, i.e., through $\mathscr{A}$-bases. Because of this, in order for an abstract LVS to become the quantum state space, the LVS should be considered as being accompanied by the concepts bases and changes thereof. Conforming to such a requirement and the formal existence of a basis is given by a nontrivial math theorem invoking the axiom of choice ([166], Section **II**(7.1)).

### 8.3. The Theorem

The states $|\boldsymbol{\Psi}\rangle$ and sums thereof, at the moment, form a formal family of different elements. Recall that symbols $\{\approx, \not\approx\}$ in pt. **R**, as from the end of Section 5.4, have been replaced with the standard ones $\{=, \neq\}$. The physical aspects of $\langle\!\langle \mathcal{S}, \mathbf{M}, \ldots \rangle\!\rangle$ were being left aside so far, and, for example, $|\boldsymbol{\Psi}\rangle$ and $\mathfrak{c} \cdot |\boldsymbol{\Psi}\rangle$ were the different vectors of the $\mathbb{H}$-space. However,

- empiricism (deals with and) yields originally *not states* and superpositions thereof *but* $|\boldsymbol{\alpha}\rangle$*-representations*.

It is these representations (alone) that carry information about statistics $(f_1, f_2, \ldots)$ through coefficients $\mathfrak{a}_j$. The replicative character of $\mathfrak{c}$-multipliers and $\Sigma$-postulate entail, however, that the two vectors $\mathbb{1} \cdot |\boldsymbol{\Psi}\rangle$ and $\mathfrak{c} \cdot |\boldsymbol{\Psi}\rangle$ should correspond to the one and same statistics $f_{(\mathscr{D})} = (1, 0, \ldots) = \tilde{f}_{(\mathscr{D})}$ under an observation $\mathscr{D}$ with the eigen collection $\{\mathbb{1}|\boldsymbol{\Psi}), \ldots\}$.

Let us write the equalities

$$
\begin{array}{ccccccc}
f_{(\mathscr{D})} & \twoheadleftarrow & \underbrace{\mathbb{1} \cdot |\boldsymbol{\Psi}\rangle}_{\text{observation } \mathscr{D}} & = & \underbrace{\mathfrak{a}_1 \cdot |\boldsymbol{\alpha}_1\rangle \hat{+} \mathfrak{a}_2 \cdot |\boldsymbol{\alpha}_2\rangle \hat{+} \cdots}_{\text{observation } \mathscr{A}} & \twoheadrightarrow & f_{(\mathscr{A})} \, , \\[2em]
\tilde{f}_{(\mathscr{D})} & \twoheadleftarrow & \overbrace{\mathfrak{c} \cdot |\boldsymbol{\Psi}\rangle} & = & \overbrace{\mathfrak{c} \cdot \left( \mathfrak{a}_1 \cdot |\boldsymbol{\alpha}_1\rangle \hat{+} \mathfrak{a}_2 \cdot |\boldsymbol{\alpha}_2\rangle \hat{+} \cdots \right)} & \twoheadrightarrow & \tilde{f}_{(\mathscr{A})}
\end{array}
\tag{100}
$$

and look at them in the following order: the first line from right to left and the second in the reverse direction. Their right-hand sides are the carriers of some statistics $f_{(\mathscr{A})}$ and $\tilde{f}_{(\mathscr{A})}$. The frequencies $f_{(\mathscr{A})} = (f_1, f_2, \ldots)$ come from the number set $(\mathfrak{a}_1, \mathfrak{a}_2, \ldots)$ under the same environments $\langle\!\langle \mathcal{S}, \mathbf{M}, \ldots \rangle\!\rangle$ that give rise to the statistics $f_{(\mathscr{D})}$. However, it is also generated by the representative $\mathfrak{c} \cdot |\boldsymbol{\Psi}\rangle$, which is associated with the same $\langle\!\langle \mathcal{S}, \mathbf{M}, \ldots \rangle\!\rangle$; hence, $\tilde{f}_{(\mathscr{D})} = f_{(\mathscr{D})}$. By virtue of the second equal sign in (100), the same $\langle\!\langle \mathcal{S}, \mathbf{M}, \ldots \rangle\!\rangle$ are associated with the second $\mathscr{A}$-collection $(\mathfrak{c} \odot \mathfrak{a}_1, \mathfrak{c} \odot \mathfrak{a}_2, \ldots)$. Therefore, the frequencies $\tilde{f}_{(\mathscr{A})}$ that emanate from it have to be identical to those emanating from the first collection $(\mathfrak{a}_1, \mathfrak{a}_2, \ldots)$. That is to say $\tilde{f}_{(\mathscr{A})} = f_{(\mathscr{A})}$, and the scale stretches $|\boldsymbol{\Psi}\rangle \longmapsto \mathfrak{c} \cdot |\boldsymbol{\Psi}\rangle$ are not recognized by any $\mathscr{A}$-instruments. A more concise reasoning is that the quantum replication $\mathfrak{c} = \mathfrak{n} + i\mathfrak{m}$ may be viewed as one-dimensional replications $\hat{n} \, (\mathfrak{a}|\alpha))$, $\hat{\imath} \, (\mathfrak{a}|\alpha))$ of all the brace $\mathfrak{a}_s|\alpha_s$)-images and of sums such as $\hat{n} \, (\mathfrak{a}|\alpha)) \hat{\pm} (\hat{\imath} \circ \hat{m})(\mathfrak{a}|\alpha))$. These replications do not change the superposition statistics as a whole.

The aforesaid gives birth to a universal—stronger than $\approx$ and irrespective of instruments—observational equivalence relation

$$
|\boldsymbol{\Psi}\rangle \approxeq \text{const} \cdot |\boldsymbol{\Psi}\rangle
$$

on the space $\mathbb{H}$, i.e., the "physical" indistinguishability (Section 2.4).

The basis vectors $|\boldsymbol{\alpha}_s\rangle$ and their $(\approxeq)$-equivalents will be referred to as *eigen vectors/states of instrument $\mathscr{A}$*. Clearly, the concepts of instrument and of (macro)-observation (**O**) should now be distinguished. Accordingly, the spectral constructions (51) and (52) should be corrected. Call the data set

$$
\left\{ |\boldsymbol{\alpha}_1\rangle_{\lfloor\alpha_1}, |\boldsymbol{\alpha}_2\rangle_{\lfloor\alpha_2}, \ldots \right\} =: [\mathscr{A}]
\tag{101}
$$

the $[\mathscr{A}]$-*representative* of instrument $\mathscr{A}$ in $\mathbb{H}$. The add-on (101) does not touch on $\mathbb{H}$-space since the spectral labels $\lfloor\alpha_j$ are the self-contained objects independent of vectors $|\boldsymbol{\alpha}_k\rangle$. These labels and their degenerations determine internal properties of the formalized notion of an instrument (101). Conversely, any state vector $|\boldsymbol{\Psi}\rangle$ or $\mathfrak{c} \cdot |\boldsymbol{\Psi}\rangle$ may be treated as a $[\mathscr{C}]$-representative for an imagined/actual instrument $\mathscr{C}$ (spectrum is arbitrary) and is a certain $(\hat{+})$-sum of the eigen elements for any other $[\mathscr{A}]$-representative.

Remembering (23), we arrive at the quantum "*kinematic framework*" [69], i.e., at the ultimate conclusion that determines the pre-dynamical theory of macroscopic data on micro-events.

- **The core first theorem of quantum empiricism:**

  (1) The mathematical representatives of physical observations and of preparations are the quantum states $|\boldsymbol{\Psi}\rangle$ and statistical mixtures of eigen $|\boldsymbol{\alpha}\rangle$-states

$$
\left\{ |\boldsymbol{\alpha}_1\rangle^{(\varrho_1)}, |\boldsymbol{\alpha}_2\rangle^{(\varrho_2)}, \ldots \right\}, \qquad \varrho_1 + \varrho_2 + \cdots = 1 \, .
\tag{102}
$$

(2) Properties (96)–(99) define objects $|\Psi\rangle$ as elements of a (complete separable) linear vector space $\mathbb{H}$ over the algebra of complex numbers $\mathbb{C}^*$.

(3) Dimension $\dim \mathbb{H} = D \geqslant 2$, representing an observable quantity ($D < \infty$), is set to the value $\max\{|\mathfrak{T}_{\mathscr{A}}|, |\mathfrak{T}_{\mathscr{B}}|, \ldots\} = D$ as required by the accuracy of the toolkit $\mathcal{O} = \{\mathscr{A}, \mathscr{B}, \ldots\}$. The eigen $|\boldsymbol{\alpha}\rangle$-vectors for each $[\mathscr{A}]$-representative provide a basis of $\mathbb{H}$ ($D < \infty$) independently of spectra (101).

(4) The $\mathscr{A}$-bases stand out because the *observational* number-notion has been associated to them—statistics of the micro-events. The frequencies $\mathfrak{f}_k(\mathfrak{a})$ are invariant under involutions (88) and states $|\Psi\rangle$ and $\mathfrak{c} \cdot |\Psi\rangle$ are statistically indistinguishable.

(5) Rules (96)–(99), for a fixed $D \neq \infty$, are categorical as an axiomatic system; they admit no non-isomorphic models.

The words "complete separable" have been supplemented here for mathematical reasons. This point is partly commented on in [6] and more fully in [130]. Indeed, the algebra constructed above calls for some amendments of a topological nature because the construction contains three infinities: continuum $\mathbb{C}$, continuum $\mathbb{H}$, and dimension D. In this connection, see the book [182]. The term "categorical" may require some explanation, and it is fully given in [130]. Here, one suffices to mention the point that one mathematical axiomatical system can in general have different inequivalent realizations/models [105,106,136]. In turn, the only thing that distinguishes two vector-space models between themselves is their dimension D.

Now, having considered the micro-destruction arrays with empirical rather than a formal take on arithmetic, the ideology of creating the quantitative theory leads to the key feature of quantum states—addition thereof—and the quantities under addition "do amount to" the complex numbers.

**Remark 16.** *As in Section 7.3, we draw attention to a hidden and (logically) unremovable extension of the physical units' concept.*

- *"… units. Despite the rudimentary nature of units, they are probably the most inconsistently understood concept in all of physics … where do units come from?".*

  *S. Gryb and F. Mercati ([102], p. 91)*

*Surprisingly, the naïve and straightforward conjunction of this concept with an abstract number seems to contravene the multiplication arithmetic but not the addition one. The typical example illustrates the point:*

$$(2\,kg) \times (3\,kg) \neq 6\,kg \qquad (3\,sheep + 5\,\psi \neq 8\,St\ddot{u}ck)$$

*(see also [172] (items (4) and (5) on p. 16)). On the other hand, $2 \times (3\,kg) = 6\,kg$ and $(2\,kg) + (3\,kg) = 5\,kg$, and the kg may be replaced here with any other entity: the classical meters, the abstract "Quanten Stücke $\psi$", and the like. They have no any operational significance, but one cannot get by without them.*

*The numeral characters acquire their usual abstract-numerical meaning—mathematization [58] —only when we throw (Section 7.3) the "units" $\lceil St\ddot{u}ck, {}^{\circ}C, sheep, \psi, \ldots \rceil$ out of data like $\lceil 5\,St\ddot{u}ck, 5\,{}^{\circ}C, 5\,sheep, 5\,\psi, \ldots \rceil$. The symbol "5" in $5\,{}^{\circ}C$ is the very same "5" as in $5 \cdot |\boldsymbol{\alpha}\rangle$. It is pointless without such a matching/abstracting. In the Newtonian spirit (epigraph to Section 7), the symbol could be defined as follows:*

$$\lceil the\ abstraction\ 5 \rceil := \frac{\widehat{5}\,{}^{\circ}C}{\widehat{1}\,{}^{\circ}C} = \frac{\widehat{5}\,|\Psi\rangle}{\widehat{1}\,|\Psi\rangle} = \cdots = \frac{\widehat{5}\,unit}{\widehat{1}\,unit} .$$

*It may be even said that creating the number $\mathsf{n}$ (from its operator $\hat{\mathsf{n}}$) as an abstracted entity reflects a kind of covariance with respect to attaching the various language tags regardless of whether they are real St\ddot{u}ck, sheep, or the abstract ones such as $|\Psi\rangle$. At the same time, the inversion of this abstraction—attaching the $\lceil St\ddot{u}ck, {}^{\circ}C, sheep, \psi, \ldots \rceil$ to the character 5—is always an*

interpretation of abstraction: *interpreting "the Stück", "°C-interpretation the Celsius", etc. It is not improbable that this is the only point when the completed* QT—QM/QFT/quantum-gravity *yet to be constructed will resort to word interpretation. See also comments by D. Darling on "sheep, fingers, tokens, numbers, things, to "add" things, abstraction—the process of addition" and the like on p. 178 in the book [2] and in [23] (pp. 263–264).*

Incidentally, within this physical and quantum context:

- The LVS itself should be regarded as no less a primary math-structure than the numbers themselves. Empiricism gives birth to both these structures together. Neither of them is more/less abstract/necessary than the other. Behind them is certainly a commutative group with operator automorphisms over it, and "numbers" is just a shortened term for that operators. Therein lies their nature (Section 7.2).

The habitual physics' construct $\lceil$number$\rceil \times \lceil$physical unit$\rceil$ exemplifies in effect the simplest (one-dimensional) LVS. However, the structure "the LVS", in contrast to the "bare" arithmetic, simply "does not forget and keeps" an operator nature (unary multiplication $\cdot$) of the structure "the number" and its empirical inseparability from the notion of the unit:

$$\underbrace{\hat{2}\left(\hat{3}\,\text{unit}\right) \iff 2\cdot(3\cdot\text{unit})}_{\text{vector space}} \;\rightarrowtail\;\cdots\;\text{abstracting}\;\cdots\;\rightarrowtail\; \underbrace{(2\times3)\;\cancel{\text{unit}}}_{\text{arithmetic}}\,.$$

A direct corollary of this point is the fact that principium **II** can in no way be given up or disregarded. This would be tantamount to impossibility to introduce the further empirical (and classical) notion of a physical unit. The "forgetfulness" of arithmetic about measuring units even leads to a new way of looking at the classical Pythagoras theorem ([130], Section 6).

At the moment, it is worthwhile to summarize where we stand. As we have seen, nothing above and beyond what was used in constructing the mathematics (96)–(99) is required to explain the nature and meaning of the quantum state. Moreover, we have obtained not merely a completion of construction (11):

$$\bigoplus(\mathfrak{a}_1,|\boldsymbol{\alpha}_1\rangle;\mathfrak{a}_2,|\boldsymbol{\alpha}_2\rangle;\ldots) = \mathfrak{a}_1\cdot|\boldsymbol{\alpha}_1\rangle\;\hat{+}\;\mathfrak{a}_2\cdot|\boldsymbol{\alpha}_2\rangle\;\hat{+}\;\cdots\,.$$

In the first place, one establishes *a genesis of the quantal discreteness. Discriminating is an isolated act in the very nature of the perception process*: "one thing is distinct from another", "the controlling the minimal begins with a distincting of something the two", and the like (Section 2.2). Accordingly, "indivisibility, or "individuality", characterizing the elementary processes" ([131], p. 203; N. Bohr) must be formalized into the *"elemental" click*.

- The classical continuality of the perceptual reality—the $(3+1)$-space, fields $\{u(\boldsymbol{x},t), \psi(\boldsymbol{x},t), \ldots\}$, and the $\mathbb{R}$-numbers—is a theorization act, whereas the nature of the perception fundamentally "contains an element of discontinuity" ([4], p. 179). The continuality of the classical-physics mathematics we are used to is a "quantum effect".

The theorization also bears on preparation $\langle\!\langle \mathcal{S}, \mathbf{M}, \ldots \rangle\!\rangle$. For example, smoothly reducing the interferometer intensity is not an empiricism but an *imagination* of abstracta the continuity/infinity. Clearly, such an (incorrect) substitution of the perception process should somewhere be replaced with a "correct understanding", such as the introduction of the categories: $\lceil$isolated micro-events$\rceil$ + $\lceil$(myriads) assemblages thereof$\rceil$. Granted, the natural language is able to describe the discontinuity only in the classical (the energy) terms—Plank's quantum of action $\hbar$, although the *quantum discreteness is not a discretization of* something classical but a discreteness on its own account.

We also clarify the formalization of measurement/preparation and of genesis of the $\mathbb{C}$-numbers. The well-known $(*)$-conjugation operation also finds its origin. Moreover, it is supplemented with a transposition $\Re(\mathfrak{a}) \rightleftarrows \Im(\mathfrak{a})$ of the real/imaginary part of the $\mathbb{C}$-number, and this transposition should be regarded just as natural operation as the conjugation. The emergent concepts of spectra and of their degenerations and eigen-states

provide a nearly comprehensive mathematical image of physical observables. The state becomes devoid of its mysteriousness [21,31,186] since it is explicitly built in terms of the unique model of the "statistical" $|\alpha\rangle$-representatives supplemented with macroscopic mixtures (102).

## 9. Numbers, Minus, and Equality; Revisited

> … quas decet numeris negativis exprimantur, additio et subtractio consueto more peracta nullis premitur difficultatibus—L. Euler (1735)

> (… if we represent the notions, which are necessary, by negative numbers, then addition and subtraction … are executed without any difficulty.)

### 9.1. Separation of the Number Matters

The empirical adequacy of QT can be based only on empirical ensembles (Sections 2.5 and 4). The creation of their mathematics tells us, then, that the "quantity of something" (68)–(70) turns into a formal operational algebra through labeling the operator replications (Sections 7.1 and 7.2) and properly yields the numbers. At first, they appear merely as

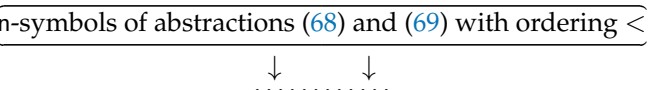

$$\boxed{\text{n-symbols of abstractions (68) and (69) with ordering} <}$$
$$\downarrow \qquad \downarrow$$
$$\cdots\cdots\cdots\cdots$$

and then as internal objects of theory:

$$\cdots\cdots\cdots\cdots$$
$$\downarrow \qquad \downarrow$$
$$\boxed{\text{numbers n as elements of arithmetic (71)–(74)}}$$
$$\downarrow \qquad \downarrow$$
$$\boxed{(\mathrm{m,n})\text{-numbers } \mathfrak{a} \text{ and their } \tilde{\mathbb{C}}^*\text{-algebra (57), (78)–(82), (88)}}$$
$$\downarrow \qquad \downarrow$$
$$\cdots\cdots\cdots\cdots$$

.

These steps are necessary and mean that not only are the complex numbers far from self-evident, but even the negative ones are; a key place (50), (55) wherein a group arises. All the other structural points, first and foremost the observational quantities, may be further produced (even as concepts) only by way of certain mathematical mappings:

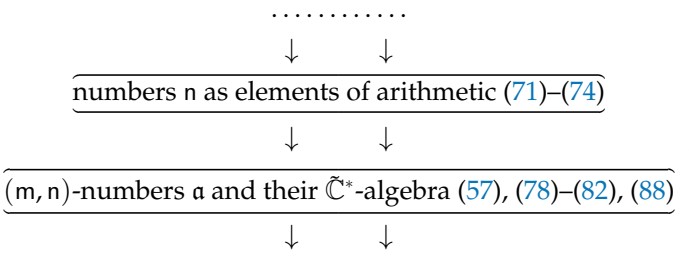

$$\cdots\cdots\cdots\cdots$$
$$\downarrow \qquad \downarrow$$
the observations numbers $\mathfrak{f}_j$ and $\alpha_j$:
$$\mathfrak{a} \mapsto \lceil \text{statistics } \mathfrak{f}_j \rceil, \quad |\alpha_j\rangle \mapsto \lceil \text{spectra } \alpha_j \rceil \quad . \tag{103}$$
tensorial structure of the $\mathbb{H}$-space

In other words, if a concept is a numerical one already in empiricism—frequencies, spectra, etc.—then its meaningful formalization by means of a mathematical definitio can only resort to mathematics that we have at our disposal: LVS and algebra of numbers.

Thus, numerical quantities in the entire theory are initially divided up by their emergence mechanism (**II**): the intrinsic abstracta and reifications (103). Without such a division, the *circular logic is inevitable*, and the above-mentioned "unit" treatment of numbers would still be supplemented with the task of their observational interpretation complicated by two-dimensionality. This task would be present in formalism not merely as a problem but as an inherently intractable challenge. Actually, *any entity can be identified with numbers*,

and this is why, the quantum empiricism and principium **II**—paradigm of the very number in the physical theory—insist on the need to pay the closest possible attention to all these things.

**Remark 17.** *In this regard, the situation is parallel with the familiar history of electrodynamics of moving bodies, as was pointed out just before principium* ***III****. Lorentz's contraction theory is inconsistent, if the space-time tags to events are not linked up to the empirically precise and operationally defined concepts in different reference frames: clocks, simultaneity, rigid rods, distances, and the like.*

*In the quantum case, the chief subject of empirical definition is a concept of the number and of the "numerical value of . . . ". Otherwise, the meaning given to the conception of a quantitative theory itself has been blurred. The "quantum numbers"* $\mathbb{C}$ *are built up from the reals, and the latter have an operator nature (Section 7.2). However, the complexes* $\mathbb{C}$*, being also operators and unlike the reals, never act (operationally) on the reified quantities. They do act on the abstract* $|\mathbf{\Psi}\rangle$*-elements of the abstract commutative group* $\mathbb{H}$*. Recall that this group and superposition principle were arising before the numbers.*

We have seen now that "it is quite wrong to try founding a theory on observable magnitudes[/categories] alone" [8] (p. 504; Einstein, in a talk (1926) to Heisenberg), and resorting to the physical notions—the camouflaged **M**-observations—is prohibited (see also Remark 2). The attempts to use statistics at the very beginning of the theory are known [29,34,87,90,93,187,188], and rightly so; they were initiated by H. Margenau (1936) [33] (Ch. 15). However, the scheme just given is rigid. To obviate the premature appearance of the very need for an interpretation, the scheme must not be varied. Being a sequence of steps, it provides in essence an answer to principium **II**.

*9.2. Operations on Numbers*

The last step in this scheme contains, in particular, the map $\mathfrak{a} \mapsto \mathfrak{f}$, i.e., measurement (48). Its form should be established in its own right—Born's rule [6]. To illustrate, the naïve transformation of negative numbers $\mathfrak{p}$ into the actually perceived quantities by a "seemingly natural" rule such as $|\pm\mathfrak{p}|$ is not correct and does not follow from anywhere. For the built algebra (96)–(98), the operation $\pm\mathfrak{p} \mapsto |\pm\mathfrak{p}|$ is extrinsic and illegal. According to ideology of Section 1.3, not only objects—numbers, vectors, quantities, characteristics, etc.—but also all the math operations should be created because one without the other is meaningless. The numerical object of the theory—the complex pair $(\pm\mathfrak{p}, \pm\mathfrak{q})$—is as yet single, and it contains a principally "non-materializable" ingredient (Section 7.4) and behaves as a whole. With regards to empiricism, the negative and $\mathbb{C}$-numbers are equally "nonexistent, fictitious entities" since the state's mathematics (96)–(98) has not been supplemented with the doctrine of "empirically perceived" quantities (103). As a matter of fact, the step-by-step transformation of the binary operation $\cup$ to symbols $\uplus, \pm, \hat{+}$ and, finally, to operations $\{\hat{+}, \cdot, \oplus, \odot\}$ does not terminate at states. Algebra (96)–(98) will be further required to create the mathematically correct calculation rules of the proper observational quantities.

The foregoing is amplified by the fact that pr. **II** has been involved in the classical description and in vindication/refutation of, say, the hidden variable theories. Here, numbers are identified with the reified quantities, and subtraction is taken for granted from the outset. However, the negative quantities are also being created here, and they are constructed in the same manner as the "quantum zero" for the *H*-group in Section 6.3.

Indeed,

- the instrument indications and physical quantities are not numbers, nor the ("pointer") states;

"detector . . . does not measure a field or an *S*-matrix-element" (R. Haag (2010)). They are no more than notches, and "negative notches" are introduced prior to mathematics of symbols according to the following subconsciously intended scheme. The self-apparent physical

conventionality, which has been called "an addition" of two such notches, must produce, in accord with the supra-mathematical requirements of physics, what is named "nought, zero". Two waves at a point, for instance, compensate each other. The result is asserted to be identical with a mathematical zero, and that is the subtraction.

The classical "explanations" are the ones that use compensations/subtractions (see Remark 13), whereas *the minus we have become accustomed to is a fairly abstract construction in its own right*. J. Baez and J. Dolan best reflected the situation, observing on page 37 of [85] that "half an apple is easier to understand than a negative apple!"; on the same page, a good discussion of division is given. In this respect, one might state that the very classical physics needs an interpretation in terms of strictly positive "the number of Stücke". The mathematization of empiricism into numbers is not a distinctive feature of a quantum description. However, comprehending "abstracting the minus sign" is not confined to this. A word of explanation is necessary with regard to the situation.

Mathematics formalizes [134] the positive/negative $\pm\mathsf{p}$ into the pairs' classes $(m, n)$ being equivalent with respect to an "adding" of the class $(\ell, \ell)$ (the "zero"):

$$(+\mathsf{m}) := (m, 0) \approx (m + \ell, 0 + \ell), \qquad (-\mathsf{n}) := (0, n) \approx (0 + \ell, n + \ell),$$
$$\pm\mathsf{p} \quad\Longleftrightarrow\quad (\mathsf{m} - \mathsf{n}) := (m, n) \approx (m + \ell, n + \ell), \tag{104}$$

where $m, n, \ell$ are to be seen as "something strictly positive". This "adding" is yet another tacitly assumed and much more abstract action: the addition of objects of some other kind—"positive couples" $(m, n)$. Technically, at an appropriate place in Section 7, we had to introduce such classes and to assign their own algebraic operations for them. The result might be called the "genuine" arithmetic (of "the positives") and could be enlarged to the "complete arithmetic" with multiplication and division.

As a consequence, the single-token object $(+\mathsf{m})$ or $(-\mathsf{n})$, which we perceive as self-evident (cf. pr. **II**), is a highly unobvious construction—the generic equivalence-class of two-token $(m, n)$-abstractions (104). The essence of the symbol of a negative number $(-\mathsf{n})$ is revealed only when contrasting the two positive ones. Exactly the same situation has occurred when deriving the superposition principle in (49), (50), and (55).

It is clear that once all the $\pm\mathsf{p}$-numbers and the "normal" positive $+\mathsf{p}$'s among them have been formalized into the equivalences (104), the fact that they possess any "natural meanings", such as the "operation of the quantity $\mathsf{p} \mapsto |\mathsf{p}|$" invented above, becomes more than unnatural; the abstract class operations appear out of nowhere. Similarly with $\mathbb{Q}$-numbers and their $\mathbb{R}$-extension: classes of equivalent pairs $(\mathsf{n}/\mathsf{m}) := (n, m) \approx (n\ell, m\ell)$.

*9.3. Naturalness of Abstracta*

We thus infer that the rejection or disregard of the similar "naturally abstract" set-theoretic models would be tantamount to the rejection of the minus sign even in elementary physics. This is absurd, but its root is a need for abstracting. On the other hand, the motivated deduction of these models cannot be replaced with (hidden) axiomatic assumptions or with ready-made math-structures. Such an ambivalence is, in our view, one reason why the problem with "decrypting" quantum postulates is so difficult; it touches on the metamathematical and metaphysical aspects of the very thinking [84,105,148,169,179]. The stream of subconsciously abstractive homomorphisms

$$\lceil \text{pt. } \mathbf{S}, \mathfrak{T} \rceil \rightarrowtail \{\underline{\alpha}, \underline{\Psi}\} \rightarrowtail \{\underline{\Psi}, \cup\} \rightarrowtail \cdots \rightarrowtail \left\{ (\Xi)_{\mathscr{A}}, \overset{\mathscr{A}}{\uplus}; \dots \right\} \rightarrowtail \cdots \rightarrowtail$$
$$\left\{ \begin{bmatrix} \mu \\ \lambda \end{bmatrix} \underline{\alpha}, \uplus \right\} \rightarrowtail \cdots \rightarrowtail \{\mathfrak{a} | \underline{\Psi}), \pm\} \rightarrowtail \left\{ \{\overset{\underline{\Psi}}{\mathsf{m}}, \overset{\mathfrak{a}}{\mathsf{n}}\} \underline{\alpha}, \hat{+} \right\} \rightarrowtail \{\mathfrak{a} \cdot |\alpha\rangle, \hat{+}\} \rightarrowtail$$
$$\rightarrowtail \{\mathfrak{a}, |\mathbf{\Psi}\rangle; \hat{+}, \cdot\} \implies \mathbb{H} \tag{105}$$

is considerable and is always larger than it seems. In Sections 3–9, we have described not all of them. Each such homomorphism is a mapping into a representation by a model, and for a philosophical discussion of these representations and the origin of these models, see pages 1–230 in [160]. For another comment concerning abstracting/realism, we refer the

reader to the first half of a letter from A. Einstein to H. Samuel in [189] (pp. 157–160); see also [180].

Thus, "difficulties" with complex numbers, stricto sensu, should already be attributed to the level of the usual negative ones. Bearing in mind that the minus comes from the equal sign = [86] and the equality comes from the scheme (49) $\rightarrowtail$ (50), both principia **II** and **III** are very important (and functioning) in the classical case. In quantum case, they are just fundamentally unavoidable for the very creation of the theory. The nature of QM theory, of arithmetic, of complex numbers, and of their algebras is one and the same.

Transferring the reasoning above to the natural numbers $\mathbb{N}$, the degrees of classical and quantum abstractions become indistinguishable. Empirical motivation leads, in one way or another, to the standard von Neumann's representation for ordinals

$$0 := \varnothing, \qquad 1 := \{\varnothing\}, \qquad 2 := \{\varnothing, \{\varnothing\}\}, \qquad 3 := \{\varnothing, \{\varnothing\}, \{\varnothing, \{\varnothing\}\}\}, \qquad \dots \,, \quad (106)$$

i.e., to using the ZF-axiom of union: $n + 1 := \{n\} \cup n$ [134]. Therefore, the $\mathbb{N}$-numbers are less obvious themselves, followed by the ordering $<$, topologies, extensions, generalizations, etc. The formal characterization of all the experimental values reduces to the successive creation of the set-theoretic atoms—unions of sets—some direct products thereof and mappings into other constructions of the same kind. Hence, both the physical images "being under a ban above" and auxiliary structures—dimensionalities/orders, etc.—should equally become homomorphisms onto certain formal constructions regardless of the description's classicality/quantumness. The presence of, say, non-binary operations (88) does not stand out because their nature does not differ from the one of habitual subtraction and of division. All of these are involutory structures that have been mathematically inherited from the empirical meta-requirements: repetitions (**M**-paradigm), experimental context $\langle\!\langle \mathcal{S},$ **M**$, \dots \rangle\!\rangle$, and covariance **III** (Section 5.4).

To close the section, we add that the distancing of concepts of state/`DataSource` and of a physical property is the continuation of a more primary idea—detaching the proper macro-perceptions from what is being *represented* theoretically [87,93,160] and from conceptualizing the notions [84]. As B. Mazur has noted in [86] (p. 2), "This issue has been with us ... forever: the general question of *abstraction*, as separating what we want from what we are presented with", i.e., the separating the "bare" empiricism from mathematics with $\Sigma$-limit and the number.

The atomic constituent of perception—sensory experience—is an elementary quantum event [93,97,123,126], and it begins and terminates in the $(\not\approx_{\mathscr{A}})$-distinguishability of $\underline{\alpha}$-clicks (Section 2.1). Any continual is a "speculative theory" (act of abstracting), not the underlying empiricism. Therefore, all the further matters—numbers, arithmetic, cause/effect, (non)inertial reference frames, the notions of an observer, of a classical event in the Minkowski space, the spacetime concept itself and coordinates in the relativity theory (a quantum view of the equivalence principle), device read-out, tensors, composite systems, symbols $\otimes$, and the like—self-evident as they may seem, are the math add-ons, which could originate only in the "$\cup$-theory" of Section 5. Following von Weizsäcker, it might be coined the name "Ur-theory". There are no contradictions in the observations themselves, regardless of whether we call them macro- or micro-scopic. Contradictions do arise in the "mathematicae being constructed".

## 10. About Interpretations

> It is ... not ... a question of a re-interpretation ... quantum mechanics would have to be objectively false, in order that another description ... than the statistical one be possible—J. von Neumann ([25], p. 325)

> ... one begins to suspect that all the deep questions about the meaning of measurement are really empty—S. Weinberg

"At this point in time it appears that a stalemate has been reached with regard to the interpretation of quantum mechanics" (E. Tammaro [190] (p. 1)). A "stalemate in which

each side refuses to cede territory but is unable to produce a defining argument that would change the hearts and minds of the opponents" (M. Schlosshauer [16] (p. 227)).

*10.1. Click, Again*

The source of the "foundational skirmishes" [16] (p. 227) and the numerous treatments of QM [7,8], [9] (Ch. 10), [30] (Ch. 10), [33,112]—"the Copenhagen" among them—is the fact that the $\underline{\alpha}$-event and intuitive sense of the $\underline{\Psi}$-primitive (pt. **S**) are a priori endowed with physical properties, observational/determinative characteristics of the `DataSource`, and operationality of the canonical QM-concept of the ket-vector $|\Psi\rangle$. A representative example in this regard is one of the first sentences from Everett's PhD: "The state function $\psi$ is thought of as objectively characterizing the physical system ... at all times ... independently of our state of knowledge of it" [107] (p. 73), [124] (p. 3), and also, on p. 48, "The general validity of pure wave mechanics, *without any statistical assertions*, is assumed for *all* physical systems, including observers and measuring apparata". Furthermore, again the Everett's: "The physical 'reality' is assumed to be the wave function of the whole universe itself" [67] (p. 100), [107] (p. 70). However, none of these initially exist. The primitive $\underline{\alpha}$-events' abstractions $\underline{\Psi} \dashrightarrow \underline{\alpha}$ are all there is.

An important point is that the eigen $\underline{\alpha}$-click (of a photon/electron in the EPR-experiment, say) should not be identified with an $|\alpha\rangle$-state. The latter is re-developable with respect to eigen-states associated with other click-sets of any other instruments:

$$|\boldsymbol{\alpha}_1\rangle = \mathfrak{b}_1 \cdot |\boldsymbol{\beta}_1\rangle \,\hat{+}\, \mathfrak{b}_2 \cdot |\boldsymbol{\beta}_2\rangle \,\hat{+}\, \cdots = \mathfrak{c}_1 \cdot |\boldsymbol{\gamma}_1\rangle \,\hat{+}\, \mathfrak{c}_2 \cdot |\boldsymbol{\gamma}_2\rangle \,\hat{+}\, \cdots = \cdots ,$$

which is why it is logically meaningless to attribute one $\underline{\alpha}$-click to that which carries the statistics of other clicks $\underline{\beta}_k$, $\underline{\gamma}_k$, ... and has nothing in common with $\underline{\alpha}$. All the more so the click may not be related with the physical texts or physically descriptive collocations, such as "the measuring act on Bob's electron reveals the spin-up state". As in the "cat case" (Section 6.3.1), the spin-up here is a click-up rather than a state $|\uparrow\rangle$. Similarly, a click (allegedly of a photon) with Alice/Bob has nothing to do with distance between photons (the locality "problem"), with speed of light, nor with a kinematic "understanding" of the photon.

The quantum-detection micro-event is not a classical one as we have been understanding it, say, in the special relativity. The "click does not establish the presence of something" [126] (p. 761), it "is an elementary act of "fact creation."" [23] (Wheeler). The facts and phenomena are made up of clicks. That is to say, the distinguished $\underline{\alpha}$-clicks are not the events spaced at some distance from each other or at different points in time. These are just clicks without accounting to them such descriptive notions, such as distance, coordinates, point of time $t$, or the picturesque words, such as "dead/alive/.../cat"; of course, the click itself has no size/duration. Exempli gratia, the particles at accelerators and their physical properties are observed not as "material bodies in the proper sense of the word" [109] (p. 62)—this is impossible—but rather through the abstract detector-snaps. Neither the electron in interferometer nor the Higgs boson at a collider are observed in a detector as objects that are finite in extent; they are not observable entities. "A Higgs" is just a frequency $5\sigma$-histogram at LHC.

Likewise, the math-properties of the eigen $|\alpha\rangle$ and of the abstract $|\Psi\rangle$ can in no way be "syncretized" with $\underline{\alpha}$-events when they are still being accumulated. The screen scintillation is not a photon and a photon is not a scintillation. Similarly, "the arrival of an electron" [164] (p. 3) at the screen does not mean "what is here at this point in time with a given coordinate is the materialized particle-electron". $|\alpha\rangle$-states and $\underline{\alpha}$-clicks are accompanied by the phenomenological and dramatic words "up/down/.../alive/dead", and this has nothing to do with physics, which is yet to be created. The click should not be an element of the language in which $|\Psi\rangle$-terminology, numbers, and physical properties have been employed whatsoever.

### 10.2. Abstraction the State

Then, something subsequently referred to as a state (the abstract) and a measurement (the concrete) is created. However, as already stressed in Section 2.4, the process of abstracting is a rather multistage one (Sections 3–9), and a reduction in the long sequence (105) "for physical reasons" does always contain phenomenological axioms a priori. Clearly, in the reverse direction, we confront hard-to-disentangle assumptions and the well-known axiomatic cycle. The physical considerations and phenomenology should not be present in fundamentals of quantum mathematics.

To avoid paradoxes with "quantum cats", "state vector does not describe ... a single cat" [68] (p. 37)), and "One cannot think about it as in a superposition" [16] (p. 134; D. Greenberger), with "the presence of a particle here and there", or with "quantum bomb-testing" (Elitzur–Vaidman) [27,33]:

- It is imperative to keep a severe conceptual differentiation [3] (first column) between the term "the state" and "physically sounding" adjectives/verbs and the spatiotemporal or cause-effect images.

Similarly T. Maudlin: "... we need to keep the distinction between mathematical and physical entities sharp. Unfortunately, the usual terminology makes this difficult" [151] (p. 129). Even indirect usage of the terminology borrowed from the classical description can be a source of confusion. For example, a so-called exchange interaction as a "cause of correlation" between identical parts of system.

It seems preferable to radicalize the non-connectivity of these categories, i.e., to proclaim it a postulate. For instance, boldface italics in Remark 12 or the selected thesis on page 41. At least, the differentiation between them should not be neglected in reasoning, inasmuch as it seems unrealistic to change the deeply ingrained [59] (p. 7) and ill-defined terminological locutions, such as a "photon is in a certain *state of polarization ... one* photon being in a particular place" [26] (p. 5, 9), "an observable has/acquires a (numerical) value when being measured" [92] (p. 310; criticism), "outcome of a measurement" [9,30,33], "quantum parallelism" [30] (p. 282), or "simultaneous measurability"; see Section 2.1 and pr. **I**. (As if we have had some micro-physics prior to math; there is nothing a priori.) With this mixing, the circular logic pointed out in Remark 10 will be present at all times. See, for instance, pages 29–30 in the work [74] and notably an emphasized warning by D. Foulis about "*a mistake, and a serious one*!", including criticism addressed to von Neumann on p. 29. This

- trap of the "*braket*ting the ClassPhys'—|physical words⟩ or |in⟩/|out⟩—is the very "somewhere ... hidden a concept" that M. Born spoke of (Section 1.3, p. 4), i.e., the mistaken "*physicality of* $|\psi\rangle$ *and of* $+$" in (1).

Again (see p. 23 and Section 6.4), even the indirect attempts to physically characterize the state function or "reconcile" its non-classicality with any *observational prototypes* are hopeless. "The wave function is in the head and not in nature" ascribed to A. Zeilinger (2014) by A. Khrennikov. The function is the very information DataSource around which all sorts of words on physics—readings, frequencies, objects, phenomena, particles, events, and other entities—are only slated to create.

- "We cannot ... manage to make do with such old, familiar, and seemingly indispensible terms" (Schrödinger (1933)) as the "" physikalische Realität " .... " Realität der Aussenwelt ", " Real-Zustand eines Systems "" [89] (p. 34) in the way we are doing this in classical physics, even philosophically. To put it both informally and more precisely, the automatic speech—stereotype—"the system in a state" (pt. **S**) [93,94] (criticism) should be dismissed from QT-fundamentals because the microscopy of quantum $\underline{\alpha}$-clicks shows that this colloquial habit is an unmeaning collocation.

This term may only be a theoretical conventionality in the follow-up *physical* theory. See also the first sentence in [61] and the selected theses on page 23 and at the beginning of Section 2.4.

The principled abstractness of the $|\Psi\rangle$-object [13] (pp. 27–28) is a core attribute of quantum theory as contrasted to the classical one. This abstraction cannot be "struggled"; it is not an idealization of something phenomenological. It is absolute. An interpretative comprehension such as $|\text{dead}\rangle \,\hat{+}\, |\text{alive}\rangle$, even if it is permissible, may issue only from the $|\boldsymbol{\alpha}\rangle$-representations $\mathfrak{a}_1 \cdot |\boldsymbol{\alpha}_1\rangle \,\hat{+}\, \mathfrak{a}_2 \cdot |\boldsymbol{\alpha}_2\rangle \,\hat{+}\, \cdots$, i.e., from a treatment of the $(\hat{+})$-addition (of quantum amplitudes) as an accumulation of $\underline{\alpha}$-microevents—many "cat boxes".

- In other words, *the* interpretation of the quantum state *is its very definiendum* (96)–(98). Even with the physical terminology created, there may be only one paraphrase for the meaning to the state: an abstract element of the abstract, linear (not Hilbert [130]) vector space over $\mathbb{C}^*$. (Point (4) in *Theorem* determines a supplement—the number add-on over the utterly abstract LVS.)

The "not Hilbert" here is because the norm and inner-product are the extra, nonessential math add-ons over $\mathbb{H}$ [130], which come from the follow-up introducing the Born statistics [6]. In and of itself, the state needs none and knows nothing of them. These concepts, similarly to the descriptive physical notions and a measurement, will be required further but not now for the calculation of observable quantities: math-calculus of statistics $\mathfrak{f}_k$ and of means.

We may not blend the fundamentally abstract part of quantum mathematics—pre-physics and the structural properties of $\mathbb{H}$—with those in charge of its observational/physical constituent, i.e., we may not ascribe the ontological status [95,186] to everything. In the strict sense, the ontology of/and physics, the classical one included, cannot arise before the statistical processing of quantum micro-events. (Parenthetically, the sixth Hilbert problem on "Mathematical Treatment of the Axioms of Physics" [191] becomes an ill-posed problem ([130], Section 8).) The processing itself begins with the Born rule [6].

Continuing Scheme (68), a certain parallel takes place between the following couples:

$$\text{observations' language:} \quad \lceil \mathbb{R}\text{-numbers} \rceil + \lceil \text{physical quantities}/\ldots \rceil$$
$$\downarrow\uparrow \qquad\quad \downarrow\uparrow \qquad\qquad\qquad .$$
$$\text{quantum language:} \quad \lceil \mathbb{C}\text{-}, |\Psi\rangle\text{-objects} \rceil + \lceil \text{physical properties/data}/\ldots \rceil$$

Just as we are not raising the question about the abstractness/treatment of the $\lceil \mathbb{R}$-numbers$\rceil$ in isolation from the $\lceil$physical quantities$/\ldots \rceil$ (Remark 16), so also we should not question a treatment or the physical meaning of the $\lceil \mathbb{C}$-, $|\Psi\rangle$-objects$\rceil$. By analogy, being torn away from the \$-symbol in \$5, the number 5 in and of itself may carry neither the financial nor any other ("bank/(non)commuting$/\ldots$") treatment, nor does it contain some hidden "microeconomic" content. The number has no a "retrograde memory".

The first "summands" in the aforementioned $(+)$-conjunctions are the abstracta of principle. They may exist as the "math-things-in-themselves", and we know that they really do just as we are comprehending the existence of the $\mathbb{N}$-arithmetic that has been constructed in Section 7. The second "summands" are the interpretative supplementations in their own rights. If a theory does not spell out a nature of accounting the second to the first (pr. **II**), then it is impossible to find-out/guess the "true" interpretation or nonexistent physical "protosource" of the abstracta $\mathfrak{n}$, $\mathfrak{a}$, and $|\Psi\rangle$ "ab intra" their algebras (57), (71)–(74), (79)–(81), (88), and (96)–(98) or from the Hilbert-space mathematics. See again Remarks 2 and 16, and warnings by Ludwig of "a mistake. ... false notion that "mathematical objects" must be pictures of physical objects" [94] (p. 228) and of a "reality [of the] word "state," a reality in which one must not believe!" [58] (p. 78). A. Peres also makes special note of the analogous: "... physicists have been tempted to elevate the state vector $\psi$ to the status of substitute of reality" [113] (p. 645); D. Mermin puts this as "a regretable atavistic tendency to reify the quantum state" [23] (p. 144).

*10.3. Measurement "Problem"*

The most representative example is the (in)famous problem of measurement [25] (Section V.4 and Ch. VI)—"tyranny of thinking of von Neumann measurements" [23] (p. 534) with the collapse postulate. This is the subject of an "endless stream of publications suggesting new theories … unending discussions … symposia" [91] (p. 519) and of "the mountains of literature" [94] (p. 118) containing opposing opinions [91] (Ch. 11), [9,18,19,30,45,56,159]. It is, indeed, the source of questions around locality in QM. As we have seen, this problem "is simply not a problem at all!" [50] (p. 1013). It is a nonexistent—"the alleged … does not exist as a problem of quantum theory" [12] (p. 15)—as well as a pseudo problem and a non-issue [93] (p. 79 (!)), [94] (p. 118 (!)) [110], because

- in measurements, nothing propagates (much less at superluminal speeds) or interacts; nothing collapsed [92] (Section XVII.4.3), [9] (p. 328), [58,68], nullified, or localized; there are no such things as quantum jumps [161]; no "pieces" of the wave function are "cut out" [192] (p. 57, 158).

It is no exaggeration to say that the need to projective postulate—"a fruit of realist thinking" [4] (p. 172)—is much the same as the necessity for the world ether supporting the electromagnetic waves. All the more so because such a view of the theme has been present in the literature for quite a while [4,13,15,32,37,93,94,124,188] even as appeals.

"There is nothing … problematic about measurement"

L. Ballentine (1996)

"… there is no collapse of wave packets in reality. Do not believe in fairy tales!"

G. Ludwig [58] (p. 104)

"A state vector … does not evolve continuously between measurements, nor suddenly "collapse" into a new state vector whenever a measurement is performed"

A. Peres [113] (p. 644)

"This "reduction" … is not a new fundamental process, and, … has nothing … to do with measurement"

L. Ballentine [34] (p. 244)

"The mystifying notions arise from attributing physical reality to the "jump" at a given time $t$"

G. Ludwig [92] (p. 327)

"Really bad books … claim that the state of the physical system … collapses into the corresponding $u_n$. This is sheer nonsense. (Finding appropriate references is left as an exercise for the reader.)"

A. Peres (2003)

Englert [12] (p. 8) does particularly object to the "folklore that "a measurement leaves the system in the relevant eigenstate" … It is puzzling that some textbook authors consider it good pedagogy to elevate this folklore to an "axiom" of quantum theory". See also the second epigraph to Section 2.

The point here, put very briefly, is that the measuring the "problem" is one of principle not of practice. Expressed by Bell's words, "the word [measurement] has had such a damaging effect on the discussion, that … it should now be banned altogether in quantum mechanics" [28] (p. 216). J. Bub and I. Pitowsky do insist in the book [8] (p. 453) that presumptions "about the ontological significance of the quantum state and about the dynamical account of how measurement outcomes come about, should be rejected as unwarranted dogmas about quantum mechanics".

Another example of circular logic is the critiqued [30,71,193] meaning of the phrase "an ensemble of similarly prepared systems" [8,30,90]. The revision of this (by and large correct) idea, as was set forth above, does actually demonstrate that, like in the ensemble approaches, "quantum mechanics is a *statistical* theory" [4] (p. 2), [58] (p. 123), [129]

(p. 223), [5,34,40,58,64,90,93,187], with a frequency content of randomness and the classical logic [58] but with a different math-calculus of the statistical weights. The "different" is due to the fact that the theory is not tied, as in the classical description, to the notion of an observable quantity, and the $\mathtt{f}$'s are calculated from the "other/abstract" numbers [6]. However, for the same reason, emphasizing a close resemblance with the statistical mechanics [11,194], ref. [124] (p. 72) and "explanations" with playing cards/dice, coins/balls/.../urns/"socks" [28] (Ch. 16) or with the classical phenomena—unusual as it may sound—are in error. The case in point is not a drastic dismissal of the classical ideas, but rather a "quantum audit" of the classical-physics language [130] (Section 8). The correct "audit" of the classical is a *re*creation of the classical:

$$
\begin{aligned}
&\lceil \underwave{\text{classical phenomena}} \rceil \rightarrowtail \lceil \text{classical events/objects} \rceil \rightarrowtail \lceil \text{micro world} \rceil \rightarrowtail \\
&\quad \lceil \text{micro-event} \rceil \rightarrowtail \lceil \text{quantum micro-event} \rceil \overset{(!)}{\rightarrowtail} \lceil \textit{abstract}\ \text{click} \rceil \rightarrowtail \\
&\quad \lceil \text{abstraction}\ \underline{\Psi} \overset{\mathscr{A}}{\rightsquigarrow} \underline{\alpha} \rceil \rightarrowtail \lceil (\Xi)\text{-brace } (24), \dots \rceil \rightarrowtail \lceil \text{state } |\Psi\rangle \rceil \rightarrowtail \\
&\quad \lceil \text{observable concepts} \rceil \rightarrowtail \lceil \text{observable numbers} \rceil \rightarrowtail \cdots \rightarrowtail \\
&\quad \lceil \text{statistics, the concept of a mean} \rceil \rightarrowtail \lceil \text{state, objects}, \dots, \text{physics} \rceil \rightarrowtail \\
&\lceil \underwave{\text{classical phenomena}} \rceil
\end{aligned}
\tag{107}
$$

and, consequently, the creation of the classical concept of a measuring process. Thus, this scheme along with quanta's statistics and LVS-mathematics all add up to a positive answer to Wheeler's question: "Is the entirety of existence, rather than being built on particles or fields of force or multidimensional geometry, built upon billions upon billions of elementary quantum ..., ... acts of "observer-participancy," ...?" [16] (p. 286), [23].

*10.4. Interpretations and Self-Referentiality*

Although we have not yet touched on other significant attributes—the means over statistics, operators, and products of $\mathbb{H}$-spaces will be considered in their own rights—it is clear that the need to quest for a description in terms of hidden variables is also eliminated. Even from a formalistic perspective, the proof of the presence/absence [25,27,195] of these "physical" quantities should be attributed to the semantic conclusions of meta-theory (=physics) [106], i.e., to theorems *about* formal theory rather than to theorems of its *inner* calculus. In our case, and more generally, the formal theory is the syntactical axioms of QM. The corollaries of such axioms are inherently unable to lead to statements about interpretations [106] since theorems *about* object-theory itself is not provable by means *of* its object-language [106,120]. In a word, the nature and interpretation of axioms are not recovered from the very axioms or from the replacement thereof by the other ones.

A similar line of reasoning has accompanied QT for quite a while: "claim that the formalism by itself can generate an interpretation is unfounded and misleading" [68] (p. 38). It is known that even the mathematics itself cannot be grounded in a self-contained way [98] (p. 201), [105,149], [173] (!). All of this stands in stark contrast with the known statement of DeWitt to the effect that "mathematical formalism of the quantum theory is capable of yielding its own interpretation" [80] (pp. 160, 165, 168) or that "conventional statistical interpretation of quantum mechanics thus emerges from the formalism itself" [80] (p. 185). In particular, if we take account of the fact that it is not the theory itself but only its formal interpretation that determines the very semantic terms truth/falsity/provability of sentences (K. Gödel). In turn, "interpretation ... allows a certain freedom of choice" [78] (p. 310). See also [96] and specifically Ch. III in [105]. In other words, the subconscious striving for "to interpret" and transporting the macro into the micro is the very thing that prevents us from truly gaining an understanding of quantum mathematics.

In any case, the fact that we were initially constructed the set-theoretic model (cf. [96]) rather than an interpretation simply eliminates the problem or, at most, transfers it into the domain of questions about micro-transitions $\overset{\mathscr{A}}{\dashrightarrow}$ and $\mathfrak{T}$-family as entities being employed (see Remark 2). This is the domain of questions that invoke the set theory and touch on

the ontological status of sets at all [149] (Sections V.8 and 9 (!)). Be this as it may, logic—formalized or not—does not allow us to make statements about statements, much less a statement that refers to itself. The self-referentiality ("von Neumann catastrophe") is almost the chief trouble [4,18] encountered in quantum foundations.

All of this, of course, does not depend on whether interpretation is built in a strictly formalized form [120] or in a physically natural one. In effect, the issue of interpretations—in the rigorous definition sense [106] (Ch. 2), [120], ref. [136]—is simply nonexistent. Accordingly, the demystification of the known and the quest for ontological interpretations to 𝔞-coordinates of the $|\Psi\rangle$-vector [33,165,188]—the wave function—is no longer a problem, and with it, disappears the Feynman question of "the *only* consistent interpretation of this quantity" [164] (p. 22). See also M. Leifer's review [186] and extensive list of references therein.

## 11. Closing Remarks

> ... quantum mechanics has been a rich source for the invention of fairy tales—G. Ludwig and G. Thurler ([58], p. 122)

> I simply do not know how to change quantum mechanics by a small amount without wrecking it altogether ... any small change ... would lead to logical absurdities—S. Weinberg (1994)

### 11.1. Language and "Philosophy of Quanta"

Remembering and continuing Section 1.3, it is generally tempting to infer that when creating the theory, we may not rest on any meanings that are tacitly associated with the typical terminology—no matter physical or mathematical—and on the tacit assumption that customary concepts are substantially correct [98].

One should also be very cautious about the wording of statements concerning the phenomena outside the everyday experience. One means that even the very natural utterances—"here/there, electron *with* Alice/Bob" (locality), "big(ger)/small(er)", "let there be a two-particle $\mathcal{S}$" (quantitative statements), "subsystem $\mathcal{S}_1$ in such-and-such system $\mathcal{S}$, consisting of ..." (statements about structure)—are de facto "(apparently) "plausible" conclusions from the observed phenomena" [92] (p. 334). These have comprised an equivalent of a measurement/preparation ([4] (pp. 195–196) and Section 3.1) and of physical (pre)imagery and thereby imitate the way of thinking and schemes of classical mechanics; see also the second epigraph to Section 6.

- The "particle, here/there, big/small, this/that/another one, before/after", and the like are *already* "illegal" observations, numbers of sorts, and a premature arithmetization, i.e., this is *already* the subconscious quantifying the micro-events or the arrays thereof by a theory and classical (18) and (19) at that.

Reality's attributes are only slated to create. Say, when we decrease the particles in experiments and reach the atomic level, we still stay in the atomistic paradigm of the particle and of numbers: the objects having mass, their coordinates, degrees of freedom, etc. This is a mistaken intuition. Very informally, we should "religionize themselves" to the quantum micro-events, while the return to the words "particle/.../macro" must be performed by a new reasoning mechanism. It comprises, apart from the quantum-LVS apparatus (Section 8.1), the re-creation of the very classical concept of the particle, as schematized in (107).

At the other extreme is an attempt to "hurry up" and bring the reasoning to a Hilbert-space theory or to the quantum mixtures (102). As in Section 6.5, all this may well be incorrect [151]. A source of antinomies is implicit, implying, i.e., in the eclectic—this we stress once again—confounding the observations, clicks, numbers, physics, time, math, and imagination, followed by the *uncontrollable* lexical-"branching", such as *replacing the symbol $\hat{+}$ with a meaning* taken from reality. For instance, the emerging the word "simultaneously" in the sequence $\lceil$the $(\hat{+})$-superposition of multiple states$\rceil \rightarrowtail \lceil$simultaneously$\rceil \rightarrowtail \lceil$quantum parallelism$\rceil$ [152] (p. 26). W. James has underscored that the "*viciously*

*privative employment of abstract characters and class names* is, ..., one of the great original sins of the rationalistic mind" [23] (p. 547). This results in the sense messes, well-known no-go (meta)theorems [28,44], the locality "problem" in QM, and paradoxes such as the EPR [92] (Section XVII.4.4) or the jocular Bell question: "Was the world wave function waiting to jump for thousands of millions of years ... for some more highly qualified measurer—with a Ph.D.?" [28] (p. 117), [9] (p. 18), [15,37]. As to the no-go theorems, Ballentine remarks that "the growing number [thereof], combined with some peculiar terminology, has led to confusion ... A woefully common feature, ... each protagonist had some interpretation of the quantum state in mind, but never stated clearly what it was" [133] (pp. 2 and 6). Ludwig, echoing Ballentine, asks: "However, what do we mean by the notion of a state?" [87] (p. 5).

Clearly, the quantum-clicks do not depend on whether personified homo sapiens interpret the arrays thereof or a biological observer such as a "Heisenberg–Zeilinger dog" [196] (pp. 171–174) [12,65] does simply perceive. The observer—without their "subjective features" [98] (pp. 55, 137) or "the anthropomorphic notions "specifying" and "knowing" "[113] (p. 645)—is just a formally logical element **O** in theory. Without numbers, solely a quantitative theory is not possible (Section 6.5) because the entire terminology becomes indefinite.

Thus, once a mathematics and *unambiguous* language—spectra, means, and macroscopic dynamical models—have been created, not only is there no longer a need to call on the "otherworldly", eccentric, or anthropic explanations, but the very presence of a certain share of mysticism, of subjectivity, and of (circum-)philosophy [192]—"a philosophical *Überbau*" [12] (p. 12)—in quantum foundations becomes extremely questionable. Ludwig is much more thoroughgoing in his assessment of the language games, which he refers to as the "philosophical gymnastics" [93] (p. 79).

Eventually, we no longer have any freedom to invent exegeses of the quantum-postulate as "a Bible" or "a sacred text" [23] (p. 1038). Moreover, the liberty to ask questions is no longer there since the created object-language of states, of spectra, and of frequencies narrows down the entire admissible lexicon. It is able to generate questions that are not only ill-posed but must, as in Section 6.5, be qualified as "meaningless" [14] (p. 422). For example, those that are based on (human-beings') *intuition* taking the term observation or questions about "the underlying nature of reality". As we mentioned earlier, the notion of "a physical level of rigor" (in reasoning) and the physical justification will not help us with regard to the grounds of QT. Another example is the attempts at (or "to refrain from") "tying description to a clear hypothesis about the real nature of the world" (Schrödinger (1933)) and, in general, the question of "how it should function" at the micro-level. See also [58] (p. 100) on "reality".

In the classical framework, the language sentences are always interrelated since *all* of them, one way or another, handle the observational notions. In the famous Como address, N. Bohr had remarked that "every word in the language refers to our ordinary perception". These notions, in medias res, form our natural speech when describing experiments but are inadequate in the quantum [98] (!). That is, these concepts do not make clear the fact that behind the QT are some structureless abstracta, rather than an "improved" physico-mathematical axiomatics or sophisticated math vehicles, e.g., non-commutative calculus; we believe that these are $\{\underline{\Psi} \ \xrightarrow{\mathscr{A}} \ \underline{\alpha}, \ \cup\}$ and procedures (105)—rather than an 'improved' physicomathematical axiomatics or sophisticated math vehicles; e.g., non-commutative calculus.

Language intuition usually makes it easy for us to do away with paradoxes the semantic closedness causes. However, the quantum situation is just one of a misuse of the vocabulary, i.e., when contradictions are inevitable, and this unlimited source of confusion demands *control over the language itself*. One does create the other ("relative") languages within itself [132]: at first, the language of quantum mathematics and thereafter the language of math-physical description and of classical physics, followed by the language of the semantic interpretations. This is just what we call the metamathematics and math-logic [105], discriminating between metamathematics and philosophy [106]. If this is not

the case—the "quantum conclusions" from thinking (even if partly/implicitly) in terms of physical influences between the classical objects (Deutsch's "bad philosophy")—then we obtain an everlasting source of paradoxes since human intuition has roots in the classical world and is a rather problematic and personal category. A. Stairs calls upon "Do not trust intuition" [73] (p. 256) because it is not meant for QM.

Inasmuch as the conceptual autonomy in quantum fundamentals is minimal (Section 2), the quantum scheme of things must commence with an extremely "ascetic" language (Remark 10), and it should be independent of our intuitive knowledge, which "tend to declare war on our deductions" (van Fraassen). To avoid collisions between theory and meta-language, the subconscious striving of the natural language to include one in the other has to be limited. Einstein adds also the situations when "er führt dazu, überhaupt alle sprachlich ausdrückbaren Sätze als sinnleer zu erklären" [89] (p. 33). A. Leggett's comments on "pseudoquestions" and "gibberish" at the end of Section 1.2 may then be strengthened so that the meaninglessness by itself should become a constitutive element of language, including the language of "philosophy of quanta".

- The rudimentary quantum (meta)mathematics creates the notion of a *prohibited* statement/phrase/question, one that is devoid of meaning. These are sentences that involve the classical analogies in the circumvention of 1) the $|\alpha\rangle$-representatives to the non-interpretable abstraction $|\Psi\rangle$ and of 2) the numerical quantities' nature (Section 9.1).

It is appropriate at this point to quote the 't Hooft remark: "I go along with everything [Copenhagen] says, except for one thing, and the one thing is you're not allowed to ask any questions" and the Einstein reasoning on page 669 in the collected articles [131]: "One may not merely ask ... not even ask what this ... *means*". See also Heisenberg's discussion of the problem ⌈language ⇄ concepts⌉ on pages 48–54 in [109], their work [197], the pages 234–235 in [131] with Bohr's appeals regarding the "necessity of a radical revision of basic principles for physical explanation ... revision of the foundation for the unambiguous use of elementary concepts", and their comments on words "phenomena", "observations", "attributes", and "measurements" on p. 237.

The literature on this subject, even taking only the qualified sources into account, is vast [1–3,8,9,16,24,27,29–31,33,41,44,57,64,71,77,119,151,154,159,165] and abounds with terminology—"words, *ostensibly* English" (A. Leggett [9] (p. 300; emphasis ours))—that defies translation into the language of events or of concretization: observer's consciousness, parallel/branching universes/worlds, free will, catalogue of knowledge, world branch, and also such collocations as rational agents, information ("*Whose*" and "about *what*?" [28], by "Bell's sardonic comments" [30] (p. 262)) has been recorded/transmitted/(not)reached an observer (Wigner's friend), teleporting a state, many-minds/worlds/words, quantum psychology, psycho-physical parallelism (in this connection, see [148] (p. 86 (!))), and many other "bad words" by Bell. He italicizes them on p. 215 of [28].

Of course, "without philosophy, science would lose its critical spirit and would eventually become a technical device" [33] (p. 800), but, on the other hand, "the concept of the free will cannot be defined by indications on devices" [94] (p. 151), and "one must not confuse physics with philosophy" [12] (p. 12). Furthermore, yet, we should like to remember a Heisenberg attitude [197] on "a misconception ... [and 'possibility'] to avoid philosophical arguments ... and the way of thinking of ... physicists who insisted on not dealing with philosophy". Namely, "[w]e can not avoid using a language bound up with the traditional philosophy". One cannot but mention the Rovelli article [198] that is entirely devoted to this topic. Therefore, "[i]t must be our task to adapt our thinking and speaking—indeed our scientific philosophy—to the new situation" with regard to the *abstract* meaning of the linear quantum addition $\hat{+}$ and quantum math altogether; all of the quotations are from pages 32 and 37–38 of the work [197].

As concerns the attitudes towards QM—at the suggestion of M. Tegmark in the 1990s, polls and statistical analysis of their correlations were even carried out [7]. There are also known attempts to involve here the biology of consciousness/brain [71], [119] (Ch. 9), [125],

[199] (Section 6). Regarding them, however, there have been not merely skeptical but quite the opposite opinions [94] (Section XII.5 (!)), [200] (Sections 17.5–6). Of special note are Ballentine's remark "to stop talking about "consciousness" or "free will"" on the last page of the preprint [133] and Popper's criticism of "the alleged ... *intrusion of the observer, or the subject*, [or of consciousness] *into quantum theory* ... based on bad philosophy and on a few very simple mistakes" [108] (pp. 11, 17, 42; everything as in the original) with an appeal "to exorcize the ghost called "consciousness" or "the *observer*" from quantum mechanics" [108] (p. 7). "[Q]uantum mechanics is a physical theory, not psychology" [4] (p. 83).

### 11.2. Math-"Assembler" of Quantum Theory

As a result, we gain "a contribution to philosophy, but not to physics" [82] (p. 86). At the same time, the proposed math "∪-assembler"

$$\underline{\alpha}_j \not\approx \underline{\alpha}_k, \qquad \underline{\Psi} \xrightarrow{\mathscr{A}} \underline{\alpha}, \qquad (\underline{\Xi})\text{-brace (32)}, \qquad (\vee, \in, \cup)\text{-logic (37)–(40)}$$

is quite sufficient for creating the object-language. Giving a natural form to it would be acceptable; however, it is clear that the set-theoretic ∪-base of the language cannot be avoided [96,149]. Nevertheless, the syntactically more formal description of the sequence ⌈transitions ↣ brace ↣ numbers⌉ is surely of interest until the way of looking at quanta's mathematics is harmonized with the math-logic. This would turn, however, all the above material into a pure-logic text, which we eschew in the present work. It is probably for this reason that the very important and extremely thorough works (Pre-theories, 76 axioms [93] (p. 241), ordered sets, morphisms, absence of the word superposition in monographs [87,93], the (valid) criticism of "theories of ... so-called states" [58] (p. 78), etc.) by Günther Ludwig [87,92–94] and by their school are often left out of the literature on quantum foundations. Among other things, in spite of explicitly pointing out a "solution in principle of the measuring problem" in [93] (p. V) and "Derivation of Hilbert Space Structure" of [93], this author has not been mentioned in the detailed reviews [112,118,186] or even in the books [2,5,8,18,31,119].

### 11.3. Well, Where is Probability?

An answer to this question in *quantum* elements is brief enough—nowhere. "There is no probability meter" [8] (p. 185; S. Saunders), and the relationship of this concept with empiricism is unique [34] (p. 46)—the statistical proportions $\mathfrak{f}_k$. Cf. the famous de Finetti's (1970) claim that "probability does not exist" [2] and A. Khrennikov's remarks to the effect that "*the only bridge between "reality" and our subjective description is given by relative frequencies*" [23] (p. 139) and that "Experimenters are only interested in ... frequencies" [150] (p. 36). Moreover, more carefully stated by von Mises' words,

> "If we base the concept of probability, *not on* the notion of relative frequency, ... at the end of the calculations, the meaning of the word 'probability' **is silently changed** from that adopted at the start to a definition based on the concept of frequency" ([129], p. 134; all the emphasis ours).

Indeed, suppose that the word "frequencies" has been banned [19] (p. 44) in substantiating the QT-elements and so have the usage of the words "over/repetition/.../statistics". Then, the questions do immediately arise: why the Kolmogorovian axiomatic, and why does it have this very quantification? In other words, why zero/one/.../positive? Why not the $(-1 \ldots 1)$-interval? Whence the single-case probability postulates? ... subjectivity? Well, what is the quantification thereof, and what does subjectivity do in the natural-scientific theory?

> "... it is very doubtful that quantum probabilities can be introduced as a measure of our personal belief. Well, it may be belief, but belief based on frequency information"
>
> A. Khrennikov; Växjö Conference (2001)

One way or the other, quantum foundations *would demand an interpretation* of Kolmogorov's axioms (besides, these are not categorical in contrast to LVS), and the latter, in turn, demand interpreting the concept of the number—an axiomatic add-on *over* the ZF theory [134].

Bearing in mind the primary nature of numbers and nontriviality of their emergence in physical theory (Section 7.2), it is not just impossible to avoid the statistical weights $\mathfrak{f}_k$ [121] (p. 25). Logic also forbids them from being subsidiary with reference to probability in any definition: "probability is the picture for reproducible frequencies; and it is the [only] *prescription* for a *correct* experiment" [94] (p. 144). Pauli, among the few, had been "convinced that

- the concept of 'probability' should not occur in the fundamental laws of a satisfying physical theory".

(an excerpt from their 1925 letter to Bohr)

Ensemble empiricism, for its part, is self-sufficient, and the only conventionality within it is an infinite number of repetitions. In this connection, we cannot agree with a statement of theorem III in van Kampen's work [56] (p. 99) and with further comment as to "a single system" and "calculation of spectra". At the same time, for formalizing the infinite, there is an appropriate axiom *in* the ZF-theory [134,135].

To say all this still informally, any non-statistical/non-ensemble framework for what we have been calling QM-probability does explicitly or implicitly—if the expression may be tolerated—"parasitize" on statistics by addressing the words "repetitions, multiplied, . . . " and, at the same time, does "attract the empirically vague justifications" in terms of anthropomorphic surrogates: potentiality, tendency, propensity, the amount of ignorance, subjective uncertainty [8] or likelihood, degree of belief, and the like [165]. However, even from a philosophical point of view "probability is a deeply troublesome notion" [16] (p. 78; L. Hardy), which is supported by the vast literature on this subject [17,24,33], [66] (!), [81] (pp. 41–43), [170] (Chs. 3–4), [196], [201] (!). According to Deutsch, D. Papineau calls "this state of affairs . . . a scandal" [8] (p. 550).

An Einsteinian "scientific instinct" [30] (p. 174) against the probability is very well known [78], and Pauli, again, had been recollecting their (Einstein's) frequent remarks in this regard: "One cannot make a theory out of a lot of "maybe's" [= probably] . . . deep down it is wrong, even if it is empirically and logically right". More to the point, the question of what exactly is meant by a probability *event*, i.e., "Probability of *what* exactly?" [28] (p. 228), is also a matter of principle. The answer to it, as seen above, is this: "not of the classical events", i.e., "[n]ot of the . . . *being*" [28] (p. 228) such as "QM-cats", "particle is here/there", "roll of the dice", and the like. An excellent text about probability and the aspects of the probability-physical constructs is the work [202]. Its "verdict" concerning the treatment of this concept [202] (Sections 4.5 and 8) is clearly Misesian [129], i.e. the "*ensemble* and *frequency*" [66] (p. xiii).

Thus, to sum up, the philosophy/axioms of probability or its "quantum deformations" should not be present in *quantum* foundations. There cannot be hidden details underlying the quantum probability because the "details" imply some terminology with a classical content. Quantum probability is the statistical regularity. It comes from Kollektivs [129] of abstracta (32) and may only be a shortened term for the relative "frequencies in long runs" (von Neumann) or "the Einstein hypothesis" by M. Jammer [91] (p. 441). The realistic/physical/. . . /pictorial adjectives and descriptive supplementations to the term "long runs" are prohibited. This is why the conventional tractability of the quantum-postulates' mathematics, i.e., the calculation of *probabilities for the classical* events to occur in the reality—"alive/cat/. . . /imploded/bomb"—"is not adequate" neither as a doctrinal point of departure nor as a post-math interpretation. It presents us with a circulus vitiosus of re-exegeses. Fuchs, referring to de Finetti's words in an interview with Quanta Magazine (4 June 2015), prognosticates that this conception "will go the way of phlogiston". The "not adequate" is a R. Haag quotation, and he expresses this "conviction", applying it even to "the conceptual structure of standard Quantum Theory" [101] (p. 743).

The ultimate conclusion completes Remarks 2 and 7. If we accept the set-theoretic eye on things, Section 5.1, by all appearances, provides a positive answer to the question about the rigidity of QT [103]—"change any one aspect, and the whole structure collapses" [57] (p. 1); see also the second epigraph to this section. At least, it is hard to imagine *what any other* axiom-free way of turning empiricism into quantum mathematics would look like, as soon as we abandon the primitive minimality of the scheme

$$\lceil \text{distinguishable micro } \underline{\alpha}\text{-events} \rceil + \lceil \text{ensembles of abstracta } \underline{\Psi} \xrightarrow{\mathscr{A}} \underline{\alpha} \rceil .$$

**Funding:** This research received no external funding.

**Institutional Review Board Statement:** Not applicable.

**Informed Consent Statement:** Not applicable.

**Acknowledgments:** The author wishes to express their gratitude to the QFT-department staff of TSU for stimulating conversations. A special word of thanks is due to Ivan Gorbunov and to Professor V. Bagrov, who initiated considering the matters on quantum logic [72,74,111]. The work was supported by the Tomsk State University Development Programme (Priority–2030).

**Conflicts of Interest:** The authors declare no conflict of interest.

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
