# Peer review of "Linear Superposition as a Core Theorem of Quantum Empiricism"

_universe, doi:10.3390/universe8040217_

Round 1
Reviewer 1 Report
In this paper, the author advocates for a new, so-called "empirical," derivation of the mathematics of quantum theory without assuming the machinery of Hilbert spaces. The author presents in great details, step by step, this new reconstruction of the theory. The manuscript is based on numerous references related to the foundations, interpretation of quantum mechanics, and even to the quantum philosophy. He develops new ideas, referred to as principia I-III, and proves the central theorem on the superposition principle.
It seems that the author believes that the theory may be defined on a purely empirical basis, free of any mathematical prerequisites. Yet, this kind of empiricism has been widely disputed during the whole 20th century by many philosophers of science such as Duhem, Hanson, Quine or Kuhn, by stressing the idea of "theory-ladeness" of any observational statement. On the other hand, the author is of the view that one can do without philosophy.
In a nutshell, the author's development appears quite original, although it is not easy to grasp so lengthy text in a couple of weeks. The reconstruction of the theory the author proposes has certainly a potential, and I consider his new statistical view of quantum superposition, by and large, as a right way of thinking. The text can be improved in some respects, though. In particular, why does not the author refer the famous Hardy preprint "Quantum Theory From Five Reasonable Axioms"? This reference is almost a standard in the field.
The "concept of a system" (sec. 2) is introduced without definition. This might appear surprising in the frame of an empirical approach since other writers advocating for an empirical approach have proposed definitions of this term "the system," for instance (Peres, 1995).
The main author's idea is a detector click and ensembles thereof. In this connection, I read in the last sentence of sec. 2.2 that the "detector click, and can be characterized as the information bit; but not the information itself or the classical bit." I do not get this, even consulting the reference. The information view of quantum theory is actually a burgeoning research field, with a number of results that appear to intersect the author's motivation. This should, to my mind, be reflected in the manuscript to a greater extent.
Also, the author mentions some works on quantum logic but I think it's not enough. Since 1936, this became an important piece of the modern research in the field.
To better reflect the present progress towards foundational problems the author should pay greater attention to the point that the problem with the C-numbers in quantum theory is currently being studied not only from theoretical standpoint. There are purely experimental works, which rule out the real-number QM. See, for example, recent articles in the January'2022 issue of PRL (volume 128) with the numerous Chinese author names.
Finally, I have no substantive objection to publication of the manuscript in Universe.
Author Response
This is reply to referee#1 comment

Reviewer 2 Report
Please, find attached file

Author Response
This is a reply to referee #2

Reviewer 3 Report
I found this paper interesting and I suggest strongly the publication in UNIVERSE.
Author Response
I thanks reviewer for the positive report. Nevertheless, I made some further improvements as much as possible.